# PRIVACY LEAKAGE VIA OUTPUT LABEL SPACE AND DIFFERENTIALLY PRIVATE CONTINUAL LEARNING

## ABSTRACT

Differential privacy (DP) is a formal privacy framework that enables training machine learning (ML) models while protecting individuals' data. As pointed out by prior work, ML models are part of larger systems, which can lead to so-called privacy side-channels even if the model training itself is DP. We identify the output label space of a classification model as such privacy side-channel and show a concrete privacy attack that exploits it. The side-channel becomes highly relevant in continual learning (CL) as the output label space changes over time. We propose and evaluate two methods for eliminating this side-channel: applying an optimal DP mechanism to release the labels in the sensitive data, and using a large public label space. We explore the trade-offs of these methods through adapting pre-trained models.

## 1 INTRODUCTION

Differential privacy (DP; Dwork et al., 2006b) enables training machine learning (ML) models that are robust to various privacy attacks (Shokri et al., 2017; Balle et al., 2022; Haim et al., 2022). However, ML models are part of a larger systems which has other components. These components, e.g., model output filtering or input preprocessing, can leak privacy despite the actual model being privacy-preserving. Debenedetti et al. (2024) calls any privacy-leaking component, a privacy side-channel. These side-channels can completely invalidate privacy guarantees w.r.t. the sensitive data. Regardless whether a concrete attack exists, all operations accessing sensitive data must adhere to the DP definition (see Sec. 2). We focus on the output label space of a classifier as a privacy side-channel.

As illustrated in Fig. 1, there are two training datasets, where the second dataset has one more sample than the first, with a new label that does not appear elsewhere. To classify the first dataset, a classifier must have "Healthy" and "Type I" in its output label space, i.e., the labels which it can classify. To classify the second dataset, the second classifier must also have the "Type 2" label in its output label space. An attacker with prior knowledge about the additional sample can infer which training dataset was used with 100% accuracy (see right of Fig. 1), breaking formal DP guarantees.

This privacy side-channel is relevant to all DP classification problems, but particularly relevant in the context of continual learning (CL; McCloskey & Cohen, 1989; De Lange et al., 2021; Wang et al., 2024). CL studies models that learn from data arriving as a sequential stream of tasks. Therefore, the output label space needs to be constantly updated as new labels appear with the sequential stream of

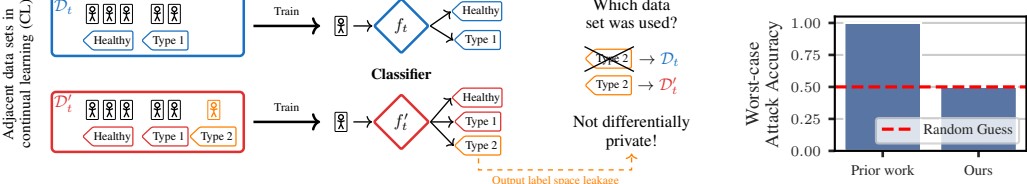

Figure 1: Attack on output label space: Challenger uses either the dataset $\mathcal{D}_t$ or adjacent $\mathcal{D}'_t$ (one more point 👤) for training a classifier $f_t$ or $f'_t$. The classifier output label space is released to an attacker and an attacker can guess the dataset (see right for worst-case attack accuracy). Observing the classifier output space can leak catastrophically when one of the datasets contains one more label ⬭.

tasks. CL methods can run over years, which means that new labels can be unknown at the beginning of training. CL is a difficult problem setting as it faces the challenge of catastrophic forgetting, where a model loses performance on earlier tasks as it learns new ones (French, 1999). Prior work combining DP and CL only considers preventing privacy leakage through the model weights (Farquhar & Gal, 2019; Desai et al., 2021; Hassanpour et al., 2022; Lai et al., 2022). However, naively adding the new labels to the model's output label space during training can lead to a catastrophic privacy failure, even if the model weights are trained with a DP algorithm. This has been overlooked in existing work (potentially due to the focus on standard benchmarks with a fixed list of labels).

To avoid the privacy side-channel, we offer two DP methods to protect the model's output label space. Prior DP CL work uses the entire privacy budget for training weights. However, our first method (Sec. 4.2), inspired by the private partition selection problem (Desfontaines et al., 2022; Gopi et al., 2020; Korolova et al., 2009), splits the budget between selecting class labels from the sensitive dataset and training the weights. Therefore, in this case, there is a trade-off where spending more of the privacy budget on selecting the labels will leave less of the budget for training the weights. The second method that we propose decouples the class labels from the sensitive dataset by using a large data-independent prior labels set that likely covers all or most dataset labels (Sec. 4.3). Unmatched labels can be remapped using known label relationships, or can be simply dropped.

To demonstrate the methods with challenging enough benchmarks, we propose DP adaptations of CL methods based on pre-training. Improving model utility using pre-trained models has been studied separately in DP, and in CL (see Sec. 2). However, to the best of our knowledge, there is no prior work that uses pre-trained models in DP CL. Our contributions can be summarized as follows:

1. **Classifiers' output label space as privacy side-channel:** We show that naively releasing the output label space is not DP (Fig. 1 and Sec. 4.1), as it can leak privacy. We offer two DP alternatives that eliminate the side-channel: (*i*) spending part of the privacy budget on releasing the labels (Sec. 4.2), and (*ii*) leveraging a large public prior label set (Sec. 4.3).
2. **Task-wise DP:** We introduce an adjacency relation for DP CL in Definition 3.2 (Sec. 3) and a generalized version of it in Definition B.4. We also introduce a DP definition for CL in Definition B.5, and show how it is related to the usual DP definition in Theorem B.7 and Corollary B.8.
3. **DP CL experiments:** We adapt CL methods using pre-trained models to DP (Sec. 5) and evaluate the two DP alternatives for choosing the output label space in different CL settings (Sec. 6). We also provide recommendations for choosing among the alternatives (Sec. 7).

We formalised our main propositions and task-wise DP in Lean 4 (Moura & Ullrich, 2021).

## 2 RELATED WORK

**Differential Privacy**  DP provides provable privacy guarantees, but the main challenge with DP is the unavoidable trade-off between privacy and utility. Recently, using pre-trained models has been shown to mitigate this trade-off (Kurakin et al., 2022; De et al., 2022; Tobaben et al., 2023; Mehta et al., 2023; Cattan et al., 2022; Li et al., 2022; Yu et al., 2022; Tito et al., 2024; Wahdany et al., 2024), with most state-of-the-art DP models relying on the assumption that any pre-training data is public. However, if any private information is contained in the pre-training data, the DP privacy guarantees w.r.t. the fine-tuning data become meaningless (Tramèr et al., 2024), but low-risk pre-training data exist (Kuznetsova et al., 2020; Asano et al., 2021). All operations that utilize sensitive data (directly or indirectly) must adhere to the definition of DP. For instance, model selection and hyperparameter tuning can leak privacy (Thakurta & Smith, 2013; Liu & Talwar, 2019). Other interesting examples are data pre-processing (Hu et al., 2024), data discretisation (Ganev et al., 2025), and even the selection of pre-trained model weights (Gu et al., 2025).

**Private Partition Problem**  An example for the private partition problem (Korolova et al., 2009; Gopi et al., 2020; Desfontaines et al., 2022) is creating histograms under DP, as the partitioning into bins can leak privacy if it depends on the sensitive data. We adapt the optimal DP mechanism, in terms of preserving as much classes as possible, by Desfontaines et al. (2022), to determine the output label space.

**Privacy Side-Channels**  Debenedetti et al. (2024) introduce privacy side-channels which are attacks that exploit privacy-leaking components apart from the ML model, e.g., training data filtering or input pre-processing. Through these privacy side-channels, the DP protection of ML models can be invalidated, Debenedetti et al. (2024) show that this can lead to near-perfect membership inference attacks.

**Continual Learning**   Pre-trained models in CL settings have been combined with replay (Ostapenko et al., 2022), prompt tuning (Wang et al., 2022a;b), prototype classifiers (Janson et al., 2022; McDonnell et al., 2023), and expandable Parameter-Efficient Fine-Tuning (PEFT) adapters (Zhao et al., 2024b; Zhou et al., 2024b;a). We propose DP-variants of prototype classifiers and expandable adapters with pre-trained models to enhance the utility and mitigate forgetting with DP. We experiment in both the standard class-incremental learning setting without task labels (van de Ven et al., 2022), as well as in blurry task boundary settings (Aljundi et al., 2019; Koh et al., 2022; Moon et al., 2023).

**Differentially Private Continual Learning**   Previous works combining DP with CL have leveraged episodic memories (Lopez-Paz & Ranzato, 2017) or DP synthetic samples:
- **Episodic memories:** Desai et al. (2021); Lai et al. (2022); Hassanpour et al. (2022) use episodic memories to mitigate catastrophic forgetting. However, this introduces privacy risks.
- **DP synthetic samples:** Farquhar & Gal (2019) train a generative model under DP, while Chen et al. (2022) learn a small set of synthetic samples optimised towards the downstream task. However, these methods have only reported results on MNIST or CIFAR-10. We do not use synthetic data, but recognize this as a potential future work due to powerful DP generation methods (Lin et al., 2024).

Desai et al. (2021) operate in a setting similar to ours but rely on memory buffers. We include their method as a baseline in Fig. 3 and demonstrate that we outperform their method without using memory buffers. Hassanpour et al. (2022) use an easier CL setting where the task label is given at inference time, and their method depends on replay memory and robust continual-learning updates built up during training, which a pretrained model lacks, making the approach incompatible with simple fine-tuning. Lai et al. (2022) also use memory buffers and their method is tied to a specific network design, preventing its application to general or pretrained models.

Another limitation of prior work is the direct link of the output label space to the sensitive data which results in a privacy side-channel (see Fig. 1). Our proposed methods eliminate this privacy side-channel. To the best of our knowledge, we are the first to discuss this limitation of applying DP to CL.

## 3   BACKGROUND

**Differential Privacy**   DP formalises the idea of protecting individuals' sensitive data.

**Definition 3.1** (DP; Dwork et al. 2006b;a).   A randomised algorithm $\mathcal{A}$ is $(\epsilon, \delta, \simeq)$-DP for a binary relation $\simeq$ over the space of possible datasets $\mathbb{D}$, if for any two datasets $\mathcal{D}, \mathcal{D}' \in \mathbb{D}$ that satisfy $\simeq$, denoted $\mathcal{D} \simeq \mathcal{D}'$, and for any subset $S$ of possible outcomes of $\mathcal{A}$, denoted $S \subseteq \mathrm{Range}(\mathcal{A})$,

$$\Pr\left[\mathcal{A}(\mathcal{D}) \in S\right] \leq \exp\left(\epsilon\right) \times \Pr\left[\mathcal{A}(\mathcal{D}') \in S\right] + \delta. \tag{1}$$

Any two datasets $\mathcal{D} \simeq \mathcal{D}'$ are said to be adjacent datasets and $\simeq$ is called the adjacency relation.

In deep learning, the algorithm $\mathcal{A}$ can be, for instance, a DP optimisation method such as DP-SGD (Rajkumar & Agarwal, 2012; Song et al., 2013; Abadi et al., 2016), which minimises the empirical loss under DP constraints. DP-SGD works by clipping per-sample gradients and adding noise to their sum in each iteration, resulting in a set of privacy-preserving model updates.

Applying DP mechanisms multiple times is referred to as composition (McSherry, 2010; Dwork et al., 2010b; Whitehouse et al., 2023). The process of quantifying the total privacy loss under composition is known as privacy accounting (Koskela et al., 2020; Gopi et al., 2021). In CL, each task $t$ is determined by a dataset $\mathcal{D}_t$. In this work, we generalise the standard sample-level add/remove adjacency relation to be applicable for CL. Formally, we introduce this as task-wise adjacency in the following.

**Definition 3.2** (Task-wise Adjacency).   Let $\simeq$ be an adjacency relation between datasets, $I \subseteq \mathbb{N}$ be a set of task indices that can be either finite or infinite. A sequence of datasets $(\mathcal{D}_t)_{t \in I}$ is said to be task-wise adjacent to another sequence of datasets $(\mathcal{D}'_t)_{t \in I}$, if there exists $t^* \in I$ such that $\mathcal{D}_{t^*} \simeq \mathcal{D}'_{t^*}$, and $\mathcal{D}_t = \mathcal{D}'_t$ for all $t \neq t^*$.

Our experiments assume that a datapoint can appear in only one $\mathcal{D}_t$. Since we use task-wise adjacency with only one task $t^*$ having differing adjacent datasets $\mathcal{D}_{t^*} \simeq \mathcal{D}'_{t^*}$, but we do not know which task it is, this corresponds to parallel composition when we protect all tasks. We discuss the limitations of prior adjacency relations for DP CL in Appendix B.1. In Appendix B.2, one can find Definition B.4 which is a generalization of Definition 3.2. We also introduce Definition B.5 which is a DP definition for CL. Theorem B.7 shows how it relates to Definition 3.1. For more general

composition results, especially when an individuals' data can appear in multiple tasks, one can apply Corollary B.8 and any composition result. A full discussion can be found in Appendix B.3. For parallel composition, the privacy level does not degrade with more compositions (see Theorem B.10).

**Continual Learning**   In the general CL setting, we aim to learn a function $f_{\boldsymbol{\theta}} : \mathcal{X} \to \mathcal{Y}$, parameterized by $\boldsymbol{\theta}$, from a sequence of supervised classification tasks (De Lange et al., 2021). Each task $t \in \{1, \ldots, T\}$ (where $T$ can be $\infty$) is defined by a dataset $\mathcal{D}_t = \{(\boldsymbol{x}_t^{(k)}, y_t^{(k)})\}_{k=1}^{N_t}$, where $\boldsymbol{x}_t^{(k)} \in \mathcal{X}_t$ is the $k$-th input, $\mathcal{X}_t$ is the set of inputs, $y_t^{(k)} \in \mathcal{Y}_t$ is the class label, $\mathcal{Y}_t$ is the set of labels, and $N_t \in \mathbb{N}$ is the number of samples. In the CL setting we consider, we have following constraints for the learning algorithm: (i) during task $t$, it can only access data $\mathcal{D}_t$ (ii) it cannot store any samples between tasks; (iii) the total number of tasks $T$ and the complete label space $\mathcal{Y} = \cup_{t=1}^{T} \mathcal{Y}_t$ are unknown in advance. As a result, the algorithm does not directly produce a single final function $f_{\boldsymbol{\theta}}$ after all the tasks, but instead learns an intermediate function $f_{\boldsymbol{\theta}_t} : \mathcal{X} \to \mathcal{Y}_{1:t}$, parametrised by $\boldsymbol{\theta}_t$, after each task $t$, where $\mathcal{Y}_{1:t} = \cup_{i=1}^{t} \mathcal{Y}_i$. We call the range of $f_{\boldsymbol{\theta}_t}$, namely $\mathcal{Y}_{1:t}$, the output (label) space.

In our CL setting, the model does not use the task index $t$ at inference time, and must be able to remember earlier classes. Moreover, a class may appear multiple times across the task sequence. As required by point (ii) in the previous paragraph, for privacy reasons we discard all training samples after completing each task, and consequently do not use any memory buffers or replay data.

**Blurry Tasks**   Prior work in CL (Koh et al., 2022; Moon et al., 2023) differentiates between *disjoint classes*, where the data of a class is only contained in one task, and *blurry classes*, that are split over multiple tasks. The blurry sample ratio controls the blurriness with a value of $0$ being equivalent to having only disjoint classes and $100$, where data of blurry classes is uniformly distributed over all tasks.

## 4   CLASSIFIER OUTPUT SPACE PRIVACY

As discussed at the end of Sec. 3, our goal is to learn intermediate classifiers $f_{\boldsymbol{\theta}_t} : \mathcal{X} \to \mathcal{Y}_{1:t}$ from a sequence of datasets $\mathcal{D}_t, t = 1, \ldots, T$. However, releasing a classifier requires publishing both the model weights $\boldsymbol{\theta}_t$ and the output label space $\mathcal{Y}_{1:t}$. Since each $\mathcal{Y}_t$ is a function of the sensitive data $\mathcal{D}_t$, namely the projection function $(\boldsymbol{x}_t^{(k)}, y_t^{(k)}) \mapsto y_t^{(k)}$, publishing $\mathcal{Y}_t$ leaks sensitive information as a privacy side-channel even if $\boldsymbol{\theta}$ is DP, as illustrated in Fig. 1.

To address this issue, we reframe the problem by denoting the output label space for each task $t$ as $\mathcal{O}_t$. This allows us to separate the released $\mathcal{O}_t$ from the sensitive labels $\mathcal{Y}_t$, and to explore alternative choices for $\mathcal{O}_t$ that ensure the classifier remains DP. Additionally, for each task $t$, we will denote the parameters of the classifier as $\boldsymbol{\theta}_t$ which implicitly depend on $\mathcal{O}_t$. Furthermore, we denote the classifier learning algorithm, which we also call the classifier release mechanism, at task $t$ by

$$\mathcal{M}_t : (\mathcal{D}_t) \mapsto (\boldsymbol{\theta}_t, \mathcal{O}_t). \qquad (2)$$

The following section (Sec. 4.1) explores two alternatives to the leaky $\mathcal{O}_t = \mathcal{Y}_{1:t}$; (*i*) applying a DP mechanism to obtain a DP output label space, and (*ii*) utilising prior knowledge. We note that our theory holds not only for CL but for any classification problem, i.e., when $T = 1$.

### 4.1   CHOOSING CLASSIFIER OUTPUT SPACE $\mathcal{O}_t$

As discussed at the beginning of Sec. 4, releasing the DP parameters $\boldsymbol{\theta}_t$, given a particular $\mathcal{O}_t$, together with $\mathcal{O}_t$ itself might still not be DP. One way to select $\mathcal{O}_t$ is by taking the labels directly from the task's dataset $\mathcal{D}_t$, i.e. setting $\mathcal{O}_t := \mathcal{Y}_{1:t}$. This is the standard approach used in existing work on DP CL (see Sec. 2). For notational consistency, define $\mathcal{O}_t^{\text{data}} = \{y : (\boldsymbol{x}, y) \in \mathcal{D}_t\} = \mathcal{Y}_t$. Generally, we can either set $\mathcal{O}_t := \mathcal{O}_t^{\text{data}}$ or $\mathcal{O}_{1:t}^{\text{data}}$, i.e., either the set of sensitive labels in the current task, or the set of all sensitive labels up to the current task. In the following proposition we show that choosing the output label space based on the sensitive data is not DP. The full proof of Proposition 4.1 is in Appendix C.1.

**Proposition 4.1.** *For any $t$, the classifier release mechanism $\mathcal{M}_t : (\mathcal{D}_t) \mapsto (\boldsymbol{\theta}_t, \mathcal{O}_t)$ is not $(\epsilon, \delta)$-DP for $0 \leq \delta < 1$ if $\mathcal{O}_t = \mathcal{O}_t^{data}$ or $\mathcal{O}_t = \mathcal{O}_{1:t}^{data}$, where $\mathcal{O}_t^{data} = \{y : (\boldsymbol{x}, y) \in \mathcal{D}_t\}$.*

*Proof sketch.* Consider the following counterexample. Let $\mathcal{D}_t \simeq \mathcal{D}_t'$, where $\mathcal{D}_t'$ has one more example than $\mathcal{D}_t$, namely $(\boldsymbol{x}^*, y^*)$. Also, assume that $y^* \in \mathcal{O}_t'$ but $y^* \notin \mathcal{O}_t$. This means that the classifier $\mathcal{M}_t(\mathcal{D}_t')$ has one more label in its range than $\mathcal{M}_t(\mathcal{D}_t)$. This makes the sets of possible outcomes of $\mathcal{M}_t(\mathcal{D}_t)$ and $\mathcal{M}_t(\mathcal{D}_t')$ disjoint, which is not DP. $\qquad \square$

We propose two alternatives for setting $\mathcal{O}_t$. The first is to learn the labels by a DP mechanism, denoted as $\mathcal{O}_t^{\text{learned}}$. The second possibility is to construct a set of labels, denoted as $\mathcal{O}_t^{\text{prior}}$, which reflects our prior knowledge about the task and does not depend on $\mathcal{D}_t$. This knowledge can change and get updated in the following tasks. Thus, we have three possible settings for $\mathcal{O}_t$:

**$S_{\text{learned}}$**: $\mathcal{O}_t := \mathcal{O}_t^{\text{learned}}$, i.e., labels are learned from the private dataset $\mathcal{D}_t$ using a separate DP mechanism.

**$S_{\text{prior}}$**: $\mathcal{O}_t := \mathcal{O}_t^{\text{prior}}$, i.e., a prior label set is provided for each task $t$. The labels in this case are public, and $\mathcal{O}_t$ might contain all or most of the true labels in $\mathcal{O}_t^{\text{data}}$. Unmatched labels in $\mathcal{Y}_t$ can be remapped to one of the labels in $\mathcal{O}_t^{\text{prior}}$ or dropped without changing the DP guarantees w.r.t. $\mathcal{D}_t$.

In Secs. 4.2 and 4.3, we will discuss $S_{\text{learned}}$ and $S_{\text{prior}}$ in more detail, and argue that they preserve the classifier's DP guarantees. For further details regarding DP CL mechanisms, see Appendix B.

## 4.2 DP RELEASE OF OUTPUT LABEL SPACE USING A SEPARATE MECHANISM ($S_{\text{learned}}$)

We introduce a DP mechanism $\mathcal{L}$ that takes any $\mathcal{D}_t$ as input and outputs $\mathcal{O}_t^{\text{learned}}$. Since, in each task $t$, we can group the inputs $\boldsymbol{x}$ into classes according to their label $y$, denoted as $\mathcal{C}_t(y) = \{\boldsymbol{x} : (\boldsymbol{x}, y) \in \mathcal{D}_t\}$, then the problem of releasing the labels is similar to the private partition selection problem (Desfontaines et al., 2022; Gopi et al., 2020; Korolova et al., 2009). In a classification task, each input $\boldsymbol{x}$ is associated with a unique label $y$, and thus every individual's data $\boldsymbol{x}$ contributes to exactly one partition $\mathcal{C}_t(y)$. Under this setup, Desfontaines et al. (2022) proposed an optimal DP mechanism for selecting partitions, which in our case corresponds to the set of labels. The optimality is in terms of preserving as much classes as possible, especially small classes. For each $y \in \mathcal{O}_t^{\text{data}}$, let $Z_y \sim k\text{-TSGD}(p)$ be i.i.d. random variables distributed according to the $k$-truncated symmetric geometric distribution (Desfontaines et al., 2022). The optimal DP label release mechanism can be defined through a noisy thresholding operation

$$\mathcal{O}_t^{\text{learned}} := \mathcal{L}(\mathcal{D}_t) := \{y \in \mathcal{O}_t^{\text{data}} : |\mathcal{C}_t(y)| + Z_y > k\}. \tag{3}$$

According to Theorem 6 of Desfontaines et al. (2022), if we set $p = 1 - \exp(-\epsilon)$ and $k = \left\lceil \frac{1}{\epsilon} \ln\left(\frac{\exp(\epsilon) + 2\delta - 1}{(\exp(\epsilon) + 1)\delta}\right) \right\rceil$, then $\mathcal{L}$ is $(\epsilon, \delta)$-DP. In Appendix C.3, we provide an optimal lower bound on the probability of dropping a class given its size for any DP mechanism, providing further justification on the optimality of this mechanism. We also provide a general method for constructing DP mechanisms that protect the set of labels (see Definition C.2, Theorem C.3). A drawback of this DP mechanism, and any mechanism that provides the labels, is the non-zero probability of dropping classes, which increases as class size or privacy budget decrease, as illustrated in Fig. 2. Another drawback is that the privacy budget must be split with the DP training mechanism, further reducing utility (see Sec. 6.2).

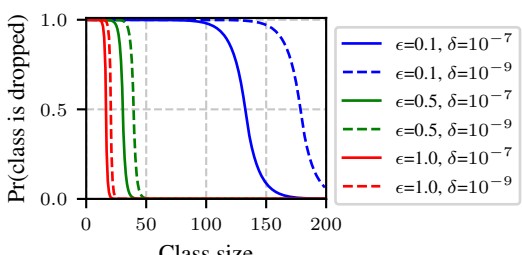

Figure 2: Probability that a new label is not added to the output label space $\mathcal{O}_t^{\text{learned}}$. Even with $\epsilon = 1.0$ and $\delta = 10^{-7}$, classes having fewer than 13 samples are discarded with at least 99% probability, and thus cannot be learned.

## 4.3 DATA-INDEPENDENT OUTPUT LABEL SPACE USING PRIOR KNOWLEDGE ($S_{\text{prior}}$)

Choosing the output label space based on data-independent prior knowledge is DP, since it does not depend on the sensitive dataset. The prior labels can include all or most of the true labels in the sensitive training set. Let $\mathcal{O}_t = \mathcal{O}_t^{\text{prior}}$ (the set of public labels in task $t$). Since $\mathcal{O}_t$ might not contain all labels in $\mathcal{O}_t^{\text{data}}$, we must re-write Eq. (2) to allow remapping the labels not in $\mathcal{O}_t$. Let $\mathcal{R}_t^{\text{label}} : \mathcal{O}_t^{\text{data}} \to \mathcal{O}_t \cup \{\text{drop}\}$ be any function mapping a label $y \in \mathcal{O}_t^{\text{data}}$ to a public label in $\mathcal{O}_t$ or to the special symbol drop, used to discard it. Now, we define the task remapping function $\mathcal{R}_t^{\text{task}}$:

$$\mathcal{R}_t^{\text{task}}(\mathcal{D}_t) = \{(\boldsymbol{x}, \mathcal{R}_t^{\text{label}}(y)) : (\boldsymbol{x}, y) \in \mathcal{D}_t \text{ and } \mathcal{R}_t^{\text{label}}(y) \neq \text{drop}\}, \tag{4}$$

i.e., we transform $(\boldsymbol{x}, y)$ in $\mathcal{D}_t$ to $(\boldsymbol{x}, \mathcal{R}_t^{\text{label}}(y))$, or drop it if $\mathcal{R}_t^{\text{label}}(y) = \text{drop}$. The function $\mathcal{R}_t^{\text{label}}$ may use label relationships (e.g., a hierarchy) to map mismatched labels, drop them, or map them

to a dummy label (e.g., unknown). We show that these operations do not interfere with DP in the following:

**Proposition 4.2.** *For any $t$, let $\mathcal{R}_t^{label} : \mathcal{O}_t^{data} \to \mathcal{O}_t^{prior} \cup \{\text{drop}\}$ be any function. The classifier release mechanism $\mathcal{M}_t : (\mathcal{R}_t^{task}(\mathcal{D}_t); \mathcal{O}_t^{prior}) \mapsto (\boldsymbol{\theta}_t, \mathcal{O}_t^{prior})$ is $(\epsilon, \delta)$-DP w.r.t. $\mathcal{D}_t$, if $\boldsymbol{\theta}_t$ is obtained by an $(\epsilon, \delta)$-DP mechanism from the remapped data $\mathcal{R}_t^{task}(\mathcal{D}_t)$.*

*Proof sketch.* We show that the mechanism $\mathcal{M}_t$ is $(\epsilon, \delta)$-DP by analysing the cases for the remapped adjacent dataset and by using the original $(\epsilon, \delta)$-DP guarantees of the mechanism that provides the DP weights $\boldsymbol{\theta}_t$. The full proof can be found in Appendix C.2. □

The utility impact of label remapping is inherently problem-dependent. For example, in the ICD-11 (World Health Organization, 2019) medical coding, merging fine-grained diagnostic categories into broader parent classes leads to a loss in fine-grained classification performance. At the same time, such remapping may improve generalization, since higher-level labels are easier to learn and provide more training examples per class.

## 5 DP CL METHODS USING PRE-TRAINED MODELS

We adapt two families of CL methods utilising pre-trained models to DP CL. These are prototype classifiers (Janson et al., 2022; McDonnell et al., 2023) in Sec. 5.1, and expandable PEFT adapters (Zhao et al., 2024b; Zhou et al., 2024b;a) in Sec. 5.2. Pre-trained models are known to be able to boost utility in both DP and CL communities and are thus a fitting choice for studying the combination of DP and CL but have not been proposed for the combination of both.

### 5.1 COSINE SIMILARITY CLASSIFIER

We use a pre-trained model $f_{\boldsymbol{\theta}}^{\text{pre}} : \mathcal{X} \to \mathbb{R}^K$ as a frozen feature extractor without additional training during CL (Janson et al., 2022) to map inputs $\boldsymbol{x}$ to feature vectors $\boldsymbol{v} = f_{\boldsymbol{\theta}}^{\text{pre}}(\boldsymbol{x}) \in \mathbb{R}^K$. The idea is to accumulate class-specific sums of these vectors under DP, and then classify points according to their cosine similarity with the class sums (see Alg. 1).

---

**Algorithm 1** Cosine Classifier

**Require:** Number of tasks $T$, per-task DP noise level $\sigma$
1: **for** $t \in [T]$ **do**
2:      Get task dataset $\mathcal{D}_t$ and set of labels $\mathcal{O}_t$
3:      **// Compute DP sum for each $o \in \mathcal{O}_t$ with class task data $\mathcal{D}_{t,o}$ and add to cumulative sum:**
4:      **for** $o \in \mathcal{O}_t$ **do**
5:          $\boldsymbol{s}_o \leftarrow \boldsymbol{s}_o + \left(\sum_{\boldsymbol{x} \in \mathcal{D}_{t,o}} \frac{f_{\boldsymbol{\theta}}^{\text{pre}}(\boldsymbol{x})}{\|f_{\boldsymbol{\theta}}^{\text{pre}}(\boldsymbol{x}))\|_2}\right) + \mathcal{N}(0, \sigma^2 I)$
6:      **end for**
7: **end for**

**Output:** $\{\boldsymbol{s}_1, \ldots, \boldsymbol{s}_{|\cup_{t=1}^T \mathcal{O}_t|}\}$ (set of cumulative DP sums)

---

We denote $\mathcal{D}_{t,o}$ as all samples from class $o$ at task $t$. At each task $t = 1, \ldots, T$, we accumulate a per-class sum of features (normalized to bound the sensitivity of each summand) with the Gaussian mechanism (Balle & Wang, 2018). This will result in a vector $\boldsymbol{s}_{t,o} \in \mathbb{R}^K$ for each of the classes $o \in \mathcal{O}_t$, and writing $\boldsymbol{s}_{0,o} = \mathbf{0}_K$ for any $o$, implies that:

$$\boldsymbol{s}_{t,o} = \begin{cases} \boldsymbol{s}_{t-1,o} + \left(\displaystyle\sum_{\boldsymbol{x} \in \mathcal{D}_{t,o}} \frac{f_{\boldsymbol{\theta}}^{\text{pre}}(\boldsymbol{x})}{\|f_{\boldsymbol{\theta}}^{\text{pre}}(\boldsymbol{x})\|_2}\right) + \boldsymbol{z}_t & \text{if } o \in \mathcal{O}_t \\ \boldsymbol{s}_{t-1,o} & \text{if } o \notin \mathcal{O}_t \end{cases} \tag{5}$$

where $\boldsymbol{z}_t \sim \mathcal{N}(\mathbf{0}, \sigma^2 \mathbf{I})$ is Gaussian noise with scale $\sigma$ corresponding to the desired $(\epsilon, \delta)$-DP privacy budget, and $\mathcal{O}_t$ is the set of classes for task $t$. We compute the sum-of-features rather than the mean-of-features (Janson et al., 2022; Rebuffi et al., 2017) to avoid the need to release the number of examples per class under DP, which would require adding more noise.

To predict the label of a test sample $\boldsymbol{x}^*$ after training up to some time step $t \leq T$, we assign it the class label that maximizes the cosine similarity of $\boldsymbol{v}^*$ with the corresponding per-class feature sum:

$$\hat{y}^* = \arg\max_{o \in \cup_{k=1}^t \mathcal{O}_k} \text{CosineSimilarity}(\boldsymbol{v}^*, \boldsymbol{s}_{t,o}). \tag{6}$$

We only need to store the per-class sums and the pre-trained model, thus memory requirements would be $K|\mathcal{O}_t|$ (cumulative sum) + $\dim(\theta)$ (pre-trained model weights). The memory requirements scale as $O(|\cup_{t=1}^T \mathcal{O}_t|)$, growing with the size of the output label space.

## 5.2 PARAMETER-EFFICIENT FINE-TUNING (PEFT) ENSEMBLE

We construct an ensemble of prediction models by fine-tuning task-specific models $f_t : \mathcal{X} \to \mathcal{O}_t$ on the data set $\mathcal{D}_t$, $t = 1, \ldots, T$ using DP-SGD (see Alg. 2). To avoid having to store $T$ copies of the full model, we can fine-tune either only the classifier head, or, more generally, use Parameter-Efficient Fine-Tuning (PEFT, Houlsby et al. 2019) with adaptation methods such as LoRA (Hu et al., 2022). In this case, we need to store only the task-specific adapter weights and the final classification layers, as well as a single copy of the pre-trained model.

---

**Algorithm 2** PEFT Ensemble

**Require:** Number of tasks $T$, DP-SGD hypers $\xi$
1: **for** $t \in [T]$ **do**
2:     Get task dataset $\mathcal{D}_t$ and set of labels $\mathcal{O}_t$
3:     **// Initialise model $f_t$ (predicting $\mathcal{O}_t$):**
4:     $f_t \leftarrow \text{init}(f_{\boldsymbol{\theta}}^{\text{pre}})$
5:     **// Train model $f_t$ on current task data $\mathcal{D}_t$**
6:     $f_t \leftarrow \text{DP-SGD}(f_t; \mathcal{D}_t, \xi)$
7: **end for**

**Output:** $\{f_1 \ldots f_T\}$ (a set of $T$ DP models)

---

Thus the memory requirements would be $T(|\mathcal{O}_t|K + |\mathcal{O}_t|)$ (last layer) + $T \dim(\theta_{\text{PEFT}})$ (adapter weights) + $\dim(\theta)$ weights pre-trained model. The memory requirements and compute therefore scale as $O((\max_t |\mathcal{O}_t| + \dim(\theta_{\text{PEFT}}))T)$. Throughout the paper we employ parameter-efficient FiLM (Perez et al., 2018) adapters, as this approach has been found effective in prior works on transfer learning, including with DP (Shysheya et al., 2023; Tobaben et al., 2023).

Concretely, considering fine-tuning the last layer only, denote the feature vector of the pre-trained model by $\boldsymbol{v} = f_{\boldsymbol{\theta}}^{\text{pre}}(\boldsymbol{x}) \in \mathbb{R}^K$. Then the full task-specific model is $f_t(\boldsymbol{x}) = g_{\phi_t}(\boldsymbol{v})$, where $g_{\phi_t} : \mathbb{R}^K \to \mathcal{O}_t$ is the output head with trainable parameters $\phi_t$. With FiLM we additionally fine-tune a subset of the backbone normalisation layers' parameters of the pre-trained model to shift and scale the activations throughout the backbone. For example, for the model considered in the experiments the FiLM parameters are $0.04\%$ of the total number of pre-trained model parameters.

At test time, say at $t \in [T]$, we predict the label of a test sample $\boldsymbol{x}^*$ by assigning the class label with the largest logit over all the tasks and all the classes seen so far:

$$\hat{y}^* = \underset{o \in \cup_{k=1}^t \mathcal{O}_k, l \in \{1, \ldots, t\}}{\arg \max} f_l(\boldsymbol{x}^*)_o, \tag{7}$$

where $f_l(\boldsymbol{x}^*)_o$ denotes the logit corresponding to label $o$ for the $l$-th model in the ensemble. In Appendix D.2 we consider alternatives to $\arg\max$ including the aggregation rule by Zhao et al. (2024a).

## 6 EXPERIMENTS

We evaluate how our proposed solutions to update labels in CL perform using our proposed methods using pre-trained models under DP. We first focus on an idealistic CL setting where all data of a class is in one task only (Sec. 6.1) and then move on to a more realistic setting where the classes are not distributed uniformly (Sec. 6.2). In Sec. 6.3, we benchmark our methods on standard CL settings.

In all experiments, we utilise a ViT-Base-16 (ViT-B; Dosovitskiy et al., 2021) network pre-trained on the ImageNet-21K (Russakovsky et al., 2015) dataset. We assume that the pre-training data is public and that the task datasets $\mathcal{D}_t$ are sensitive and need to be protected with DP. All experiments are in the class-incremental learning setting where no task labels are available at test time (van de Ven et al., 2022). We run five repeats with different data splits, DP noise and hyperparameter optimization and plot the second worst, median and second best seed. See Appendix D for full details.

**Datasets** We experiment with the following benchmarks for CL: Split-CIFAR-100 which is CIFAR-100 (Krizhevsky, 2009) split into 10 tasks with 10 classes/task and Split-ImageNet-R (Wang et al., 2022a) which is ImageNet-R (Hendrycks et al., 2021) split into 10 tasks with 20 classes/task. We refer to the non-CL versions of the datasets as base datasets.

**Task-wise DP** We apply task-wise DP with add/remove base adjacency (see Sec. 3), by assuming that each data point appears in only one task, then the final privacy guarantees are obtained through parallel composition. Therefore, if a mechanism is $(\epsilon, \delta)$-DP, then the composition is also $(\epsilon, \delta)$-DP.

**Metrics** We report accuracy and forgetting from Chaudhry et al. (2018) for evaluation like prior CL work (Mirzadeh et al., 2021; Yoon et al., 2022). The average accuracy measures the test set accuracy across all seen tasks, where Final Acc. denotes the average accuracy of all $T$ tasks for the final model.

Forgetting is given by the difference between the highest accuracy of a task and its accuracy at the current task. See Appendix D for formal definitions.

**Label Methods** We use the following setup for the label methods proposed in Sec. 4:

- **Release Labels** ($\mathbf{S_{learned}}$ in Sec. 4.2): At every task $t$, we only update the DP CL methods with the labels that have been released at that task $t$. To determine the optimal split in privacy budget between the label release mechanism and the actual DP training, we run a grid search that allocates between $0.01$ and $1.0$ for releasing the labels and the rest for the DP training.
- **Public Labels** ($\mathbf{S_{prior}}$ in Sec. 4.3): We artificially enlarge the label space updated at every task to simulate a large but imprecise prior public label space. We experiment with two sizes of enlarged label space that are $10\times$ and $1000\times$ the number of labels of the base dataset and contain all labels of the full base dataset. We do not experiment with a label space that changes between tasks $t$, but theoretically that is possible as long as the changes are decoupled from changes in the sensitive dataset.

**Baselines** We consider two baselines ($\mathbf{S_{data}}$ in Sec. 4.1) to compare our label methods against:

- **Label Oracle:** At every task $t$, we only update labels contained in the task dataset $\mathcal{D}_t$.
- **Base Dataset:** We update all labels of the full base dataset at every task $t$.

## 6.1 COMPARISON OF METHODS USING DISJOINT SPLIT

We start with an idealistic baseline where all the data of a class is only contained in the dataset $\mathcal{D}_t$ of one task $t$. Prior work (Koh et al., 2022; Moon et al., 2023) describes this setting as disjoint split as any given class will only appear in one task $t$ and never re-appear. See tabular results in Appendix E.2.

**Comparison with Desai et al. (2021)** Fig. 3 shows a comparison with an enhanced implementation of the method of Desai et al. (2021) (see Appendix D.3 for further details) to fine-tune the same ViT-B model with PEFT. In this benchmark, all models share the same output label space at every task $t$, containing all classes from the base dataset. For each seed, we randomly permute the class order so that no class is tied to a fixed task. Our main finding is that both of our methods (Cosine Classifier and PEFT Ensemble) consistently outperform the method of Desai et al. (2021) at $\epsilon = 1$ and $\epsilon = 8$, despite not using any memory buffers (see Sec. 3).

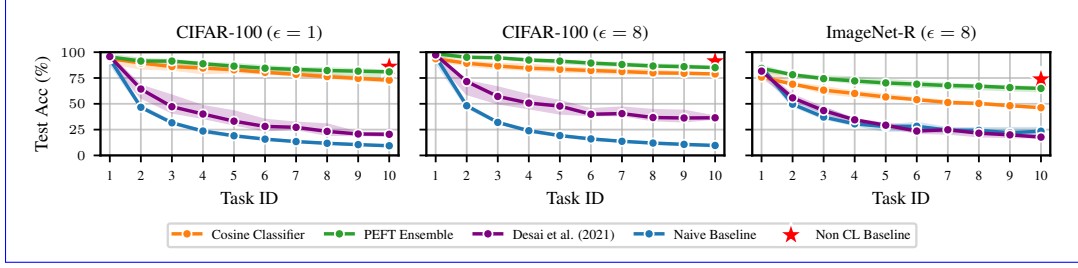

Figure 3: Comparison with Desai et al. (2021): our methods (Cosine Classifier and PEFT Ensemble) clearly outperform the enhanced implementation of Desai et al. (2021) without needing memory buffers in the Base Dataset setting. We show the median test accuracy per task on Split-CIFAR-100 and Split-ImageNet-R. The error bars are the min/max accuracies obtained over five repeats with different class ordering, and with $\delta = 10^{-5}$.

**Comparison of Label Methods** In Fig. 4, we provide a first overview of the different methods. The main finding is that in this disjoint split, DP releasing labels can be superior (for Cosine Classifier under some settings) or has similar performance than using public labels. Releasing labels performs better because then labels in a model only get updated when the class actually occurs in the task $t$ (i.e. in one task) and not in each of the 10 tasks as with the public labels. Only a very small percentage of the privacy budget ($< 10\%$) is required to release the labels, because of the large number of examples per class per task. The reduced privacy budget remaining for the actual DP training has a very small effect on the utility of the CL methods. In Sec. 6.2 we will show the impact of a less optimal split.

**Details on Release Labels** An obvious drawback of releasing the labels is determining the split in privacy budget between learning the labels and the actual DP training. In Fig. 5, we visualize the trade-off. Allocating a too small DP budget on releasing the labels (left of red dotted line) results in

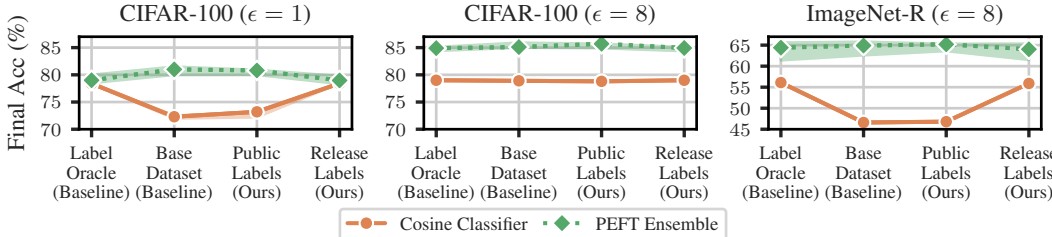

Figure 4: Comparison of Label Methods: Release Labels is superior or comparable to Public Labels (1000× size) in the disjoint split. The PEFT Ensemble that uses DP-SGD is only slightly affected by the additional noise due to spending privacy budget on learning the labels or the larger label space. Results are the median over five repeats at $\delta = 10^{-5}$ with the errorbars showing 2nd/4th best seed.

suboptimal performance as not all data is used for learning, while spending too much DP budget (right of red dotted line) increases the DP noise during training leading eventually to dropping test accuracy.

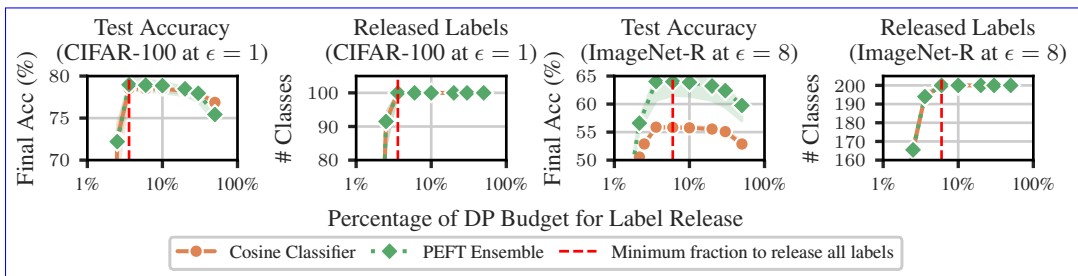

Figure 5: Details on Release Labels: The fraction of DP for label release needs to be tuned for achieving optimal test accuracy (at the red dotted line). Allocating too little or too much DP budget results in suboptimal performance as either not all labels are learned or too much noise is added during training. Lines are median over five repeats at $\delta = 10^{-5}$ with the error bars showing 2nd/4th best seed.

**Details on Public Labels** In Fig. 6, we compare different sizes of prior label space and their impact on the utility. The drop of utility with stronger privacy (Cosine Classifier with $\epsilon = 1$) originates from more frequently updating occurring labels and not from unused additional labels (see difference between the baselines and no drop between base dataset and $10 - 1000×$ more labels).

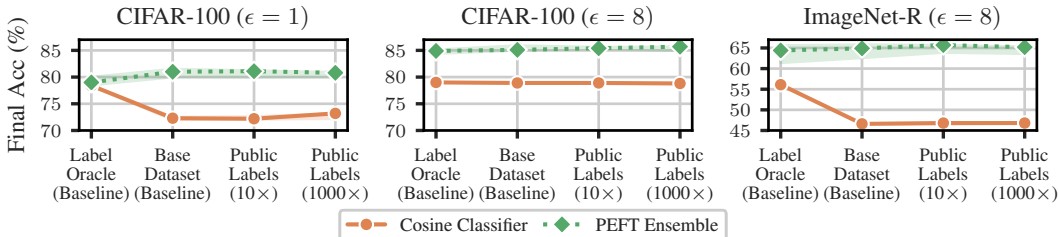

Figure 6: Details on Public Labels: Unused additional labels do not further degrade utility, but the cost from the public label method comes from more frequently updating labels (see difference between the baselines and no drop between base dataset and $10×$ more labels). Results are the median over five repeats at $\delta = 10^{-5}$ with the error bars showing 2nd/4th best seed.

## 6.2 IMPACTS ON NON-UNIFORMLY DISTRIBUTED LABELS (BLURRY TASKS)

In Sec. 6.1 each class was only contained in one task, but in many real life examples this is not the case. We consider i-Blurry (Koh et al., 2022) with 50 disjoint classes and different blurry sample ratios (see right plot of Fig. 7).
We experiment with Split-CIFAR-100 in Fig. 7. The main finding is that under tighter privacy ($\epsilon = 1$) releasing labels degrades with higher blurry sample ratio, because because a higher privacy fraction

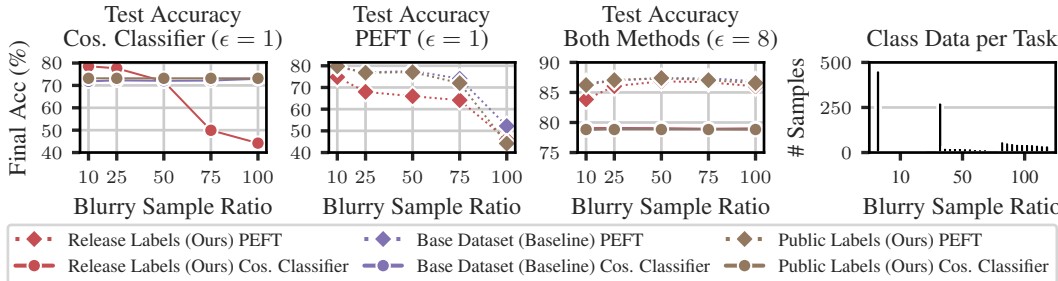

Figure 7: Impact of Blurriness: Increased blurriness degrades the utility at $\epsilon = 1$ when releasing the labels. For PEFT Ensemble this is the case with all label methods at $\epsilon = 1$. At $\epsilon = 8$ blurriness has a much smaller effect. Right panel shows the number of samples per blurry class per task. We show the median over five repeats at $\delta = 10^{-5}$ (see more in Table A7 and Fig. A3). Public Label is $1000\times$ size.

is required to release the labels leaving less privacy budget for the DP training. Under weaker privacy a large enough fraction can be allocated to release the labels without distorting the DP training. The PEFT Ensemble degrades with higher blurry sample ratio at $\epsilon = 1$ regardless of the label method as too little data is available for training models at every task, while the Cosine Classifier is invariant.

### 6.3 ADDITIONAL EXPERIMENTS

To establish our DP methods we provide experiments on established CL benchmarks in Appendix F. Comparing to two baselines, we show that the Cosine Classifier is a viable alternative when storage or compute are limited and the domain shift to the pre-training data is small. However, with larger domain shift, PEFT is the only option to achieve good privacy/utility trade-offs.

## 7 DISCUSSION AND CONCLUSION

We studied an attack on the output label space as a privacy-side channel, introduced two methods to eliminate the side-channel, and finally studied the two methods by adapting pre-trained models.

**Recommendations for Label Methods** The impact on utility depends on the privacy budget and classifier (Fig. 4). The release labels method requires to tune the DP budget split between learning labels and the actual DP training (Fig. 5) and small classes get dropped (Fig. 7). The large public label space can be used when there is such prior public information, which is not necessarily always the case. Then the size of the public label space has only a minor impact (Fig. 6). Thus we recommend to use release label method when the classes are large or no public information is known and the public label method when classes are smaller and public information is available.

**Privacy budget split heuristic for $S_{learned}$** As a simple heuristic motivated by our baselines, when $\epsilon \geq 1$ one may allocate $10\%$ of the target privacy budget to $S_{learned}$. If data-independent prior knowledge of class sizes is available (e.g., each class has at least 500 samples), one can use the distribution of the mechanism in Eq. (3) to compute the minimum budget split that releases all labels with probability close to 1. This split, however, may not yield optimal accuracy, as a slightly lower privacy budget (higher noise) in the training mechanism can sometimes improve generalization. Without such data-independent priors, the minimum split cannot be computed from sensitive class sizes, since doing so is not DP.

**Recommendations for DP CL methods** The PEFT Ensemble outperforms in non-blurry settings as it can adapt the feature extraction (Figs. 4 to 7). The Cosine Classifier is an alternative with blurry tasks (Fig. 7) and with limited compute and memory and the pre-training fits well to the sensitive dataset.

**Limitations** We do not use methods based on synthetic data but they are promising for future work. Also, we did not derive any utility bounds for our models, which can be useful for choosing the privacy budget split between DP label learning and the DP training, which we leave to future work.

## REPRODUCIBILITY STATEMENT

We provide proofs and detailed discussion of the results of Sec. 4 in Appendix C. Appendix B expands on DP CL privacy definitions and theoretical grounding for the method that we use. Algs. 1 and 2 provide pseudo-codes of our proposed DP CL methods that use pre-trained models. We provide experimental details in Appendix D and pointers to accessing the used models and datasets in Appendix G. The detailed tabular results in Appendices E.2 and F help to verify the correctness of reproduced experiments. We also provide the source code of the experiments at `https://anonymous.4open.science/r/iclr-dpcl-046B`.

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

## APPENDICES

We provide an overview of the notation in Appendix A and expand the discussion on existing DP CL definitions in Appendix B.1. We define task-wise DP and revisit the task-wise adjacency relation in Appendix B.2, then provide some composition results in Appendix B.3. In Appendix C, we provide more details and the proofs for Sec. 4. Appendix D contains experimental details about our experiments in Sec. 6, Appendix E.2 has tabular results for the figures in Sec. 6, and Appendix F contains results for checking that our methods work well in established CL settings. Appendix G lists the licenses and access possibilities for the used models and datasets in Sec. 6. Appendix H explains the usage of LLMs in our work.

## LIST OF APPENDICES

## A  NOTATION

Table A1: Notations

| | |
|---|---|
| $t$ | task index |
| $T$ | total number of tasks |
| $I$ | set of task indices |
| $\text{units}(\mathcal{D}_t)$ | a mapping that provides the privacy unit of a dataset |
| $\boldsymbol{x}_t^{(k)}$ | $k^{\text{th}}$ sample features in task $t$ |
| $y_t^{(k)}$ | $k^{\text{th}}$ sample label in task $t$ |
| $N_t$ | total number of samples in task $t$ |
| $m$ | input data dimensionality |
| $\mathcal{X}, (\mathcal{X}_t)$ | (task-specific) feature space |
| $\mathcal{O}, (\mathcal{O}_t)$ | (task-specific) general output space |
| $\mathcal{Y}, \mathcal{Y}_t, \mathcal{O}^{\text{data}}, (\mathcal{O}_t^{\text{data}})$ | true (task-specific) label space |
| $\mathcal{O}^{\text{prior}}, (\mathcal{O}_t^{\text{prior}})$ | publicly known (task-specific) label set |
| $\mathcal{O}_t^{\text{learned}}$ | (task-specific) labels learned from the dataset by a DP mechanism |
| $\mathcal{D}, (\mathcal{D}_t)$ | (task-specific) dataset (features and labels) |
| $\varepsilon, \delta$ | DP privacy parameters |
| $\mathcal{D} \simeq \mathcal{D}'$ | DP neighboring datasets |
| $\mathcal{A}$ | randomized algorithm |
| $\text{Range}(\mathcal{A})$ | set of all possible outcomes for $\mathcal{A}$ |
| $S$ | outcome event for a randomized algorithm |
| $\mathcal{M}_t$ | DP algorithm for releasing the classifier in task $t$ |
| $\mathcal{M}$ | DP algorithm for composing the classifiers for tasks $1, \dots, T$ |
| $\mathcal{R}_t^{\text{label}}$ | (task-specific) label remapping function |
| $\mathcal{R}_t^{\text{task}}$ | (task-specific) task dataset remapping function |
| $\mathcal{L}$ | label release mechanism |
| $\boldsymbol{\theta}$ | model parameters |
| $f_{\boldsymbol{\theta}}^{\text{pre}}$ | pre-trained model parameterized by $\boldsymbol{\theta}$ |
| $f_{\boldsymbol{\theta}}$ | model parameterized by $\boldsymbol{\theta}$ |
| $\mathbf{S_{data}}$ | setting where the dataset labels are directly released |
| $\mathbf{S_{prior}}$ | setting where public labels are used for each task |
| $\mathbf{S_{learned}}$ | setting where the labels are learned from the dataset |
| $s_{t,o}$ | noisy feature sum for task $t$, class $o$ |
| $K$ | feature extractor dimensionality (omitting classifier layer) |
| $\boldsymbol{v} = f_{\boldsymbol{\theta}}^{\text{pre}}(\boldsymbol{x}) \in \mathbb{R}^K$ | feature vector from pre-trained model |
| $\mathcal{D}_{t,o}$ | samples with class $o$ in task $t$ |
| $\xi$ | DP-SGD algorithm-specific parameters |
| $g_{\boldsymbol{\phi}_t} : \mathbb{R}^K \to \mathcal{O}_t$ | task-specific head parameterized by $\phi_t$ |
| $\boldsymbol{z}_t \sim \mathcal{N}(\mathbf{0}, \sigma^2 I)$ | Gaussian noise with variance $\sigma^2$ |

## B  OTHER DETAILS RELATED TO DP CL MECHANISMS

### B.1  LIMITATIONS OF CURRENT DP CL THEORY

Regarding CL, defining what is meant for a learning algorithm to be DP is not trivial, as the data is not available to the algorithm at once. Instead, the algorithm receives a datasets $\mathcal{D}_t$ gradually over time $t = 1, \ldots, T$, and releases intermediate classifiers $f_{\theta_t} : \mathcal{X} \to \mathcal{O}_t, t = 1, \ldots, T$. In this section, we discuss existing approaches and their limitations. In the next section, we introduce an alternative definition.

**DP CL with Global Adjacency**  Desai et al. (2021) introduced an adjacency relation for DP CL based on the union of the datasets from all the tasks $t = 1, \ldots, T$ and their episodic memories. For the following definition, for each task $t$, $\mathcal{D}_t$ is a dataset, $\mathfrak{M}_t$ is an episodic memory, and for any two sets $A$ and $A'$, $\|A - A'\|_1 \le 1$ means that they are adjacent such that one of the sets differs from the other by at most one sample.

**Definition B.1** (Definition 2 in Desai et al. (2021)).  Two databases $D = (\mathcal{D}, \mathfrak{M})$ and $D' = (\mathcal{D}', \mathfrak{M}')$, where $\mathcal{D} = \cup_{t=1}^T \mathcal{D}_t, \mathcal{D}' = \cup_{t=1}^T \mathcal{D}'_t, \mathfrak{M} = \cup_{t=1}^T \mathfrak{M}_t$, and $\mathfrak{M}' = \cup_{t=1}^T \mathfrak{M}'_t$, are called continual adjacent, if $\|\mathcal{D} - \mathcal{D}'\|_1 \le 1$ and $\|\mathfrak{M} - \mathfrak{M}'\|_1 \le 1$.

If we disregard the episodic memories ($\mathfrak{M} = \mathfrak{M}' = \varnothing$), the main issue with Definition B.1, is that in the general case where multiple adjacent datasets $\mathcal{D}_t, \mathcal{D}'_t$ are allowed to differ (i.e., $\|\mathcal{D}_t - \mathcal{D}'_t\| = 1$), then the global adjacency relation $\|\mathcal{D} - \mathcal{D}'\|_1 \le T$ does not imply that $\|\mathcal{D}_t - \mathcal{D}'_t\|_1 \le 1$. Moreover, even in the restricted case of $\|\mathcal{D} - \mathcal{D}'\|_1 \le 1$, encoding the adjacency at the global level loses information about which task $t$ has differing adjacent datasets $\mathcal{D}_t \ne \mathcal{D}'_t$. We can even choose $\mathcal{D}_t$ and $\mathcal{D}'_t$ such that $\mathcal{D}_t \cap \mathcal{D}'_t = \varnothing, t = 1, \ldots, T$, while still satisfying $\|\cup_t \mathcal{D}_t - \cup_t \mathcal{D}'_t\|_1 \le 1$.

The privacy accounting in Desai et al. (2021) is done for each task separately, and then they calculate the total privacy guarantees over all tasks through basic sequential composition, i.e., $\epsilon = \sum_{t=1}^T \epsilon_t$ (Desai et al., 2021, Lemmas 1 & 2). Therefore, their privacy accounting does not strictly correspond to their Definition B.1, and it could benefit from a more granular adjacency at the task level.

**DP CL for Data Streams**  Epasto et al. (2023) introduced a DP definition for data streams, where two streams $\mathcal{S} = (x_1, \ldots, x_T)$ and $\mathcal{S}' = (x'_1, \ldots, x'_T)$ are neighbouring or adjacent if there exists at most one timestamp $t^*$ for which $x_{t^*} \ne x'_{t^*}$ and $x_t = x'_t$, for all $t \ne t^*$, see also (Hassanpour et al., 2022; Dwork et al., 2010a; Lecuyer et al., 2019). However, their definition is not suitable for our setting, where we operate on task datasets $\mathcal{D}_t, t = 1, \ldots, T$. Hassanpour et al. (2022) studied DP in CL for streaming data and proposed a method that provides meaningful DP guarantees in that context. Still, their method introduces complications when applying DP-SGD on a task level. In particular, standard privacy accounting techniques, especially those relying on subsampling amplification, assume that each minibatch is sampled i.i.d. from a larger dataset. This assumption does not hold under streaming adjacency (see Choquette-Choo et al. (2024) for a discussion on amplification in streaming settings). For these reasons, the streaming adjacency is not well-suited for our proposed approaches.

**Lifelong DP**  Regarding lifelong learning with DP (Lifelong DP), Lai et al. (2022) proposed a task level adjacency relation for the $\epsilon$-Lifelong learning,

**Definition B.2** (Definition 2 in Lai et al. (2022)).  Given any two lifelong databases $\mathrm{data}_T = \{\mathcal{D}, \mathfrak{M}\}$ and $\mathrm{data}'_T = \{\mathcal{D}', \mathfrak{M}'\}$, where $\mathcal{D} = \{\mathcal{D}_1, \ldots, \mathcal{D}_T\}, \mathcal{D}' = \{\mathcal{D}'_1, \ldots, \mathcal{D}'_T\}, \mathbb{M} = \{\mathbb{M}_1, \ldots, \mathbb{M}_T\}, \mathbb{M}' = \{\mathbb{M}'_1, \ldots, \mathbb{M}'_T\}, \mathbb{M}_t = \cup_{i=1}^{t-1} \mathfrak{M}_i$, and $\mathbb{M}'_t = \cup_{i=1}^{t-1} \mathfrak{M}'_i$. $\mathrm{data}_T$ and $\mathrm{data}'_T$ are called lifelong neighboring databases if, $\forall t \in \{1, \ldots, T\}$: (i) $\mathcal{D}_t$ and $\mathcal{D}'_t$ differ by at most one tuple; and (ii) $\mathfrak{M}_t$ and $\mathfrak{M}'_t$ differ by at most one tuple.

If we disregard the episodic memories ($\mathfrak{M}_t = \mathfrak{M}'_t = \varnothing$) in Definition B.2, then this adjacency relation would be suitable for our setting. Lai et al. (2022) also introduced a definition for $\epsilon$-Lifelong DP,

**Definition B.3** (Definition 3 in Lai et al. (2022)).  $\epsilon$-Lifelong DP. Given a lifelong database $\mathrm{data}_T$, a randomized algorithm $\mathcal{A}$ achieves $\epsilon$-Lifelong DP, if for any of two lifelong neighbouring databases $(\mathrm{data}_T, \mathrm{data}'_T)$, for all possible outputs $\{\boldsymbol{\theta}_t\}_{t=1}^T \in \mathrm{Range}(\mathcal{A}), \forall T \in \mathbb{N}$ we have that

$$\Pr[(\mathrm{data}_T) = \{\boldsymbol{\theta}_t\}_{t=1}^T] \le e^\epsilon \Pr[(\mathrm{data}'_T)_{i \in [1,m]} = \{\boldsymbol{\theta}_t\}_{t=1}^T]$$
$$\nexists(\epsilon' < \epsilon, t \le T) : \Pr[(\mathrm{data}_t) = \{\boldsymbol{\theta}_i\}_{i=1}^t] \le e^{\epsilon'} \Pr[(\mathrm{data}'_t) = \{\boldsymbol{\theta}_i\}_{i=1}^t] \tag{A1}$$

where $\mathrm{Range}(\mathcal{A})$ denotes every possible output of $\mathcal{A}$.

The issue with Definition B.3 is that it corresponds to one type of composition, i.e., parallel composition. Also, it does not allow for (approximate) $(\epsilon, \delta)$-DP.

To summarise the limitations of existing approaches for defining DP CL:

1. The global adjacency relation over the union of all tasks is not suitable for continual learning, and the adjacency of the task datasets $\mathcal{D}_t$, $\mathcal{D}'_t$ cannot be inferred from the unions $\mathcal{D} = \cup_{t=1}^{T} \mathcal{D}_t$, $\mathcal{D}' = \cup_{t=1}^{T} \mathcal{D}'_t$ (Desai et al., 2021).

2. DP CL definitions and methods for data streams cannot be naturally translated to the setting we consider, i.e., tasks are defined based on datasets $\mathcal{D}_t$, $t = 1, \ldots, T$ (Epasto et al., 2023; Hassanpour et al., 2022). Particularly, the method of Hassanpour et al. (2022) does not account for subsampling amplification which introduces complications when applying DP-SGD on the level of a task.

3. $\epsilon$-Lifelong DP (Lai et al., 2022) has a suitable adjacency relation for our setting, if we discard the episodic memories; however, their definition does not allow for (approximate) $(\epsilon, \delta)$-DP and it is restricted to parallel composition over the tasks.

## B.2 Task-Wise DP for DP CL

As discussed in the previous section Appendix B.1, some of the existing works on DP CL, including Desai et al. (2021); Lai et al. (2022), introduce a definition for DP with CL along with a dataset adjacency relation; however, these works have several limitations. In Desai et al. (2021), the data adjacency relation depends on the datasets from both the current and future tasks, and in Lai et al. (2022), their privacy definition is restricted to only one type of composition and does not allow for (approximate) $(\epsilon, \delta)$-DP. To guarantee privacy in CL, our basic approach is to define task-wise adjacency, given in Definition B.4, which is a more general than what was given in Definition 3.2, and task-wise DP, given in Definition B.5. We gave a minimal definition for the task-wise adjacency relation in the main text because our experiments only apply parallel composition, and thus it was sufficient. However, the generalization can be used to apply various composition methods (McSherry, 2010; Dwork et al., 2010b; Whitehouse et al., 2023), to account for the total privacy.

**Definition B.4** (Task-wise Adjacency). Let $\simeq$ be an adjacency relation between datasets, $I \subseteq \mathbb{N}$ be a set of task indices that can be either finite or infinite, and $1 \leq n \leq |I|$. A sequence of datasets $(\mathcal{D}_t)_{t \in I}$ is said to be $n$ task-wise adjacent to another sequence of datasets $(\mathcal{D}'_t)_{t \in I}$, denoted $(\mathcal{D}_t)_{t \in I} \simeq_n (\mathcal{D}'_t)_{t \in I}$, if there exists $J \subseteq I$ such that $|J| = n$, and for any $t \in J$, we have $\mathcal{D}_t \simeq \mathcal{D}'_t$, and that $\mathcal{D}_t = \mathcal{D}'_t$ if $t \in I \setminus J$. We say that the order of task-wise adjacency is $n$.

Definition B.4 is inspired by both the definitions in Epasto et al. (2023) and Lai et al. (2022), and it differs from both by being able to control the number of possible different adjacent task datasets. If the order of adjacency is 1, i.e., for only one $t^* \in I$, we have $\mathcal{D}_{t^*} \simeq \mathcal{D}'_{t^*}$ and $\mathcal{D}_t = \mathcal{D}'_t$ if $t \neq t^*$, as in Definition 3.2. However, $t^*$ is unknown in advance, and is arbitrary.

**Definition B.5** (Task-wise DP). Let $\simeq$ be an adjacency relation between datasets, and let $(\mathcal{M}_t)_{t \in I}$ be any sequence of mechanisms, such that $I \subseteq \mathbb{N}$ is a set of task indices. Define $\mathcal{M}$ as the mechanism obtained by composing $(\mathcal{M}_t)_{t \in I}$, that is, for any sequence of datasets $(\mathcal{D}_t)_{t \in I}$,

$$\mathcal{M} : (\mathcal{D}_t)_{t \in I} \mapsto (\mathcal{M}_t(\mathcal{D}_t))_{t \in I}. \tag{A2}$$

Given $n$ such that $1 \leq n \leq |I|$, the sequence of mechanisms $(\mathcal{M}_t)_{t \in I}$ is said to satisfy task-wise $(\epsilon, \delta, \simeq_n)$-DP, if for any two $n$ task-wise adjacent sequences of datasets $(\mathcal{D}_t)_{t \in I} \simeq_n (\mathcal{D}'_t)_{t \in I}$, and for any $S \subseteq \mathrm{Range}(\mathcal{M})$:

$$\Pr\left[\mathcal{M}\left((\mathcal{D}_t)_{t \in I}\right) \in S\right] \leq \exp(\epsilon) \times \Pr\left[\mathcal{M}\left((\mathcal{D}'_t)_{t \in I}\right) \in S\right] + \delta. \tag{A3}$$

For the composed mechanism $\mathcal{M}$ defined in Eq. (A2),

$$\mathrm{Range}(\mathcal{M}) = \prod_{t \in I} \mathrm{Range}(\mathcal{M}_t). \tag{A4}$$

We define the $t$th composition projection function as

$$\pi_t : \prod_{k \in I} \mathrm{Range}(\mathcal{M}_k) \to \mathrm{Range}(\mathcal{M}_t), \tag{A5}$$

such that $\pi_t\left((s_k)_{k\in I}\right) = s_t$, where $(s_k)_{k\in I}$ is any output of $\mathcal{M}$. We want to be able to translate task-wise $(\epsilon, \delta, \simeq_1)$-DP guarantees to typical $(\epsilon, \delta, \simeq)$-DP guarantees. To do that, first we need to introduce the following definition.

**Definition B.6.** Any function $\text{assign} : \mathcal{X} \times \mathcal{Y} \to I$ is called a task assignment function. That is, it splits any $\mathcal{D}$ into the following sets:

$$\mathcal{D}_t = \{(\boldsymbol{x}, y) \in \mathcal{D} : \text{assign}\left((\boldsymbol{x}, y)\right) = t\}, \tag{A6}$$

providing a means for transforming $\mathcal{D}$ into a sequence of datasets $(\mathcal{D}_t)_{t\in I}$. We will denote

$$\text{assign}\left[\mathcal{D}\right] = (\mathcal{D}_t)_{t\in I}, \tag{A7}$$

using square brackets, to clearly distinguish it from function evaluation.

The following theorem enables us to translate the task-wise $(\epsilon, \delta, \simeq_1)$-DP guarantees to typical $(\epsilon, \delta, \simeq)$-DP guarantees. Unless explicitly stated otherwise, the order of task-wise adjacency is assumed to be 1 when writing task-wise $(\epsilon, \delta)$-DP.

**Theorem B.7.** *The sequence of mechanisms $(\mathcal{M}_t)_{t\in I}$ is task-wise $(\epsilon, \delta, \simeq_1)$-DP iff $\mathcal{M} \circ \text{assign}[\cdot]$ is $(\epsilon, \delta, \simeq)$-DP, where $\mathcal{M}$ is defined in Eq. (A2), for any task assignment function $\text{assign} : \mathcal{X} \times \mathcal{Y} \to I$.*

*Proof.* (**necessary condition** $\Rightarrow$) Assume that $(\mathcal{M}_t)_{t\in I}$ is task-wise $(\epsilon, \delta, \simeq_1)$-DP. Let $\mathcal{D} \simeq \mathcal{D}'$ be any two adjacent datasets, and let $\text{assign} : \mathcal{X} \times \mathcal{Y} \to I$ be any task assignment function. Assume without loss of generality that $\mathcal{D}' = \mathcal{D} \cup \{(\boldsymbol{x}^*, y^*)\}$. From the task assignment function, we obtain two sequences of datasets $\text{assign}[\mathcal{D}] = (\mathcal{D}_t)_{t\in I}$ and $\text{assign}[\mathcal{D}'] = (\mathcal{D}'_t)_{t\in I}$. We will argue that $(\mathcal{D}_t)_{t\in I} \simeq_1 (\mathcal{D}'_t)_{t\in I}$.

Define $t^* = \text{assign}((\boldsymbol{x}^*, y^*))$, then for all $t \neq t^*$, $\mathcal{D}_t = \mathcal{D}'_t$ because $\mathcal{D}' \setminus \{(\boldsymbol{x}^*, y^*)\} = \mathcal{D}$. However, for $t = t^*$, we have $\mathcal{D}'_{t^*} = \mathcal{D}_{t^*} \cup \{(\boldsymbol{x}^*, y^*)\}$, which implies that $\mathcal{D}'_{t^*} \simeq \mathcal{D}_{t^*}$, and hence $(\mathcal{D}_t)_{t\in I} \simeq_1 (\mathcal{D}'_t)_{t\in I}$. Consider the following, for all $S \in \text{Range}(\mathcal{M})$,

$$\begin{aligned}
\Pr\left[\mathcal{M} \circ \text{assign}\left[\mathcal{D}\right]\right] &= \Pr\left[\mathcal{M}\left((\mathcal{D}_t)_{t\in I}\right) \in S\right] \\
&\leq \exp(\epsilon)\Pr\left[\mathcal{M}\left((\mathcal{D}'_t)_{t\in I}\right) \in S\right] + \delta \\
&= \exp(\epsilon)\Pr\left[\mathcal{M} \circ \text{assign}\left[\mathcal{D}'\right] \in S\right] + \delta.
\end{aligned} \tag{A8}$$

Therefore, $\mathcal{M} \circ \text{assign}[\cdot]$ is $(\epsilon, \delta)$-DP.

(**sufficient condition** $\Leftarrow$) Conversely, assume that $\mathcal{M} \circ \text{assign}[\cdot]$ is $(\epsilon, \delta, \simeq)$-DP for any task assignment function $\text{assign} : \mathcal{X} \times \mathcal{Y} \to I$. Let $(\mathcal{D}_t)_{t\in I} \simeq_1 (\mathcal{D}'_t)_{t\in I}$ be any two adjacent sequences of datasets. Define $\mathcal{D} = \cup_{t\in I}\mathcal{D}_t$ and let $\mathcal{D}' = \cup_{t\in I}\mathcal{D}'_t$. Since for some $t^*$, $\mathcal{D}_{t^*} \simeq \mathcal{D}'_{t^*}$, and for $t \neq t^*$, $\mathcal{D}_t = \mathcal{D}'_t$, then assume without loss of generality that $\mathcal{D}'_{t^*} = \mathcal{D}_{t^*} \cup \{(\boldsymbol{x}^*, y^*)\}$. Thus, $\mathcal{D}' = \mathcal{D} \cup \{(\boldsymbol{x}^*, y^*)\}$, and hence $\mathcal{D}' \simeq \mathcal{D}$.

To complete the proof, we need to craft a task assignment function $\text{assign}$ that produces the same sequences of adjacent sets $(\mathcal{D}_t)_{t\in I} \simeq_1 (\mathcal{D}'_t)_{t\in I}$ from $\mathcal{D} \simeq \mathcal{D}'$. Define $\text{assign}$ as follows

$$\text{assign}\left((\boldsymbol{x}, y)\right) = \begin{cases} t & \text{if } (\boldsymbol{x}, y) \in \mathcal{D}'_t \text{ for some } t \in I, \\ 1 & \text{otherwise.} \end{cases} \tag{A9}$$

Therefore, $\text{assign}[\mathcal{D}] = (\mathcal{D}_t)_{t\in I}$ and $\text{assign}[\mathcal{D}'] = (\mathcal{D}'_t)_{t\in I}$. Hence, by the same argument in Eq. (A8), we obtain that $(\mathcal{M}_t)_{t\in I}$ is task-wise $(\epsilon, \delta, \simeq_1)$-DP. $\square$

**Corollary B.8.** *If a sequence of mechanisms $(\mathcal{M}_t)_{t\in I}$ is task-wise $(\epsilon, \delta, \simeq_1)$-DP, then for each $t \in I$, $\mathcal{M}_t$ is $(\epsilon, \delta)$-DP.*

*Proof.* Let $t^* \in I$ be chosen arbitrarily, and define the following task assignment function:

$$\text{assign}\left((\boldsymbol{x}, y)\right) = t^* \tag{A10}$$

for all $(\boldsymbol{x}, y) \in \mathcal{X} \times \mathcal{Y}$, i.e., it is just the constant function. Let $\mathcal{D} \simeq \mathcal{D}'$ be any two adjacent datasets, then

$$\text{assign}\left[\mathcal{D}\right] = (\mathcal{D}_t)_{t\in I}, \tag{A11}$$

such that $\mathcal{D}_{t^*} = \mathcal{D}$ and $\mathcal{D}_t = \varnothing$, for all $t \neq t^*$. Similarly,

$$\text{assign}\left[\mathcal{D}'\right] = (\mathcal{D}'_t)_{t \in I}, \tag{A12}$$

such that $\mathcal{D}'_{t^*} = \mathcal{D}'$ and $\mathcal{D}'_t = \varnothing$, for all $t \neq t^*$. By Theorem B.7, $\mathcal{M} \circ \text{assign}[\cdot]$ is $(\epsilon, \delta)$-DP, and thus by post processing $\pi_{t^*} \circ \mathcal{M} \circ \text{assign}[\cdot]$ is also $(\epsilon, \delta)$-DP.

Moreover, $\pi_{t^*} \circ \mathcal{M} \circ \text{assign}[\mathcal{D}] = \mathcal{M}_{t^*}(\mathcal{D})$ and $\pi_{t^*} \circ \mathcal{M} \circ \text{assign}[\mathcal{D}'] = \mathcal{M}_{t^*}(\mathcal{D}')$, completing the proof. □

### B.3 TASK-WISE DP AND COMPOSITION (FOR SEC. B.2)

Practically, one can apply Theorem B.7 and Corollary B.8, and then apply any composition method. Parallel composition (McSherry, 2010), which we use in our experiments, applies when each individual's data appears in only one dataset $\mathcal{D}_t$, and a DP mechanism is invoked on each dataset separately. It is $(\epsilon, \delta)$-DP if each mechanism is $(\epsilon, \delta)$-DP. Although the proof already exists in (McSherry, 2010), we also show that parallel composition is task-wise $(\epsilon, \delta)$-DP in Theorem B.10 for completeness. In contrast, sequential composition degrades privacy guarantees with each use. We show that basic sequential composition of a sequence of mechanisms is task-wise $(|I|\epsilon, |I|\delta, \simeq_T)$-DP in Theorem B.11, and the proof is similar to the one in Dwork & Roth (2014). Adaptive sequential composition (Dwork & Roth, 2014; Dwork et al., 2010b) applies when privacy parameters (including the number of tasks) are known in advance; otherwise, fully adaptive composition is needed (Whitehouse et al., 2023).

A necessary condition for $\mathcal{M}$ in Eq. (A2) to be DP is that it should not leak that any of the task datasets is empty, i.e. that $\mathcal{D}_t = \varnothing$, almost surely, for some $t$. Let $\perp$ denote that a classifier was not released. We also say that $\perp$ is not a (proper) classifier. If $\mathcal{M}_t(\mathcal{D}_t) = \perp$ (almost surely) when $\mathcal{D}_t = \varnothing$ and $\mathcal{M}_t(\mathcal{D}_t) \neq \perp$ (almost surely) when $\mathcal{D}_t \neq \varnothing$, then $\mathcal{M}$ is not DP according to Proposition B.9. The reason is that this can potentially leak whether the classifier was trained including or excluding a certain sample. Therefore, we will assume that the classifier release mechanism always outputs a proper classifier $\mathcal{M}_t(\mathcal{D}_t) \neq \perp$ (almost surely) for any $t$. In practice, for the case of using pretrained models, when $\mathcal{D}_t = \varnothing$, we can initialize both the fine-tuning weights and the classifier's weights randomly before releasing the classfier.

**Proposition B.9.** *The mechanism $\mathcal{M}_t$ is not $(\epsilon, \delta)$-DP for $0 \leq \delta < 1$ when $\mathcal{M}_t(\mathcal{D}_t) = \perp$ (almost surely) if and only if $\mathcal{D}_t = \varnothing$.*

Denote the range of $\mathcal{M}_t$ as $\mathcal{H}_t \cup \{\perp\}$, i.e. the union of the hypothesis space $\mathcal{H}_t$ (proper-classifiers) and $\{\perp\}$ (non-proper classifier). For the following proof, note that $\perp \notin \mathcal{H}_t$ for any $t$.

*Proof.* We argue by constructing a counterexample. Assume that $\mathcal{M}_t$ is $(\epsilon, \delta)$-DP for $0 \leq \delta < 1$ and that $\mathcal{M}_t(\mathcal{D}_t) = \perp$ (almost surely) iff $\mathcal{D}_t = \varnothing$. Let $\mathcal{D}_t = \varnothing$ and $\mathcal{D}'_t = \{(\boldsymbol{x}^*, y^*)\}$ where $(\boldsymbol{x}^*, y^*) \in \mathcal{X} \times \mathcal{O}_t$ is arbitrary. Since $\mathcal{M}_t$ is $(\epsilon, \delta)$-DP, then for all $S \subseteq \mathcal{H}_t$,

$$\Pr[\mathcal{M}(\mathcal{D}'_t) \in S] \leq e^\epsilon \Pr[\mathcal{M}(\mathcal{D}_t) \in S] + \delta. \tag{A13}$$

Let $S = \mathcal{H}_t$, then

$$\Pr\left[\mathcal{M}(\mathcal{D}'_t) \in S\right] = 1. \tag{A14}$$

However, since $\mathcal{D}_t = \varnothing$, then $\mathcal{M}_t(\mathcal{D}_t) = \perp \notin \mathcal{H}_t$. Thus,

$$\Pr[\mathcal{M}(\mathcal{D}_t) \in S] = 0. \tag{A15}$$

Therefore, Eq. (A13) implies that

$$1 \leq e^\epsilon \times 0 + \delta \implies 1 \leq \delta, \tag{A16}$$

but we assumed that $0 < \delta < 1$, a contradiction. □

**Theorem B.10** (Parallel composition). *Let $(\mathcal{M}_t)_{t \in I}$ be a sequence of $(\epsilon, \delta)$-DP mechanisms, then their parallel composition is task-wise $(\epsilon, \delta, \simeq_1)$-DP.*

*Proof.* According to McSherry (2010), for any sequence of mechanisms $(\mathcal{M}_t)_{t \in I}$, their parallel composition is given by

$$(\mathcal{M}_t(\cdot \cap \mathcal{V}_t))_{t \in I}, \tag{A17}$$

where $\mathcal{V}_t \subseteq \mathcal{X} \times \mathcal{Y}$ are disjoint subsets, such that $\pi_{\mathcal{X}}(\mathcal{V}_i) \cap \pi_{\mathcal{X}}(\mathcal{V}_j) = \varnothing$. That is, no two sets of $(\mathcal{V}_t)_{t \in I}$ share the same individual's data.

Let $(\mathcal{D}_t)_{t \in I} \simeq_1 (\mathcal{D}'_t)_{t \in I}$, then there exists $t^*$ such that $\mathcal{D}_{t^*} \simeq \mathcal{D}'_{t^*}$ and $\mathcal{D}_t = \mathcal{D}'_t$ for all $t \neq t^*$. Define the following:

$$\mathcal{M}_{\mathrm{par}}((\mathcal{D}_t)_{t \in I}) = (\mathcal{M}_t(\mathcal{D}_t \cap \mathcal{V}_t))_{t \in I}. \tag{A18}$$

Observe that for each $t \neq t^*$, $\mathcal{D}_t \cap \mathcal{V}_t = \mathcal{D}'_t \cap \mathcal{V}_t$, and that $\mathcal{D}_{t^*} \cap \mathcal{V}_{t^*} \simeq \mathcal{D}'_{t^*} \cap \mathcal{V}_{t^*}$. From the fact that $\mathcal{M}_{t^*}$ is $(\epsilon, \delta)$-DP, for all $S_{t^*} \subseteq \mathrm{Range}(\mathcal{M}_{t^*})$, it holds that

$$\Pr\left[\mathcal{M}_{t^*}(\mathcal{D}_{t^*} \cap \mathcal{V}_{t^*}) \in S_{t^*}\right] \leq \exp(\epsilon) \times \Pr\left[\mathcal{M}_{t^*}(\mathcal{D}'_{t^*} \cap \mathcal{V}_{t^*}) \in S_{t^*}\right] + \delta \tag{A19}$$

Let $S \subseteq \mathrm{Range}(\mathcal{M})$, denote $S_t = \pi_t(S)$, for any $t \in I$. Also, denote

$$S[s_{t^*}] = \left\{ (s_t)_{t \in I \setminus \{t^*\}} : \left((s_t)_{t \in I \setminus \{t^*\}}; s_{t^*}\right)_{t^*} \in S \right\}, \tag{A20}$$

where '$(;)_t$' means insertion at $t$, e.g. $((a, b); c)_2 = (a, c, b)$. For brevity denote, for any $t$ and any $\mathcal{D}$,

$$p_t(s_t; \mathcal{D}) = \Pr\left[\mathcal{M}_t(\mathcal{D} \cap \mathcal{V}_t) = s_t\right], \tag{A21}$$

From the law of total probability, and since the mechanisms are applied independently, then

$$
\begin{aligned}
&\Pr\left[\mathcal{M}_{\mathrm{par}}((\mathcal{D}_t)_{t \in I}) \in S\right] \\
&= \mathbb{E}_{(s_t)_{t \in I}}\left[\mathbf{1}_S\left(\mathcal{M}_{\mathrm{par}}((\mathcal{D}_t)_{t \in I})\right)\right] \\
&= \mathbb{E}_{s_{t^*}}\left[\mathbb{E}_{(s_t)_{t \in I \setminus \{t^*\}}}\left[\mathbf{1}_S\left(\mathcal{M}_{\mathrm{par}}((\mathcal{D}_t)_{t \in I})\right) \big| \mathcal{M}_{t^*}(\mathcal{D}_{t^*} \cap \mathcal{V}_{t^*}) = s_{t^*}\right]\right] \\
&= \mathbb{E}_{s_{t^*}}\left[\mathbb{E}_{(s_t)_{t \in I \setminus \{t^*\}}}\left[\mathbf{1}_{S[s_{t^*}]}\left(\mathcal{M}_{\mathrm{par}}\left((\mathcal{D}_t)_{t \in I \setminus \{t^*\}}\right)\right)\right]\right] \\
&= \int_{s_{t^*} \in S_{t^*}} \int_{(s_t)_{t \in I \setminus \{t^*\}} \in S[s_{t^*}]} \Pr\left[\mathcal{M}_{\mathrm{par}}\left((\mathcal{D}_t)_{t \in I \setminus \{t^*\}}\right) = (s_t)_{t \in I \setminus \{t^*\}}\right] p_{t^*}(s_{t^*}; \mathcal{D}'_{t^*}) \\
&= \int_{s_{t^*} \in S_{t^*}} \Pr\left[\mathcal{M}_{\mathrm{par}}\left((\mathcal{D}_t)_{t \in I \setminus \{t^*\}}\right) \in S[s_{t^*}]\right] p_{t^*}(s_{t^*}; \mathcal{D}'_{t^*}),
\end{aligned} \tag{A22}
$$

where $\mathbf{1}_S[\cdot]$ denotes the indicator function. Now observe that for any $s_{t^*}$,

$$\Pr\left[\mathcal{M}_{\mathrm{par}}\left((\mathcal{D}_t)_{t \in I \setminus \{t^*\}}\right) \in S[s_{t^*}]\right] = \min\left(1, \Pr\left[\mathcal{M}_{\mathrm{par}}\left((\mathcal{D}_t)_{t \in I \setminus \{t^*\}}\right) \in S[s_{t^*}]\right]\right). \tag{A23}$$

Since $\mathcal{D}_t = \mathcal{D}'_t$ for all $t \neq t^*$, then

$$
\begin{aligned}
&\Pr\left[\mathcal{M}_{\mathrm{par}}((\mathcal{D}_t)_{t \in I}) \in S\right] \\
&= \int_{s_{t^*} \in S_{t^*}} \min\left(1, \Pr\left[\mathcal{M}_{\mathrm{par}}\left((\mathcal{D}_t)_{t \in I \setminus \{t^*\}}\right) \in S[s_{t^*}]\right]\right) p_{t^*}(s_{t^*}; \mathcal{D}_{t^*}) \\
&= \int_{s_{t^*} \in S_{t^*}} \min\left(1, \Pr\left[\mathcal{M}_{\mathrm{par}}\left((\mathcal{D}'_t)_{t \in I \setminus \{t^*\}}\right) \in S[s_{t^*}]\right]\right) p_{t^*}(s_{t^*}; \mathcal{D}_{t^*}) \\
&= \int_{s_{t^*} \in S_{t^*}} \min\left(1, \Pr\left[\mathcal{M}_{\mathrm{par}}\left((\mathcal{D}'_t)_{t \in I \setminus \{t^*\}}\right) \in S[s_{t^*}]\right]\right) \\
&\qquad\qquad \times \left(p_{t^*}(s_{t^*}; \mathcal{D}_{t^*}) - \exp(\epsilon) p_{t^*}(s_{t^*}; \mathcal{D}'_{t^*}) + \exp(\epsilon) p_{t^*}(s_{t^*}; \mathcal{D}'_{t^*})\right) \\
&= \int_{s_{t^*} \in S_{t^*}} \min\left(1, \Pr\left[\mathcal{M}_{\mathrm{par}}\left((\mathcal{D}'_t)_{t \in I \setminus \{t^*\}}\right) \in S[s_{t^*}]\right]\right) \times \exp(\epsilon) p_{t^*}(s_{t^*}; \mathcal{D}'_{t^*}) \\
&\quad + \int_{s_{t^*} \in S_{t^*}} \min\left(1, \Pr\left[\mathcal{M}_{\mathrm{par}}\left((\mathcal{D}'_t)_{t \in I \setminus \{t^*\}}\right) \in S[s_{t^*}]\right]\right) \\
&\qquad\qquad \times \left(p_{t^*}(s_{t^*}; \mathcal{D}_{t^*}) - \exp(\epsilon) p_{t^*}(s_{t^*}; \mathcal{D}'_{t^*})\right)
\end{aligned} \tag{A24}
$$

Now since for any $a, b$, we have $\min(a,b) \le a$ and $\min(a,b) \le b$, then

$$\Pr\left[\mathcal{M}_{\text{par}}\left((\mathcal{D}_t)_{t\in I}\right) \in S\right]$$

$$\le \int_{s_{t^*}\in S_{t^*}} \Pr\left[\mathcal{M}_{\text{par}}\left((\mathcal{D}'_t)_{t\in I\setminus\{t^*\}}\right) \in S[s_{t^*}]\right] \times \exp(\epsilon)p_{t^*}(s_{t^*}; \mathcal{D}'_{t^*})$$

$$+ \int_{s_{t^*}\in S_{t^*}} \left(p_{t^*}(s_{t^*}; \mathcal{D}_{t^*}) - \exp(\epsilon)p_{t^*}(s_{t^*}; \mathcal{D}'_{t^*})\right) \qquad \text{(A25)}$$

$$= \exp(\epsilon)\Pr\left[\mathcal{M}_{\text{par}}\left((\mathcal{D}'_t)_{t\in I}\right) \in S\right] + \int_{s_{t^*}\in S_{t^*}} \left(p_{t^*}(s_{t^*}; \mathcal{D}_{t^*}) - \exp(\epsilon)p_{t^*}(s_{t^*}; \mathcal{D}'_{t^*})\right)$$

$$\le \exp(\epsilon)\Pr\left[\mathcal{M}_{\text{par}}\left((\mathcal{D}'_t)_{t\in I}\right) \in S\right] + \delta,$$

from the fact that $\mathcal{M}_{t^*}$ is $(\epsilon, \delta)$-DP. Therefore, $\mathcal{M}_{\text{par}}$ is $(\epsilon, \delta, \simeq_1)$-DP which ends the proof. $\square$

While in the proof of Theorem B.10, we followed the typical setting in where each mechanism is applied to a disjoint dataset, the proof still holds if the datasets where not disjoint. The standard parallel composition theorem for $(\epsilon, \delta)$-DP requires partitioning the domain into disjoint sets $(\mathcal{V}_t)_{t\in I}$, so that for any dataset $\mathcal{D}$, adding or removing a point will only affect one of adjacent datasets of $(\mathcal{D}\cap\mathcal{V}_t)_{t\in I}$. However, the adjacency relation in task-wise DP ($\simeq_1$), already imposes this on the adjacent sequences $(\mathcal{D}_t)_{t\in I}$ and $(\mathcal{D}'_t)_{t\in I}$, making the partitioning of the domain redundant.

**Theorem B.11.** *Let $I$ be a set of task indices, and let $T = |I|$. If $(\mathcal{M}_t)_{t\in I}$ is a sequence of $(\epsilon, \delta)$-DP mechanisms, then their basic sequential composition is $(T\epsilon, T\delta, \simeq_T)$-DP.*

*Proof.* Let $(\mathcal{D}_t)_{t\in I} \simeq_T (\mathcal{D}'_t)_{t\in I}$, and let $S \subseteq \text{Range}(\mathcal{M})$. We will prove the result by induction on $T$. The case $T = 1$ holds trivially. Now assume that the statement of the theorem holds for $|I| = T - 1$, we will prove that it also holds for $|I^*| = T$ where $I^* = I \cup \{t^*\}$. Denote $S_t = \pi_t(S)$, for any $t \in I$. Similar to the steps in Eq. (A22) in the proof of the previous theorem (Theorem B.10),

$$\Pr\left[\mathcal{M}\left((\mathcal{D}_t)_{t\in I^*}\right) \in S\right] = \int_{s_{t^*}\in S_{t^*}} \Pr\left[\mathcal{M}\left((\mathcal{D}_t)_{t\in I}\right) \in S[s_{t^*}]\right] \Pr\left[\mathcal{M}_{t^*}\left(\mathcal{D}_{t^*}\right) = s_{t^*}\right]. \quad \text{(A26)}$$

From the induction hypothesis, and since $\min(a, b+c) \le \min(a,b) + c$ for all $c > 0$,

$$\Pr\left[\mathcal{M}\left((\mathcal{D}_t)_{t\in I}\right) \in S[s_{t^*}]\right] = \min\left(1, \Pr\left[\mathcal{M}\left((\mathcal{D}_t)_{t\in I}\right) \in S[s_{t^*}]\right]\right)$$

$$\le \min\left(1, \exp(|I|\epsilon)\Pr\left[\mathcal{M}\left((\mathcal{D}'_t)_{t\in I}\right) \in S[s_{t^*}]\right] + |I|\delta\right) \quad \text{(A27)}$$

$$\le \min\left(1, \exp(|I|\epsilon)\Pr\left[\mathcal{M}\left((\mathcal{D}'_t)_{t\in I}\right) \in S[s_{t^*}]\right]\right) + |I|\delta.$$

From Eq. (A26) and Eq. (A27), we obtain

$$\Pr\left[\mathcal{M}\left((\mathcal{D}_t)_{t\in I^*}\right) \in S\right]$$

$$\le \int_{s_{t^*}\in S_{t^*}} \left(\min\left(1, \exp(|I|\epsilon)\Pr\left[\mathcal{M}\left((\mathcal{D}'_t)_{t\in I}\right) \in S[s_{t^*}]\right]\right) + |I|\delta\right) \Pr\left[\mathcal{M}_{t^*}\left(\mathcal{D}_{t^*}\right) = s_{t^*}\right]$$

$$= \int_{s_{t^*}\in S_{t^*}} \min\left(1, \exp(|I|\epsilon)\Pr\left[\mathcal{M}\left((\mathcal{D}'_t)_{t\in I}\right) \in S[s_{t^*}]\right]\right) \Pr\left[\mathcal{M}_{t^*}\left(\mathcal{D}_{t^*}\right) = s_{t^*}\right]$$

$$+ |I|\delta\Pr\left[\mathcal{M}_{t^*}\left(\mathcal{D}_{t^*}\right) = s_{t^*}\right]$$

$$= |I|\delta + \int_{s_{t^*}\in S_{t^*}} \min\left(1, \exp(|I|\epsilon)\Pr\left[\mathcal{M}\left((\mathcal{D}'_t)_{t\in I}\right) \in S[s_{t^*}]\right]\right) \Pr\left[\mathcal{M}_{t^*}\left(\mathcal{D}_{t^*}\right) = s_{t^*}\right]$$

$$\le |I|\delta + \int_{s_{t^*}\in S_{t^*}} \min\left(1, \exp(|I|\epsilon)\Pr\left[\mathcal{M}\left((\mathcal{D}'_t)_{t\in I}\right) \in S[s_{t^*}]\right]\right)$$

$$\times \left(\Pr\left[\mathcal{M}_{t^*}\left(\mathcal{D}_{t^*}\right) = s_{t^*}\right] - \exp(\epsilon)\Pr\left[\mathcal{M}_{t^*}\left(\mathcal{D}'_{t^*}\right) = s_{t^*}\right] + \exp(\epsilon)\Pr\left[\mathcal{M}_{t^*}\left(\mathcal{D}'_{t^*}\right) = s_{t^*}\right]\right). \quad \text{(A28)}$$

Now by applying $\min(a,b) \le a$ and $\min(a,b) \le b$, we obtain

$$\Pr\left[\mathcal{M}\left((\mathcal{D}_t)_{t\in I^*}\right) \in S\right] \le |I|\delta + \exp\left((|I|+1)\epsilon\right) \Pr\left[\mathcal{M}\left((\mathcal{D}'_t)_{t\in I^*}\right) \in S\right]$$

$$+ \int_{s_{t^*}\in S_{t^*}} \left(\Pr\left[\mathcal{M}_{t^*}\left(\mathcal{D}_{t^*}\right) = s_{t^*}\right] - \exp(\epsilon)\Pr\left[\mathcal{M}_{t^*}\left(\mathcal{D}'_{t^*}\right) = s_{t^*}\right]\right), \quad \text{(A29)}$$

and the last integral is less than $\delta$ from the fact that $\mathcal{M}_{t^*}$ is $(\epsilon, \delta)$-DP, and hence,

$$\Pr\left[\mathcal{M}\left((\mathcal{D}_t)_{t \in I^*}\right) \in S\right] \leq (|I| + 1)\delta + \exp\left((|I| + 1)\epsilon\right) \Pr\left[\mathcal{M}\left((\mathcal{D}'_t)_{t \in I^*}\right) \in S\right], \quad \text{(A30)}$$

as required. $\qquad \square$

In other words, we can say that for a sequence of $(\epsilon, \delta)$-DP mechanisms $(\mathcal{M}_t)_{t \in I}$, parallel composition corresponds to task-wise $(\epsilon, \delta, \simeq_1)$-DP, and basic sequential composition corresponds to task-wise $(|I|\epsilon, |I|\delta, \simeq_T)$-DP.

## C  OUTPUT LABEL SPACE THEORY DETAILS (FOR SEC. 4)

### C.1  PROOF OF PROPOSITION 4.1

For the following proof, let $\pi_{\mathcal{Y}} : \mathcal{X} \times \mathcal{Y} \to \mathcal{Y}$ be the label projection function where $\mathcal{Y}$ is the space of all possible labels, i.e. $\pi_{\mathcal{Y}}(\boldsymbol{x}, y) = y$ for any $(\boldsymbol{x}, y) \in \mathcal{D}_t$.

*Proof.* We argue by constructing a counterexample. Assume that $\mathcal{M}_t$ is $(\epsilon, \delta)$-DP for $0 \le \delta < 1$. Let $\mathcal{D}_t$ and $\mathcal{D}'_t$ be any two datasets that differ in only one example such that $\mathcal{D}'_t = \mathcal{D}_t \cup \{(\boldsymbol{x}^*, y^*)\}$. If either $\mathcal{O}_t = \mathcal{O}_t^{\text{data}}$ or $\mathcal{O}_t = \bigcup_{i=1}^{t} \mathcal{O}_i^{\text{data}}$, then

$$\pi_{\mathcal{Y}}(\mathcal{D}_t) \subseteq \mathcal{O}_t. \tag{A31}$$

Let $\pi_2(\cdot)$ be the mapping that takes an ordered pair as an input and outputs the second item, i.e. $\pi_2(A, B) = B$. Denote $\mathcal{O}'_t = \pi_2(\mathcal{M}_t(\mathcal{D}'_t))$. For the counter example, since $\mathcal{O}'_t$ also contains $\pi_{\mathcal{Y}}(\mathcal{D}'_t)$, then assume that $y^* \in \mathcal{O}'_t$, and assume that $y^* \notin \mathcal{O}_t$. Note that $\pi_2(\mathcal{M}_t(\mathcal{D}_t))$ is a post-processing from $\mathcal{M}_t(\mathcal{D}_t)$, and thus it is also $(\epsilon, \delta)$-DP. Hence, for any $S \subseteq \mathcal{O}_t^{\text{data}}$,

$$\Pr[\pi_2(\mathcal{M}_t(\mathcal{D}'_t)) \in S] \le e^\epsilon \Pr[\pi_2(\mathcal{M}_t(\mathcal{D}_t)) \in S] + \delta. \tag{A32}$$

Equivalently,

$$\Pr[\mathcal{O}'_t \in S] \le e^\epsilon \Pr[\mathcal{O}_t \in S] + \delta. \tag{A33}$$

Let $S = \{\mathcal{O}'_t\}$. Since $y^* \notin \mathcal{O}_t$, then $\mathcal{O}'_t \ne \mathcal{O}_t$. This implies that,

$$\Pr[\mathcal{O}'_t \in S] = 1, \tag{A34}$$

and that

$$\Pr[\mathcal{O}_t \in S] = 0. \tag{A35}$$

From Eqs. (A33) to (A35), we obtain

$$1 \le e^\epsilon \times 0 + \delta \implies 1 \le \delta, \tag{A36}$$

but we assumed that $0 < \delta < 1$, a contradiction.  $\square$

Proposition 4.1 can be generalized to any function

$$\mathcal{U} : \mathcal{X} \times \mathcal{Y} \to \mathcal{Y},$$

such that there exists two neighboring datasets $\mathcal{D}_t$ and $\mathcal{D}'_t$ that differ in only one example and $\mathcal{U}(\mathcal{D}_t) \ne \mathcal{U}(\mathcal{D}'_t)$. Thus, setting $\mathcal{O}_t = \mathcal{U}(\bigcup_{k=1}^{t} \mathcal{D}_k)$ or $\mathcal{O}_t = \mathcal{U}(\mathcal{D}_t)$, a counterexample can be constructed similarly as in the proof of Proposition 4.1.

### C.2  PROOF OF PROPOSITION 4.2

For the following proof, for any adjacent dataset $\mathcal{D}'_t = \mathcal{D}_t \bigcup \{(\boldsymbol{x}^*, y^*)\}$, we assume that there is an associated function $\mathcal{R}_t^{\text{label}'} : \mathcal{O}_t^{\text{data}'} \to \mathcal{O}_t^{\text{prior}} \bigcup \{\text{drop}\}$ that differs from $\mathcal{R}_t^{\text{label}}$ only on the new point $\{(\boldsymbol{x}^*, y^*)\}$, such that similar to Eq. (4) we obtain $\mathcal{R}_t^{\text{task}'}(\mathcal{D}'_t)$:

$$\mathcal{R}_t^{\text{task}'}(\mathcal{D}'_t) = \{(\boldsymbol{x}, \mathcal{R}_t^{\text{label}'}(y)) : (\boldsymbol{x}, y) \in \mathcal{D}'_t \text{ and } \mathcal{R}_t^{\text{label}'}(y) \ne \text{drop}\}. \tag{A37}$$

*Proof.* Define $\mathcal{W}$ to be $(\epsilon, \delta)$-DP mechanism that provides the weights $\boldsymbol{\theta}_t$. In other words,

$$\boldsymbol{\theta}_t := \mathcal{W}(\mathcal{R}_t^{\text{task}}(\mathcal{D}_t); \mathcal{O}_t^{\text{prior}}). \tag{A38}$$

To show that using $\mathcal{R}_t^{\text{task}}(\mathcal{D}_t)$ instead of $\mathcal{D}_t$ does not change the privacy guarantees for $\mathcal{W}$, let $\mathcal{D}_t$ be any set and $\mathcal{D}'_t \simeq \mathcal{D}_t$. Assume without the loss of generality that $\mathcal{D}'_t = \mathcal{D}_t \bigcup \{(\boldsymbol{x}^*, y^*)\}$, i.e. that $\mathcal{D}'_t$ has an additional point. There are two possibilities for $\mathcal{R}_t^{\text{task}'}(\mathcal{D}'_t)$:

1. $\mathcal{R}_t^{\text{task}'}(\mathcal{D}'_t) = \mathcal{R}_t^{\text{task}}(\mathcal{D}_t)$ when $\mathcal{R}_t^{\text{label}'}(y^*) = \text{drop}$.
2. $\mathcal{R}_t^{\text{task}'}(\mathcal{D}'_t) = \mathcal{R}_t^{\text{task}}(\mathcal{D}_t) \bigcup \{(\boldsymbol{x}, \mathcal{R}_t^{\text{label}'}(y^*))\}$ when $\mathcal{R}_t^{\text{label}'}(y^*) \in \mathcal{O}_t^{\text{prior}}$.

First, if $\mathcal{R}_t^{\text{task}'}(\mathcal{D}_t') = \mathcal{R}_t^{\text{task}}(\mathcal{D}_t)$, then it is trivial that for all $S \subseteq \text{Range}(\mathcal{W})$:

$$\Pr\left[\mathcal{W}(\mathcal{R}_t^{\text{task}}(\mathcal{D}_t); \mathcal{O}_t^{\text{prior}}) \in S\right] \leq \exp(\epsilon) \times \Pr\left[\mathcal{W}(\mathcal{R}_t^{\text{task}'}(\mathcal{D}_t'); \mathcal{O}_t^{\text{prior}}) \in S\right] + \delta. \tag{A39}$$

Second, if $\mathcal{R}_t^{\text{task}'}(\mathcal{D}_t') = \mathcal{R}_t^{\text{task}}(\mathcal{D}_t) \bigcup\{(\boldsymbol{x}^*, \mathcal{R}_t^{\text{label}}(y^*))\}$, then since $\mathcal{W}$ is an $(\epsilon, \delta)$-DP mechanism, setting $\mathcal{D} = \mathcal{R}_t^{\text{task}}(\mathcal{D}_t)$ and $\mathcal{D}' = \mathcal{R}_t^{\text{task}'}(\mathcal{D}_t')$ in Definition 3.1, we obtain

$$\Pr\left[\mathcal{W}(\mathcal{R}_t^{\text{task}}(\mathcal{D}_t); \mathcal{O}_t^{\text{prior}}) \in S\right] \leq \exp(\epsilon) \times \Pr\left[\mathcal{W}(\mathcal{R}_t^{\text{task}'}(\mathcal{D}_t'); \mathcal{O}_t^{\text{prior}}) \in S\right] + \delta. \tag{A40}$$

Therefore, in both cases, using $\mathcal{R}_t^{\text{task}}(\mathcal{D}_t)$ does not change the privacy guarantees of $\mathcal{W}$. Finally, since $\mathcal{W}$ is $(\epsilon, \delta)$-DP, then the mechanism

$$\mathcal{M}_t(\mathcal{R}_t^{\text{task}}(\mathcal{D}_t); \mathcal{O}_t^{\text{prior}}) \mapsto \left(\mathcal{W}(\mathcal{R}_t^{\text{task}}(\mathcal{D}_t); \mathcal{O}_t^{\text{prior}}), \mathcal{O}_t^{\text{prior}}\right) \tag{A41}$$

is a post-processing from $\mathcal{W}$ using only additional public information ($\mathcal{O}_t^{\text{prior}}$), and thus is also $(\epsilon, \delta)$-DP. $\square$

### C.3 DP Methods Protecting the Output Label Space

When protecting the set of labels by any DP mechanism, we want to quantify how likely it is to drop a label. To accomplish this, we compute the smallest possible lower bound on the probability of dropping a label, depending on the size of the class. This bound does not depend on any particular mechanism.

Suppose $y$ appears in $k$ examples of $\mathcal{D}_t$. Dropping $y$ entails dropping all the associated examples from the dataset. From group DP, if a mechanism $\mathcal{L}$ is $(\epsilon, \delta)$-DP when adjacent datasets $\mathcal{D}_t$ and $\mathcal{D}_t'$ differ by at most one example, then it is $(k\epsilon, \delta_k)$-DP, when the adjacent datasets differ by at most $k$ examples, where

$$\delta_k = \sum_{i=0}^{k-1} \exp(i\epsilon)\delta = \frac{\exp(k\epsilon) - 1}{\exp(\epsilon) - 1}\delta. \tag{A42}$$

Let $\mathcal{D}_t'$ be the dataset without the class $y$, then $\Pr[y \in \mathcal{L}(\mathcal{D}_t)] \leq \exp(k\epsilon)\Pr[y \in \mathcal{L}(\mathcal{D}_t')] + \delta_k$. However, $\Pr[y \in \mathcal{L}(\mathcal{D}_t')] = 0$, since $y$ is not in $\mathcal{D}_t'$. Hence, $\Pr[y \in \mathcal{L}(\mathcal{D}_t)] \leq \delta_k$. Therefore, $\Pr[y \notin \mathcal{L}(\mathcal{D}_t)] \geq 1 - \delta_k$. In other words, the probability of dropping the class is at least $1 - \delta_k$ as depicted in Fig. A1 for different values of $k$ (class size), $\epsilon$, and $\delta$.

To motivate our generalized method for protecting the labels, we partition a dataset $\mathcal{D}$ into subsets $\mathcal{D}_y = \{(x, y) : (x, y) \in \mathcal{D}\}$ according to the label $y$, and consider the following binary mechanism based on the Laplace mechanism $\mathcal{B}_{\text{Lap}}(\mathcal{D}_y) := |\mathcal{D}_y| + \text{Lap}\left(\frac{1}{\epsilon}\right) > \tau$, where $|\cdot|$ denotes the size of a set, Lap is the Laplace distribution, and $\tau$ is a threshold. The output of $\mathcal{B}_{\text{Lap}}$ is either true or false. It is known that the Laplace mechanism is $\epsilon$-DP (Dwork & Roth, 2014) and $\mathcal{B}_{\text{Lap}}$ is just a post-processing of it. We can use the binary mechanism $\mathcal{B}_{\text{Lap}}$ to test whether we can add any $y$ from $\mathcal{O}_t^{\text{data}}$ to $\mathcal{O}_t^{\text{learned}}$. This mechanism is $(\epsilon, \delta)$-DP with a proper $\delta$. We generalize this to any binary mechanism $\mathcal{B}$. First, we argue that there is no data-dependent $\epsilon$-DP mechanism that can protect the labels.

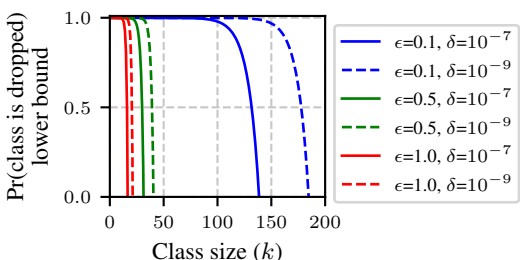

Figure A1: Lower bound of the probability that a new label is not added to the output label space $\mathcal{O}_t$. Even with $\epsilon = 1.0$ and $\delta = 10^{-7}$, classes having fewer than 13 samples are discarded with at least 99% probability, and thus cannot be learned.

**Proposition C.1.** *Given a dataset $\mathcal{D} = \{(\boldsymbol{x}_i, y_i)\}_{i=1}^N$, denote $\mathcal{O}^{data}(\mathcal{D}) = \{y : (x, y) \in \mathcal{D}\}$, then there is no $\epsilon$-DP mechanism $\mathcal{L}$ that can protect the label space such that:*

$$\Pr\left[\mathcal{L}(\mathcal{D}) \subseteq \mathcal{O}^{data}(\mathcal{D})\right] = 1 \text{ and } \Pr\left[\mathcal{L}(\mathcal{D}) \neq \varnothing\right] = 1. \tag{A43}$$

*Proof.* We prove this by constructing a counter example, assuming the negation of the theorem. Let $\mathcal{D}' = \mathcal{D} \cup \{(\boldsymbol{x}^*, y^*)\}$ such that $y^* \notin \mathcal{O}^{\text{data}}(\mathcal{D})$. Therefore, $\mathcal{O}^{\text{data}}(\mathcal{D}') = \mathcal{O}^{\text{data}}(\mathcal{D}) \bigcup \{y^*\}$. From the statement of the theorem:

$$\Pr\left[\mathcal{L}(\mathcal{D}) \subseteq \mathcal{O}^{\text{data}}(\mathcal{D})\right] = 1, \tag{A44}$$

which implies that,

$$\Pr\left[y^* \in \mathcal{L}(\mathcal{D})\right] = 0. \tag{A45}$$

Let $\mathcal{S}^* = \{\mathcal{O} \subseteq \mathcal{O}^{\text{data}}(\mathcal{D}') : y^* \in \mathcal{O}\}$, then from Equation (A45),

$$\Pr\left[\mathcal{L}(\mathcal{D}) = \mathcal{O}\right] = 0, \tag{A46}$$

for all $\mathcal{O} \in \mathcal{S}^*$. However, we require in the counterexample that for all $\mathcal{O} \in \mathcal{S}^*$.

$$\Pr\left[\mathcal{L}(\mathcal{D}') = \mathcal{O}\right] > 0. \tag{A47}$$

By applying the definition of $\epsilon$-DP,

$$\Pr\left[\mathcal{L}(\mathcal{D}') \in \mathcal{S}^*\right] \leq \exp(\epsilon)\Pr\left[\mathcal{L}(\mathcal{D}) \in \mathcal{S}^*\right]$$
$$\Longleftrightarrow \Pr\left[\mathcal{L}(\mathcal{D}') \in \mathcal{S}^*\right] \leq \exp(\epsilon) \times 0 \tag{A48}$$
$$\Longleftrightarrow \Pr\left[\mathcal{L}(\mathcal{D}') \in \mathcal{S}^*\right] \leq 0,$$

which implies that $0 < \Pr\left[\mathcal{L}(\mathcal{D}') = \mathcal{O}\right] \leq 0$, a contradiction. $\qquad\square$

According to Proposition C.1, we cannot have an $\epsilon$-DP mechanism protecting the set of labels without knowing a superset of all possible labels, which defeats the purpose of searching for such mechanism in the first place. However, we can instead have a binary $\epsilon$-DP mechanism that outputs either $\text{true}$ or $\text{false}$, split the dataset into disjoint sets $\mathcal{D}_y = \{(\boldsymbol{x}, y) : (\boldsymbol{x}, y) \in \mathcal{D}\}$ for all $y \in \mathcal{O}^{\text{data}}(\mathcal{D})$, apply the mechanism to each disjoint dataset to decide whether to include the label or not, and finally apply composition to get the privacy guarantees. The final composition is still not $\epsilon$-DP according to Proposition C.1 but we can make it $(\epsilon, \delta)$-DP with an appropriate choice of $\delta$.

**Definition C.2.** Let $\mathcal{B}$ be any binary $\epsilon$-DP mechanism, i.e., outputs either $\text{true}$ or $\text{false}$ given any set. Let $\mathcal{D} = \{(\boldsymbol{x}_i, y_i)\}_{i=1}^N$ be any dataset, and denote $\mathcal{D}_y = \{(\boldsymbol{x}, y) : (\boldsymbol{x}, y) \in \mathcal{D}\}$. We define the associated label release mechanism $\mathcal{L}_{\mathcal{B}}$, as the following mechanism:

$$\mathcal{L}_{\mathcal{B}}(\mathcal{D}) := \left\{y \in \mathcal{O}^{\text{data}} : \mathcal{B}(\mathcal{D}_y) = \text{true}\right\}. \tag{A49}$$

Now we introduce the tight bound on $\delta$ for the label release mechanism in the following theorem.

**Theorem C.3.** *The mechanism $\mathcal{L}_{\mathcal{B}}$ in Definition C.2 is $(\epsilon, \delta)$-DP such that*

$$\delta \geq \max_{\boldsymbol{x}, y} \max \left( \frac{1}{1 - \Pr\left[\mathcal{B}(\{(\boldsymbol{x}, y)\}) = \text{true}\right]} - \exp(\epsilon), \Pr\left[\mathcal{B}(\{(\boldsymbol{x}, y)\}) = \text{true}\right] \right). \tag{A50}$$

To prove Theorem C.3, we first need the following lemmas.

**Lemma C.4.** *Let $\mathcal{O} \subseteq \mathcal{O}^{\text{data}}$, then the probability of having $\mathcal{O}$ as an output for $\mathcal{L}_{\mathcal{B}}$ is given by*

$$\Pr\left[\mathcal{L}_{\mathcal{B}}(\mathcal{D}) = \mathcal{O}\right] = \prod_{y \in \mathcal{O}} \Pr\left[\mathcal{B}(\mathcal{D}_y) = \text{true}\right] \times \prod_{y \in \mathcal{O}^{\text{data}} \setminus \mathcal{O}} \Pr\left[\mathcal{B}(\mathcal{D}_y) = \text{false}\right] \tag{A51}$$

*Proof.* By applying basic definitions and rules of probability,

$$\Pr\left[\mathcal{L}_{\mathcal{B}}(\mathcal{D}) = \mathcal{O}\right] = \Pr\left[\bigwedge_{y \in \mathcal{O}} (\mathcal{B}(\mathcal{D}_y) = \text{true}) \wedge \bigwedge_{y \in \mathcal{O}^{\text{data}} \setminus \mathcal{O}} (\mathcal{B}(\mathcal{D}_y) = \text{false})\right], \tag{A52}$$

where $\wedge$ and $\bigwedge$ denote logical conjunction. Hence, the result follows from independence. We will sometimes denote $\mathcal{O}^{\text{data}}(\mathcal{D}) \setminus \mathcal{O}$ as $\mathcal{O}^c$ when $\mathcal{D}$ is known from the context. $\qquad\square$

**Lemma C.5.** *For any two neighboring datasets $\mathcal{D} \sim \mathcal{D}'$ (by the add/remove adjacency relation), and for any $y \in \mathcal{O}^{\text{data}}(\mathcal{D}) \cap \mathcal{O}^{\text{data}}(\mathcal{D}')$, we have*

$$\Pr\left[\mathcal{B}(\mathcal{D}_y) = \text{true}\right] \leq \exp(\epsilon)\Pr\left[\mathcal{B}(\mathcal{D}'_y) = \text{true}\right], \tag{A53}$$

*and*

$$\Pr\left[\mathcal{B}(\mathcal{D}_y) = \text{false}\right] \leq \exp(\epsilon)\Pr\left[\mathcal{B}(\mathcal{D}'_y) = \text{false}\right]. \tag{A54}$$

*Proof.* The proof follows from the fact that $\mathcal{B}$ is $\epsilon$-DP. $\square$

The following is the proof of Theorem C.3.

*Proof.* Assume without loss of generality, that $\mathcal{D}' = \mathcal{D} \bigcup \{(\boldsymbol{x}^*, y^*)\}$ such that $\boldsymbol{x}^* \notin \pi_{\mathcal{X}}(\mathcal{D})$. First, note that an output of the mechanism $\mathcal{L}_{\mathcal{B}}$ is a subset of $\mathcal{O}^{\text{data}}$, so we need to consider any $\mathcal{S} \subseteq \mathcal{P}\left(\mathcal{O}^{\text{data}}\right)$ where $\mathcal{P}(\cdot)$ denotes the power set. We need to consider two cases for $y^*$:

1. First, when $y^* \in \mathcal{O}^{\text{data}}(\mathcal{D})$, and thus $\mathcal{O}^{\text{data}}(\mathcal{D}') = \mathcal{O}^{\text{data}}(\mathcal{D})$. Consider the following,

$$
\Pr[\mathcal{L}_{\mathcal{B}}(\mathcal{D}) \in \mathcal{S}] = \sum_{\mathcal{O} \in \mathcal{S}} \Pr[\mathcal{L}_{\mathcal{B}}(\mathcal{D}) = \mathcal{O}]
$$

$$
= \sum_{\mathcal{O} \in \mathcal{S}} \left( \prod_{y \in \mathcal{O}} \Pr[\mathcal{B}(\mathcal{D}_y) = \text{true}] \times \prod_{y \in \mathcal{O}^{\text{data}}(\mathcal{D}) \setminus \mathcal{O}} \Pr[\mathcal{B}(\mathcal{D}_y) = \text{false}] \right),
$$
(A55)

by applying Lemma C.4. Since $\mathcal{O}^{\text{data}}(\mathcal{D}') = \mathcal{O}^{\text{data}}(\mathcal{D})$, then for each $\mathcal{O} \in \mathcal{S}$, if $y^* \notin \mathcal{O}$, then

$$
\Pr[\mathcal{B}(\mathcal{D}_y) = \text{true}] = \Pr\left[\mathcal{B}(\mathcal{D}'_y) = \text{true}\right],
$$
(A56)

for all $y \in \mathcal{O}$. Similarly, if $y^* \notin \mathcal{O}^{\text{data}}(\mathcal{D}') \setminus \mathcal{O}$, then

$$
\Pr[\mathcal{B}(\mathcal{D}_y) = \text{false}] = \Pr\left[\mathcal{B}(\mathcal{D}'_y) = \text{false}\right],
$$
(A57)

for all $y \in \mathcal{O}^{\text{data}}(\mathcal{D}') \setminus \mathcal{O}$. However, if $y^* \in \mathcal{O}$ then, by applying Lemma C.5,

$$
\Pr[\mathcal{B}(\mathcal{D}_{y^*}) = \text{true}] \leq \exp(\epsilon)\Pr\left[\mathcal{B}(\mathcal{D}'_{y^*}) = \text{true}\right],
$$
(A58)

which implies that

$$
\prod_{y \in \mathcal{O}} \Pr[\mathcal{B}(\mathcal{D}_y) = \text{true}] = \prod_{y \in \mathcal{O} \setminus \{y^*\}} \Pr[\mathcal{B}(\mathcal{D}_y) = \text{true}] \times \Pr[\mathcal{B}(\mathcal{D}_{y^*}) = \text{true}]
$$

$$
\leq \prod_{y \in \mathcal{O} \setminus \{y^*\}} \Pr\left[\mathcal{B}(\mathcal{D}'_y) = \text{true}\right] \times \exp(\epsilon)\Pr\left[\mathcal{B}(\mathcal{D}'_{y^*}) = \text{true}\right]
$$

$$
\leq \exp(\epsilon) \prod_{y \in \mathcal{O} \setminus \{y^*\}} \Pr\left[\mathcal{B}(\mathcal{D}'_y) = \text{true}\right] \times \Pr\left[\mathcal{B}(\mathcal{D}'_{y^*}) = \text{true}\right]
$$

$$
= \exp(\epsilon) \prod_{y \in \mathcal{O}} \Pr\left[\mathcal{B}(\mathcal{D}'_y) = \text{true}\right].
$$
(A59)

On the other hand, if $y^* \in \mathcal{O}^{\text{data}}(\mathcal{D}) \setminus \mathcal{O}$, then by also applying Lemma C.5,

$$
\Pr[\mathcal{B}(\mathcal{D}_{y^*}) = \text{false}] \leq \exp(\epsilon)\Pr\left[\mathcal{B}(\mathcal{D}'_{y^*}) = \text{false}\right],
$$
(A60)

and by following similar steps as Equation (A59) combined with the fact that $\mathcal{O}^{\text{data}}(\mathcal{D}) = \mathcal{O}^{\text{data}}(\mathcal{D}')$, we obtain

$$
\prod_{y \in \mathcal{O}^{\text{data}}(\mathcal{D}) \setminus \mathcal{O}} \Pr[\mathcal{B}(\mathcal{D}_y) = \text{false}] \leq \exp(\epsilon) \prod_{y \in \mathcal{O}^{\text{data}}(\mathcal{D}') \setminus \mathcal{O}} \Pr\left[\mathcal{B}(\mathcal{D}'_y) = \text{false}\right].
$$
(A61)

Combining Equation (A55), Equation (A59), and Equation (A61), and using Lemma C.4 again, we obtain

$$
\Pr[\mathcal{L}_{\mathcal{B}}(\mathcal{D}) \in \mathcal{S}] \leq \sum_{\mathcal{O} \in \mathcal{S}} \exp(\epsilon)\Pr[\mathcal{L}_{\mathcal{B}}(\mathcal{D}') = \mathcal{O}]
$$

$$
= \exp(\epsilon) \sum_{\mathcal{O} \in \mathcal{S}} \Pr[\mathcal{L}_{\mathcal{B}}(\mathcal{D}') = \mathcal{O}]
$$
(A62)

$$
= \exp(\epsilon)\Pr[\mathcal{L}_{\mathcal{B}}(\mathcal{D}') \in \mathcal{S}]
$$

$$
\leq \exp(\epsilon)\Pr[\mathcal{L}_{\mathcal{B}}(\mathcal{D}') \in \mathcal{S}] + \delta,
$$

for any $\delta > 0$, including the one in the statement of the theorem. A similar argument can also be applied to prove that

$$\Pr\left[\mathcal{L}_{\mathcal{B}}\left(\mathcal{D}'\right) \in \mathcal{S}\right] \leq \exp(\epsilon)\Pr\left[\mathcal{L}_{\mathcal{B}}\left(\mathcal{D}\right) \in \mathcal{S}\right] + \delta. \tag{A63}$$

2. Second, when $y^* \notin \mathcal{O}^{\text{data}}\left(\mathcal{D}\right)$, then $\mathcal{O}^{\text{data}}\left(\mathcal{D}'\right) = \mathcal{O}^{\text{data}}\left(\mathcal{D}\right) \cup \{y^*\}$. For any $\mathcal{O} \in \mathcal{S} \subseteq \mathcal{P}\left(\mathcal{O}^{\text{data}}(\mathcal{D})\right)$, it is obvious that $y^* \notin \mathcal{O}$ and $y^* \notin \mathcal{O}^{\text{data}}(\mathcal{D}) \setminus \mathcal{O}$. Since the range of $\mathcal{L}_{\mathcal{B}}$ is data-dependent, we must consider which of the datasets $(\mathcal{D}, \mathcal{D}')$ we are using. If we take any $\mathcal{S} \subseteq \mathcal{P}\left(\mathcal{O}^{\text{data}}(\mathcal{D})\right)$, then by Lemma C.4:

$$\Pr\left[\mathcal{L}_{\mathcal{B}}\left(\mathcal{D}\right) \in \mathcal{S}\right] = \sum_{\mathcal{O} \in \mathcal{S}} \Pr\left[\mathcal{L}_{\mathcal{B}}\left(\mathcal{D}\right) = \mathcal{O}\right]$$

$$= \sum_{\mathcal{O} \in \mathcal{S}} \left( \prod_{y \in \mathcal{O}} \Pr\left[\mathcal{B}\left(\mathcal{D}_y\right) = \text{true}\right] \times \prod_{y \in \mathcal{O}^{\text{data}}(\mathcal{D}) \setminus \mathcal{O}} \Pr\left[\mathcal{B}\left(\mathcal{D}_y\right) = \text{false}\right] \right)$$

$$= \sum_{\mathcal{O} \in \mathcal{S}} \left( \prod_{y \in \mathcal{O}} \Pr\left[\mathcal{B}\left(\mathcal{D}'_y\right) = \text{true}\right] \times \prod_{y \in \mathcal{O}^{\text{data}}(\mathcal{D}) \setminus \mathcal{O}} \Pr\left[\mathcal{B}\left(\mathcal{D}'_y\right) = \text{false}\right] \right)$$

$$= \sum_{\mathcal{O} \in \mathcal{S}} \left( \prod_{y \in \mathcal{O}} \Pr\left[\mathcal{B}\left(\mathcal{D}'_y\right) = \text{true}\right] \times \prod_{y \in \mathcal{O}^{\text{data}}(\mathcal{D}) \setminus \mathcal{O}} \Pr\left[\mathcal{B}\left(\mathcal{D}'_y\right) = \text{false}\right] \right)$$
$$\times \frac{\Pr\left[\mathcal{B}\left(\mathcal{D}'_{y^*}\right) = \text{false}\right]}{\Pr\left[\mathcal{B}\left(\mathcal{D}'_{y^*}\right) = \text{false}\right]}$$

$$= \sum_{\mathcal{O} \in \mathcal{S}} \left( \prod_{y \in \mathcal{O}} \Pr\left[\mathcal{B}\left(\mathcal{D}'_y\right) = \text{true}\right] \times \prod_{y \in \mathcal{O}^{\text{data}}(\mathcal{D}) \setminus \mathcal{O}} \Pr\left[\mathcal{B}\left(\mathcal{D}'_y\right) = \text{false}\right] \right.$$
$$\left. \times \Pr\left[\mathcal{B}\left(\mathcal{D}'_{y^*}\right) = \text{false}\right] \right) \times \frac{1}{\Pr\left[\mathcal{B}\left(\mathcal{D}'_{y^*}\right) = \text{false}\right]}$$

$$= \sum_{\mathcal{O} \in \mathcal{S}} \left( \prod_{y \in \mathcal{O}} \Pr\left[\mathcal{B}\left(\mathcal{D}'_y\right) = \text{true}\right] \times \prod_{y \in \mathcal{O}^{\text{data}}(\mathcal{D}') \setminus \mathcal{O}} \Pr\left[\mathcal{B}\left(\mathcal{D}'_y\right) = \text{false}\right] \right)$$
$$\times \frac{1}{\Pr\left[\mathcal{B}\left(\mathcal{D}'_{y^*}\right) = \text{false}\right]}$$

$$= \frac{\Pr\left[\mathcal{L}_{\mathcal{B}}\left(\mathcal{D}'\right) \in \mathcal{S}\right]}{\Pr\left[\mathcal{B}\left(\mathcal{D}'_{y^*}\right) = \text{false}\right]}. \tag{A64}$$

To prove that $\mathcal{L}_{\mathcal{B}}$ is $(\epsilon, \delta)$-DP, we have two cases:

(a) To obtain that

$$\Pr\left[\mathcal{L}_{\mathcal{B}}\left(\mathcal{D}\right) \in \mathcal{S}\right] \leq \exp(\epsilon)\Pr\left[\mathcal{L}_{\mathcal{B}}\left(\mathcal{D}'\right) \in \mathcal{S}\right] + \delta, \tag{A65}$$

we need

$$\Pr\left[\mathcal{L}_{\mathcal{B}}\left(\mathcal{D}\right) \in \mathcal{S}\right] - \exp(\epsilon)\Pr\left[\mathcal{L}_{\mathcal{B}}\left(\mathcal{D}'\right) \in \mathcal{S}\right] \leq \delta, \tag{A66}$$

equivalently, by using Equation (A64),

$$\Pr\left[\mathcal{L}_{\mathcal{B}}\left(\mathcal{D}\right) \in \mathcal{S}\right] - \exp(\epsilon)\Pr\left[\mathcal{L}_{\mathcal{B}}\left(\mathcal{D}\right) \in \mathcal{S}\right]\Pr\left[\mathcal{B}\left(\mathcal{D}'_{y^*}\right) = \text{false}\right] \leq \delta. \tag{A67}$$

Therefore,

$$\Pr\left[\mathcal{L}_{\mathcal{B}}\left(\mathcal{D}\right) \in \mathcal{S}\right]\left(1 - \exp(\epsilon)\Pr\left[\mathcal{B}\left(\mathcal{D}'_{y^*}\right) = \text{false}\right]\right) \leq \delta. \tag{A68}$$

However, since $y^*$ appears only once in $\mathcal{D}'$, then

$$\Pr\left[\mathcal{L}_{\mathcal{B}}\left(\mathcal{D}\right) \in \mathcal{S}\right]\left(1 - \exp(\epsilon)\Pr\left[\mathcal{B}\left(\{(\boldsymbol{x}^*, y^*)\}\right) = \text{false}\right]\right) \leq \delta \tag{A69}$$

Since this has to hold for all $\mathcal{S}$, we need to select $\delta$ such that:

$$\delta \geq \max_{\mathcal{S} \subseteq \mathcal{P}(\mathcal{O}^{\text{data}}(\mathcal{D}))} \Pr\left[\mathcal{L}_{\mathcal{B}}\left(\mathcal{D}\right) \in \mathcal{S}\right]\left(1 - \exp(\epsilon)\Pr\left[\mathcal{B}\left(\{(\boldsymbol{x}^*, y^*)\}\right) = \text{false}\right]\right), \quad \text{(A70)}$$

and the maximum occurs when $\mathcal{S} = \mathcal{P}\left(\mathcal{O}^{\text{data}}\left(\mathcal{D}\right)\right)$ for which $\Pr\left[\mathcal{L}_{\mathcal{B}}\left(\mathcal{D}\right) \in \mathcal{S}\right] = 1$, and thus,

$$\delta \geq 1 - \exp(\epsilon)\Pr\left[\mathcal{B}\left(\{(\boldsymbol{x}^*, y^*)\}\right) = \text{false}\right], \quad \text{(A71)}$$

and one can work backwards to obtain the $(\epsilon, \delta)$-DP guarantee. Again repeating similar steps but substituting for $\Pr\left[\mathcal{L}_{\mathcal{B}}\left(\mathcal{D}\right) \in \mathcal{S}\right]$, yields

$$\frac{\Pr\left[\mathcal{L}_{\mathcal{B}}\left(\mathcal{D}'\right) \in \mathcal{S}\right]}{\Pr\left[\mathcal{B}\left(\mathcal{D}'_{y^*}\right) = \text{false}\right]} - \exp(\epsilon)\Pr\left[\mathcal{L}_{\mathcal{B}}\left(\mathcal{D}'\right) \in \mathcal{S}\right] \leq \delta, \quad \text{(A72)}$$

and hence

$$\delta \geq \frac{1}{\Pr\left[\mathcal{B}\left(\{(\boldsymbol{x}^*, y^*)\}\right) = \text{false}\right]} - \exp(\epsilon). \quad \text{(A73)}$$

However, Equation (A73) is trivial since $\exp(\epsilon) > 1$ for any $\epsilon > 0$ and $\frac{1}{\Pr[\cdot]} \geq 1$.

(b) To obtain that

$$\Pr\left[\mathcal{L}_{\mathcal{B}}\left(\mathcal{D}'\right) \in \mathcal{S}\right] \leq \exp(\epsilon)\Pr\left[\mathcal{L}_{\mathcal{B}}\left(\mathcal{D}\right) \in \mathcal{S}\right] + \delta, \quad \text{(A74)}$$

we need

$$\Pr\left[\mathcal{L}_{\mathcal{B}}\left(\mathcal{D}'\right) \in \mathcal{S}\right] - \exp(\epsilon)\Pr\left[\mathcal{L}_{\mathcal{B}}\left(\mathcal{D}\right) \in \mathcal{S}\right] \leq \delta, \quad \text{(A75)}$$

equivalently, by using Equation (A64),

$$\Pr\left[\mathcal{L}_{\mathcal{B}}\left(\mathcal{D}'\right) \in \mathcal{S}\right] - \exp(\epsilon)\frac{\Pr\left[\mathcal{L}_{\mathcal{B}}\left(\mathcal{D}'\right) \in \mathcal{S}\right]}{\Pr\left[\mathcal{B}\left(\mathcal{D}'_{y^*}\right) = \text{false}\right]} \leq \delta. \quad \text{(A76)}$$

Therefore,

$$\Pr\left[\mathcal{L}_{\mathcal{B}}\left(\mathcal{D}'\right) \in \mathcal{S}\right]\left(1 - \frac{\exp(\epsilon)}{\Pr\left[\mathcal{B}\left(\mathcal{D}'_{y^*}\right) = \text{false}\right]}\right) \leq \delta. \quad \text{(A77)}$$

Similarly to the previous point,

$$\Pr\left[\mathcal{L}_{\mathcal{B}}\left(\mathcal{D}'\right) \in \mathcal{S}\right]\left(1 - \frac{\exp(\epsilon)}{\Pr\left[\mathcal{B}\left(\{(\boldsymbol{x}^*, y^*)\}\right) = \text{false}\right]}\right) \leq \delta, \quad \text{(A78)}$$

which implies that

$$\delta \geq 1 - \frac{\exp(\epsilon)}{\Pr\left[\mathcal{B}\left(\{(\boldsymbol{x}^*, y^*)\}\right) = \text{false}\right]}, \quad \text{(A79)}$$

which is trivial and just implies that $\delta > 0$. One can work backwards to obtain the $(\epsilon, \delta)$-DP guarantee. Again repeating similar steps but substituting for $\Pr\left[\mathcal{L}_{\mathcal{B}}\left(\mathcal{D}'\right) \in \mathcal{S}\right]$, yields

$$\Pr\left[\mathcal{L}_{\mathcal{B}}\left(\mathcal{D}\right) \in \mathcal{S}\right]\Pr\left[\mathcal{B}\left(\{(\boldsymbol{x}^*, y^*)\}\right) = \text{false}\right] - \exp(\epsilon)\Pr\left[\mathcal{L}_{\mathcal{B}}\left(\mathcal{D}\right) \in \mathcal{S}\right] \leq \delta, \quad \text{(A80)}$$

and hence

$$\delta \geq \Pr\left[\mathcal{B}\left(\{(\boldsymbol{x}^*, y^*)\}\right) = \text{false}\right] - \exp(\epsilon). \quad \text{(A81)}$$

However, Equation (A81) is also trivial since $\exp(\epsilon) > 1$ for any $\epsilon > 0$.

On the other hand, if we take $\mathcal{S} \subseteq \mathcal{P}\left(\mathcal{O}^{\text{data}}(\mathcal{D}')\right)$, then there are two cases for any $\mathcal{O} \in \mathcal{S}$:

- $y^* \in \mathcal{O}$ implies that $\Pr\left[\mathcal{L}_{\mathcal{B}}\left(\mathcal{D}\right) = \mathcal{O}\right] = 0$.
- $y^* \notin \mathcal{O}$ implies that $\Pr\left[\mathcal{L}_{\mathcal{B}}\left(\mathcal{D}\right) = \mathcal{O}\right] > 0$.

Therefore,

$$
\begin{aligned}
\Pr\left[\mathcal{L}_{\mathcal{B}}\left(\mathcal{D}\right)\in\mathcal{S}\right] &= \sum_{\mathcal{O}\in\mathcal{S}} \Pr\left[\mathcal{L}_{\mathcal{B}}\left(\mathcal{D}\right)=\mathcal{O}\right] \\
&= \sum_{\mathcal{O}\in\mathcal{S}\wedge y^*\in\mathcal{O}} \Pr\left[\mathcal{L}_{\mathcal{B}}\left(\mathcal{D}\right)=\mathcal{O}\right] + \sum_{\mathcal{O}\in\mathcal{S}\wedge y^*\notin\mathcal{O}} \Pr\left[\mathcal{L}_{\mathcal{B}}\left(\mathcal{D}\right)=\mathcal{O}\right] \\
&= 0 + \sum_{\mathcal{O}\in\mathcal{S}\wedge y^*\notin\mathcal{O}} \Pr\left[\mathcal{L}_{\mathcal{B}}\left(\mathcal{D}\right)=\mathcal{O}\right] \\
&= \sum_{\mathcal{O}\in\mathcal{S}\wedge y^*\notin\mathcal{O}} \Pr\left[\mathcal{L}_{\mathcal{B}}\left(\mathcal{D}\right)=\mathcal{O}\right].
\end{aligned}
\tag{A82}
$$

Similar to the previous argument in Equation (A64), this becomes:

$$
\Pr\left[\mathcal{L}_{\mathcal{B}}\left(\mathcal{D}\right)\in\mathcal{S}\right] = \sum_{\mathcal{O}\in\mathcal{S}\wedge y^*\notin\mathcal{O}} \Pr\left[\mathcal{L}_{\mathcal{B}}\left(\mathcal{D}\right)=\mathcal{O}\right]
\tag{A83}
$$

$$
= \sum_{\mathcal{O}\in\mathcal{S}\wedge y^*\notin\mathcal{O}} \Pr\left[\mathcal{L}_{\mathcal{B}}\left(\mathcal{D}'\right)=\mathcal{O}\right] \times \frac{1}{\Pr\left[\mathcal{B}\left(\mathcal{D}'_{y^*}\right)=\text{false}\right]},
\tag{A84}
$$

which implies that

$$
\begin{aligned}
\Pr\left[\mathcal{L}_{\mathcal{B}}\left(\mathcal{D}'\right)\in\mathcal{S}\right] &= \sum_{\mathcal{O}\in\mathcal{S}\wedge y^*\in\mathcal{O}} \Pr\left[\mathcal{L}_{\mathcal{B}}\left(\mathcal{D}'\right)=\mathcal{O}\right] + \sum_{\mathcal{O}\in\mathcal{S}\wedge y^*\notin\mathcal{O}} \Pr\left[\mathcal{L}_{\mathcal{B}}\left(\mathcal{D}'\right)=\mathcal{O}\right] \\
&= \sum_{\mathcal{O}\in\mathcal{S}\wedge y^*\in\mathcal{O}} \Pr\left[\mathcal{L}_{\mathcal{B}}\left(\mathcal{D}'\right)=\mathcal{O}\right] \\
&\quad + \Pr\left[\mathcal{B}\left(\mathcal{D}'_{y^*}\right)=\text{false}\right]\Pr\left[\mathcal{L}_{\mathcal{B}}\left(\mathcal{D}\right)\in\mathcal{S}\right].
\end{aligned}
\tag{A85}
$$

Again, to prove that $\mathcal{L}_{\mathcal{B}}$ is $(\epsilon,\delta)$-DP, we must consider when the maximum difference between:

$$
\Pr\left[\mathcal{L}_{\mathcal{B}}\left(\mathcal{D}\right)\in\mathcal{S}\right] - \exp(\epsilon)\Pr\left[\mathcal{L}_{\mathcal{B}}\left(\mathcal{D}'\right)\in\mathcal{S}\right]
\tag{A86}
$$

or

$$
\Pr\left[\mathcal{L}_{\mathcal{B}}\left(\mathcal{D}'\right)\in\mathcal{S}\right] - \exp(\epsilon)\Pr\left[\mathcal{L}_{\mathcal{B}}\left(\mathcal{D}\right)\in\mathcal{S}\right]
\tag{A87}
$$

occurs and if we choose a $\delta$ larger than both, we can guarantee DP. For the first case, from Equation (A83), observe that:

$$
\Pr\left[\mathcal{L}_{\mathcal{B}}\left(\mathcal{D}\right)\in\mathcal{S}\right] \leq \Pr\left[\mathcal{L}_{\mathcal{B}}\left(\mathcal{D}'\right)\in\mathcal{S}\right] \times \frac{1}{\Pr\left[\mathcal{B}\left(\mathcal{D}'_{y^*}\right)=\text{false}\right]}.
\tag{A88}
$$

Thus,

$$
\begin{aligned}
&\Pr\left[\mathcal{L}_{\mathcal{B}}\left(\mathcal{D}\right)\in\mathcal{S}\right] - \exp(\epsilon)\Pr\left[\mathcal{L}_{\mathcal{B}}\left(\mathcal{D}'\right)\in\mathcal{S}\right] \\
&\leq \frac{\Pr\left[\mathcal{L}_{\mathcal{B}}\left(\mathcal{D}'\right)\in\mathcal{S}\right]}{\Pr\left[\mathcal{B}\left(\mathcal{D}'_{y^*}\right)=\text{false}\right]} - \exp(\epsilon)\Pr\left[\mathcal{L}_{\mathcal{B}}\left(\mathcal{D}'\right)\in\mathcal{S}\right] \\
&= \Pr\left[\mathcal{L}_{\mathcal{B}}\left(\mathcal{D}'\right)\in\mathcal{S}\right]\left(\frac{1}{\Pr\left[\mathcal{B}\left(\mathcal{D}'_{y^*}\right)=\text{false}\right]} - \exp(\epsilon)\right) \\
&\leq \frac{1}{\Pr\left[\mathcal{B}\left(\mathcal{D}'_{y^*}\right)=\text{false}\right]} - \exp(\epsilon),
\end{aligned}
\tag{A89}
$$

which implies

$$
\delta \geq \frac{1}{\Pr\left[\mathcal{B}\left(\{(\boldsymbol{x}^*,y^*)\}\right)=\text{false}\right]} - \exp(\epsilon),
\tag{A90}
$$

and one can work backwards to obtain the DP guarantees. For Equation (A87), the largest possible difference occurs when $\mathcal{S} = \mathcal{S}^* = \left\{\mathcal{O}\subseteq\mathcal{O}^{\text{data}}(\mathcal{D}') : y^*\in\mathcal{O}\right\}$, i.e., only the

subsets of the label space of $\mathcal{D}'$ that contain the new label $y^*$. Which implies that:

$$\Pr\left[\mathcal{L}_\mathcal{B}\left(\mathcal{D}'\right) \in \mathcal{S}^*\right] - \exp(\epsilon)\Pr\left[\mathcal{L}_\mathcal{B}\left(\mathcal{D}\right) \in \mathcal{S}^*\right] = \Pr\left[\mathcal{L}_\mathcal{B}\left(\mathcal{D}'\right) \in \mathcal{S}^*\right] - \exp(\epsilon) \times 0$$
$$= \Pr\left[\mathcal{L}_\mathcal{B}\left(\mathcal{D}'\right) \in \mathcal{S}^*\right]. \tag{A91}$$

Therefore,

$$\delta \geq \Pr\left[\mathcal{L}_\mathcal{B}\left(\mathcal{D}'\right) \in \mathcal{S}\right]. \tag{A92}$$

However, since it holds that the following events satisfy

$$\{\omega \in \Omega : \mathcal{L}_\mathcal{B}\left(\mathcal{D}'\right)(\omega) \in \mathcal{S}^*\} = \{\omega \in \Omega : \mathcal{B}\left(\mathcal{D}'_{y^*}\right)(\omega) = \text{true}\}, \tag{A93}$$

because the mechanism to include $y^*$ in its output, it must be that $\mathcal{B}\left(\mathcal{D}'_{y^*}\right) = \text{true}$, and the other probabilities get marginalized out, and thus

$$\delta \geq \Pr\left[\mathcal{B}\left(\mathcal{D}'_{y^*}\right) = \text{true}\right] = 1 - \Pr\left[\mathcal{B}\left(\mathcal{D}'_{y^*}\right) = \text{false}\right]. \tag{A94}$$

Combining all the previous results from Equation (A71), Equation (A90), and Equation (A92), we obtain:

$$\delta \geq \max\left(1 - \exp(\epsilon)\Pr\left[\mathcal{B}\left(\{(\boldsymbol{x}^*, y^*)\}\right) = \text{false}\right],\right.$$
$$\frac{1}{\Pr\left[\mathcal{B}\left(\{(\boldsymbol{x}^*, y^*)\}\right) = \text{false}\right]} - \exp(\epsilon), \tag{A95}$$
$$\left.1 - \Pr\left[\mathcal{B}\left(\{(\boldsymbol{x}^*, y^*)\}\right) = \text{false}\right]\right).$$

Since for any $\epsilon > 0$, we have $\exp(\epsilon) \geq 1$, which implies that $-\exp(\epsilon) \leq -1$, and thus $1 - \exp(\epsilon)\Pr\left[\mathcal{B}\left(\{(\boldsymbol{x}^*, y^*)\}\right) = \text{false}\right] \leq 1 - \Pr\left[\mathcal{B}\left(\{(\boldsymbol{x}^*, y^*)\}\right) = \text{false}\right]$, then

$$\delta \geq \max\left(\frac{1}{\Pr\left[\mathcal{B}\left(\{(\boldsymbol{x}^*, y^*)\}\right) = \text{false}\right]} - \exp(\epsilon), 1 - \Pr\left[\mathcal{B}\left(\{(\boldsymbol{x}^*, y^*)\}\right) = \text{false}\right]\right), \tag{A96}$$

which ends the proof. $\square$

For example. we can apply Theorem C.3 to the binary mechanism based on the Laplace mechanism $\mathcal{B}_{\text{Lap}}$ to compute $\delta_{\text{Lap}}$. First, since for any singleton $\{(\boldsymbol{x}, y)\}$, we have

$$\Pr\left[\mathcal{B}\left(\{(\boldsymbol{x}, y)\}\right) = \text{true}\right] = \Pr\left[|\{(\boldsymbol{x}, y)\}| + \text{Lap}\left(\frac{1}{\epsilon}\right) \geq \tau\right]$$
$$= \Pr\left[1 + \text{Lap}\left(\frac{1}{\epsilon}\right) \geq \tau\right]. \tag{A97}$$

Define

$$\delta^* = \Pr\left[1 + \text{Lap}\left(\frac{1}{\epsilon}\right) \geq \tau\right], \tag{A98}$$

then observe that,

$$\delta_{\text{Lap}} \geq \max\left(\frac{1}{1 - \delta^*} - \exp(\epsilon), \delta^*\right). \tag{A99}$$

# D  EXPERIMENTAL DETAILS (FOR SEC. 6)

**Pre-trained Model**  Throughout all experiments, we utilise a Vision Transformer VIT-Base-16 (VIT-B) (Dosovitskiy et al., 2021) with 85.8M parameters, pretrained on the ImageNet-21K (Russakovsky et al., 2015) dataset. We assume that the pre-training data (ImageNet-21K) is public, and the downstream data $\mathcal{D}$ is private and needs to be protected with DP.

For all methods but the Cosine Classifier we fine-tune the pre-trained model. We set the weights of the last linear layer of the ViT-B to zero and always learn them when fine-tuning on $\mathcal{D}$. Additionally, we employ Parameter-Efficient Fine-Tuning in some experiments by learning FiLM (Perez et al., 2018) layers. Although there are many other such adapters such as Model Patch (Mudrakarta et al., 2019), LoRA (Hu et al., 2022), CaSE (Patacchiola et al., 2022) etc., we chose FiLM as it has proven to be highly effective in prior works on (DP) parameter-efficient few-shot transfer learning (Shysheya et al., 2023; Tobaben et al., 2023). The number of FiLM parameters for the ViT-B are 38400 as we implement it by freezing all weights but the layer norm scale and bias weights. The the number of the last layer in comparison are $768C + C$ weights, so for example for Split-CIFAR-100 we fine-tune $38400 + 76900 = 115300$ parameters in the Base Dataset Baseline.

We implement our methods using PyTorch (Paszke et al., 2019), continuum (Douillard & Lesort, 2021), and opacus (Yousefpour et al., 2021) with the PRV accounting (Gopi et al., 2021).

**Metrics**  We report the average accuracy and the average forgetting as Chaudhry et al. (2018); Mirzadeh et al. (2021); Yoon et al. (2022):

- **Accuracy**: The average accuracy at task $t$ is defined as

$$\text{average accuracy}_t = \frac{1}{t} \sum_{i=1}^{t} \text{test set accuracy}_{t,i} \in [0, 1], \qquad (A100)$$

  where test set accuracy$_{t,i} \in [0, 1]$ is the test set accuracy for task $i$ after learning task $t$. Note that average accuracy$_T$ is the average accuracy across all tasks after the final task $T$ has been learned.
- **Forgetting**: The forgetting of task $i$ is defined as the difference between its highest accuracy and its accuracy at the current task $t$ as

$$\text{forgetting}_{t,i} = \max_{k \in \{1, \ldots, t-1\}} (\text{test set accuracy}_{k,i} - \text{test set accuracy}_{t,i}) \in [-1, 1], \qquad (A101)$$

  such that the average forgetting at task $t$ is then given by

$$\text{average forgetting}_t = \frac{1}{t-1} \sum_{i=1}^{t-1} \text{forgetting}_{t,i} \in [-1, 1]. \qquad (A102)$$

  Note that average forgetting$_T$ is the average forgetting across the first $T - 1$ tasks as there is no forgetting of the final learned task $T$.

**Data sets**  We experiment with the following data sets:

- **Split CIFAR-100**: Split CIFAR-100 which is CIFAR-100 (Krizhevsky, 2009) split into 10 tasks with 10 classes/task. We randomly permute the class order for each seed we run.
- **Split ImageNet-R**: This data set consists 30,000 images of renditions (art, cartoons, etc.) of 200 ImageNet classes (Hendrycks et al., 2021) and was introduced by Wang et al. (2022a) as a benchmark for CL with pre-trained models. As in Janson et al. (2022); Wang et al. (2022a), we split the classes into 10 tasks with 20 classes/task. We make a random 80/20% train/test split across the whole dataset. The samples per class are imbalanced in the original data set, and we obtain a 41-334 samples/class in our training sets with our split.
- **5-Datasets**: This data set concatenates the five 10-class data sets, MNIST (LeCun et al., 2010), SVHN (Netzer et al., 2011), notMNIST (Bulatov, 2011), FashionMNIST (Xiao et al., 2017) and CIFAR-10 (Krizhevsky, 2009). Each data set is considered a task, such that there are 5 tasks with 10 classes/task. We run experiments with all permutations of the tasks and use it in Appendix F.

## D.1  HYPERPARAMETERS

We tune the hyperparameters for the DP-SGD methods for each combination of privacy budget $(\epsilon, \delta)$ and seed once using the hyperparameter tuning library Optuna (Akiba et al., 2019) with the Gaussian

process (Rasmussen & Williams, 2006) sampler for 20 iterations. The ranges for the hyperparameters can be found in Table A2.

Table A2: Hyperparameter ranges used for the tuning.

|  | LOWER BOUND | UPPER BOUND |
|---|---|---|
| EPOCHS | 40 | |
| LEARNING RATE | 1E-7 | 1E-2 |
| BATCH SIZE | 10 | TUNING DATASET SIZE |
| CLIPPING NORM | 0.2 | 10 |
| NOISE MULTIPLIER | BASED ON TARGET $\epsilon$ | |

The tuning is done using a smaller subset than the final training dataset to reduce the required compute budget. For the 5-dataset datasets (CIFAR-10, FashionMNIST, MNIST, notMNIST and SVHN) we tune using a subset of CIFAR-10 that is 10% the size of the final training dataset and select the hyperparameters that yield the best validation accuracy on a validation dataset of size 4.28% of the final training dataset. For Split-CIFAR-100 and ImageNet-R we tune using a dataset that is representative of one task (10/20 classes) by splitting the representative dataset 70:30 into tuning and validation dataset. For all datasets, we linearly scale the (expected) batch size to keep the subsampling ratio constant for the new training dataset size (Koskela & Kulkarni, 2023) and scale the learning rate of Adam by $\sqrt{scaling\_factor}$ as suggested by prior work (Granziol et al., 2022; Malladi et al., 2022). We do not scale the clipping bound and keep the number of epochs constant at 40.

Similarly to prior work (De et al., 2022; Mehta et al., 2023; Tobaben et al., 2023) we do not account for additional privacy budget spending during the tuning of the hyperparameters.

## D.2 PEFT ENSEMBLE AGGREGATION RULES

We also compared two other aggregation rules in Fig. A2 to the aggregation rule introduced in Eq. (7), which we refer to as ArgMax. The Median aggregation rule is obtained by subtracting the median of the logits for each model separately, given by the following equation:

$$\hat{y}^* = \arg\max_{o\in\cup_{k=1}^t \mathcal{O}_k, l\in\{1,...,t\}} \left( f_l(\boldsymbol{x}^*)_o - \underset{o'\in\cup_{k=1}^t \mathcal{O}_k}{\text{median}} f_l(\boldsymbol{x}^*)_{o'} \right). \tag{A103}$$

We additionally implement the entropy-based aggregation rule proposed by (Zhao et al., 2024a) in Equation (11), using the recommended parameter value $\gamma = 1$ from Section 4.1, which is further supported by the ablation study in Appendix D. We refer to this aggregation rule as Entropy.

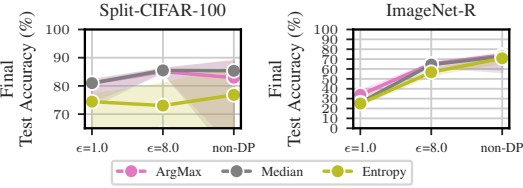

Figure A2: Median accuracy comparison between ArgMax (Eq. (7)), Median (Eq. (A103)), and Entropy (Equation (11) of Zhao et al. (2024a)) aggregation rules for the PEFT model Ensemble. The error bars are the min/max accuracies obtained over five repeats with different class ordering, hyperparameter tuning runs and DP noise.

## D.3 EXPERIMENTS COMPUTE RESOURCES

The expected runtime of the experiments depends on the dataset and method. We run all experiments with the Cosine Classifier on CPU with the runtime being in the order of five minutes per repeat. We use 10 cores of a Xeon Gold 6230 and 10 GB of host memory. We avoid excessive runtime by caching the feature space representations on disk and thus avoid forwarding passes through the model. The experiments that fine-tune a model with FiLM layers (e.g. PEFT Ensemble) use additionally one

NVIDIA V100 GPU with 32 GB of VRAM. The runtime for one repeat is around 16 hours then. All experiments together consumed of the order of 30 GPU days and some CPU days.

### D.4 THE DESAI ET AL. (2021) BASELINE

We implemented an enhanced version of Algorithm 1 of Desai et al. (2021) with enhanced privacy accounting to fine-tune the same ViT-B model (with PEFT) that we used in our experiments, by doing the following:

1. Instead of adding noise to each per-example gradient, we add noise to the sum of per-example gradients of each training batch. Similarly, we add the noise to the sum of per-example gradients of each memory buffer batch.

2. We implement Poisson subsampling for both the training data and the memory buffer.

3. Since for each task $t$, the dataset $\mathcal{D}_t$ is split into two disjoint sets $\mathcal{D}_t^{\text{train}}$ and $\mathcal{D}_t^{\text{ref}}$ ($\mathcal{D}_t = \mathcal{D}_t^{\text{train}} \cup \mathcal{D}_t^{\text{ref}}$ and $\mathcal{D}_t^{\text{train}} \cap \mathcal{D}_t^{\text{ref}} = \varnothing$), we can do the following privacy accounting:

   (a) Parallel composition between the memory buffer DP gradient computations and the training data DP gradient computations.

   (b) The standard DP-SGD sequential composition for the number of epochs required for each task. The projection in step 15 of Algorithm 1 (Desai et al., 2021):

   $$\tilde{g} = g - \frac{g^\top g_{\text{ref}}}{g_{\text{ref}}^\top g_{\text{ref}}} \, g_{\text{ref}}, \tag{A104}$$

   is just post-processing from the training data DP gradient $g$ and the memory buffer DP gradient $g_{\text{ref}}$.

   (c) Parallel composition over all the tasks for the training data DP gradient computations, since it is assumed that each individuals' data appears in only one task's data.

   (d) Since the memory buffer is empty in the first task, and then the memory buffer is nonempty in all the following tasks, we apply $T - 1$ sequential compositions for the memory buffer DP gradient computations (where $T$ is the total number of tasks).

   (e) We take into account the privacy amplification by Poisson subsampling for both the training and memory buffer batches.

   (f) We use the PRV accountant (Gopi et al., 2021) to compute the noise multipliers for $g$ and $g_{\text{ref}}$ separately.

We tried several different memory buffer sizes for CIFAR-100, including 13 samples per class that (Desai et al., 2021) used in their code, and observed that 50 samples per class was the best. This is 10% the size of a class in the CIFAR-100 training data. We also used 10% the size of a training class size for the ImageNet-R dataset, which is 20 samples per class. We used the same DP-SGD hyperparameters that we use in our PEFT method for this experiment.

### D.5 RUNTIME SCALABILITY (W.R.T. NUMBER OF CLASSES)

**Cosine classifier** For training the cosine classifier, the pre-trained model forward pass takes the same time and only depends on the number of classes in the training dataset; however, the prototype noise addition scales linearly with the number of labels in the output label space $|\cup_{i=1}^t \mathcal{O}_i|$. The inference runtime scales linearly with the number of prototypes which is $|\cup_{i=1}^t \mathcal{O}_i|$. In our experiments we did not measure any significant runtime differences in terms of wall-clock runtime.

**PEFT model** On the other hand, training the PEFT model depends on the number of FiLM parameters and the size of the last layer. Theoretically speaking, since the last layer is a linear operation $Wx + b$, the rows of $W$ and the size of the vector $b$ depend on the number of labels in the output label space $|\cup_{i=1}^t \mathcal{O}_i|$, which implies that the gradient computations in DP-SGD for the last layer will also scale linearly with the number of labels. The inference runtime also scales by the number of FiLM parameters and the size of the last layer which depends on the number of labels in the output label space $|\cup_{i=1}^t \mathcal{O}_i|$.

# E   DETAILED RESULTS (FOR SEC. 6)

## E.1   ADDITIONAL FIGURES

Fig. A3 provides an additional illustration of Fig. 7.

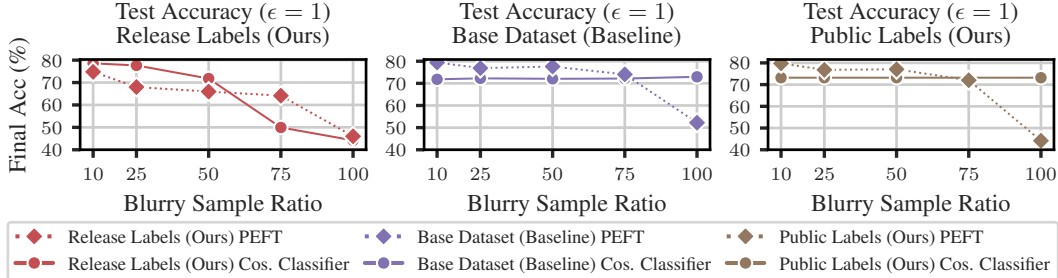

Figure A3: Impact of Blurriness: Increased blurriness degrades the utility at $\epsilon = 1$ when releasing the labels. For PEFT Ensemble this is the case with all label methods at $\epsilon = 1$. We show the median over five repeats at $\delta = 10^{-5}$ (see full results in Table A7). Public Label is $1000\times$ size.

## E.2   TABULAR RESULTS (FOR FIGURES IN SEC. 6)

The following section contains the tabular results for Figs. 4 to 7 in Tables A3 to A7.

Table A3: Performance (final test accuracy in %) of different methods when varying the label space as tabular version for Fig. 4. We report the median as well as the 2nd and 4th best performing seed in parentheses.

| Dataset | $\epsilon$ | Classifier | Label Oracle (Baseline) | Base Dataset (Baseline) | Public Labels (Ours) | Release Labels (Ours) |
|---|---|---|---|---|---|---|
| CIFAR-100 | 1 | Cosine Classifier | (78.3), 78.4, (78.5) | (72.0), 72.3, (72.5) | (72.2), 73.2, (73.2) | (78.4), 78.4, (78.4) |
| | | PEFT Ensemble | (78.5), 79.0, (80.0) | (80.0), 81.0, (81.5) | (80.0), 80.8, (80.9) | (78.3), 79.0, (79.8) |
| | 8 | Cosine Classifier | (79.0), 79.0, (79.0) | (78.8), 78.9, (78.9) | (78.8), 78.8, (78.9) | (79.0), 79.0, (79.0) |
| | | PEFT Ensemble | (84.4), 84.9, (85.1) | (85.0), 85.1, (85.9) | (84.9), 85.7, (85.7) | (84.4), 84.9, (85.1) |
| ImageNet-R | 8 | Cosine Classifier | (56.0), 56.1, (56.2) | (46.5), 46.6, (46.7) | (46.7), 46.8, (46.8) | (55.9), 55.9, (55.9) |
| | | PEFT Ensemble | (61.4), 64.4, (65.6) | (62.6), 64.9, (65.8) | (63.6), 65.2, (65.3) | (61.5), 64.0, (65.4) |

Table A4: Performance (test accuracy in %) of different methods when varying the budget for releasing the labels as tabular form of Fig. 5. We report the median as well as the 2nd and 4th best performing seed in parentheses. We will add one missing cell during rebuttal.

| Dataset | $\epsilon$ | Classifier | Budget 2.5% | Budget: 3.59% | Fraction: 5.99% | Budget: 10% | Budget: 20% | Budget: 30% | Budget: 50% |
|---|---|---|---|---|---|---|---|---|---|
| CIFAR-100 | 1 | Cosine Classifier | (69.5), 71.9, (73.1) | (78.4), 78.5, (78.5) | (78.3), 78.5, (78.5) | (78.2), 78.5, (78.5) | (78.2), 78.2, (78.3) | (77.9), 78.0, (78.1) | (76.7), 76.7, (77.0) |
| | | PEFT Ensemble | (70.5), 72.2, (73.4) | (78.3), 79.0, (79.8) | (78.2), 78.9, (79.6) | (78.0), 78.9, (79.3) | (77.6), 78.5, (78.7) | (76.9), 78.0, (78.1) | (74.6), 75.4, (76.5) |
| ImageNet-R | 8 | Cosine Classifier | (52.6), 52.6, (53.2) | (55.7), 55.8, (55.9) | (55.8), 55.8, (55.8) | (55.7), 55.7, (55.8) | (55.4), 55.5, (55.6) | (54.9), 55.1, (55.1) | (52.6), 52.9, (52.9) |
| | | PEFT Ensemble | - | (60.6), 64.0, (65.1) | (61.5), 64.0, (65.4) | (61.4), 63.9, (65.0) | (60.6), 63.2, (64.2) | (59.4), 62.4, (62.9) | (57.0), 59.8, (60.1) |

Table A5: Number of released labels when varying the the budget for releasing the lables as tabular form of Fig. 5. Both Cosine Classifier and PEFT Ensemble have the same numbers as they are independent from the label release mechanism. We report the median as well as the 2nd and 4th best performing seed in parentheses.

| Dataset | $\epsilon$ | Budget 2.5% | Budget: 3.59% | Fraction: 5.99% | Budget: 10% | Budget: 20% | Budget: 30% | Budget: 50% |
|---|---|---|---|---|---|---|---|---|
| CIFAR-100 | 1 | (87), 90, (93) | (100), 100, (100) | (100), 100, (100) | (100), 100, (100) | (100), 100, (100) | (100), 100, (100) | (100), 100, (100) |
| ImageNet-R | 8 | (164), 164, (167) | (194), 194, (194) | (200), 200, (200) | (200), 200, (200) | (200), 200, (200) | (200), 200, (200) | (200), 200, (200) |

Table A6: Performance (final test accuracy in %) of different methods when varying the label space method as tabular version for Fig. 6. We report the median as well as the 2nd and 4th best performing seed in parentheses.

| Dataset | $\epsilon$ | Classifier | Label Oracle (Baseline) | Base Dataset (Baseline) | Public Labels ($10\times$) | Public Labels ($1000\times$) |
|---|---|---|---|---|---|---|
| CIFAR-100 | 1 | Cosine Classifier | (78.3), 78.4, (78.5) | (72.0), 72.3, (72.5) | (72.1), 72.2, (72.6) | (72.2), 73.2, (73.2) |
| | | PEFT Ensemble | (78.5), 79.0, (80.0) | (80.0), 81.0, (81.5) | (80.0), 81.1, (81.2) | (80.0), 80.8, (80.9) |
| | 8 | Cosine Classifier | (79.0), 79.0, (79.0) | (78.8), 78.9, (78.9) | (78.9), 78.9, (79.0) | (78.8), 78.8, (78.9) |
| | | PEFT Ensemble | (84.4), 84.9, (85.1) | (85.0), 85.1, (85.9) | (85.3), 85.4, (85.6) | (84.9), 85.7, (85.7) |
| ImageNet-R | 1 | Cosine Classifier | (32.9), 33.3, (33.5) | (12.4), 13.1, (13.1) | (12.8), 13.3, (13.4) | (10.0), 10.1, (10.6) |
| | | PEFT Ensemble | (33.2), 33.3, (35.6) | (32.0), 33.2, (35.2) | (30.6), 33.5, (34.9) | (27.4), 27.8, (35.4) |
| | 8 | Cosine Classifier | (56.0), 56.1, (56.2) | (46.5), 46.6, (46.7) | (46.5), 46.8, (46.9) | (46.7), 46.8, (46.8) |
| | | PEFT Ensemble | (61.4), 64.4, (65.6) | (62.6), 64.9, (65.8) | (63.9), 65.7, (65.7) | (63.6), 65.2, (65.3) |

Table A7: Performance (test accuracy in %) of different methods when varying blurry sample ratio as tabular version for Fig. 7. The dataset is Split-CIFAR-100. We report the median as well as the 2nd and 4th best performing seed in parentheses.

| $\epsilon$ | Classifier | Label Method | Blurry Sample Ratio | | | | |
|---|---|---|---|---|---|---|---|
| | | | 10 | 25 | 50 | 75 | 100 |
| 1 | Cosine Classifier | Base Dataset (Baseline) | (72.2), 72.7, (73.1) | (72.6), 72.8, (72.9) | (72.4), 73.0, (73.4) | (72.2), 72.5, (73.7) | (72.5), 73.0, (73.3) |
| | | Public Labels (Ours) | (72.2), 73.2, (73.2) | (72.2), 73.2, (73.2) | (72.2), 73.2, (73.2) | (72.2), 73.2, (73.2) | (72.2), 73.2, (73.2) |
| | | Release Labels (Ours) | (78.5), 78.6, (78.6) | (77.2), 77.6, (77.9) | (70.9), 71.8, (72.1) | (49.2), 49.9, (50.3) | (43.4), 44.2, (44.3) |
| | PEFT Ensemble | Base Dataset (Baseline) | (77.7), 79.4, (80.5) | (70.2), 76.9, (80.5) | (48.2), 77.6, (79.5) | (44.4), 74.0, (74.6) | (44.2), 52.2, (54.5) |
| | | Public Labels (Ours) | (76.3), 78.9, (79.8) | (70.4), 75.4, (76.8) | (48.9), 75.4, (77.1) | (44.4), 72.0, (72.5) | (43.7), 44.2, (44.3) |
| | | Release Labels (Ours) | (73.5), 74.8, (78.6) | (61.1), 68.0, (77.9) | (44.0), 66.0, (75.4) | (43.6), 64.1, (67.0) | (43.8), 46.0, (47.2) |
| 8 | Cosine Classifier | Base Dataset (Baseline) | (78.6), 79.0, (79.0) | (78.9), 79.0, (79.0) | (78.9), 79.0, (79.1) | (78.8), 78.8, (78.9) | (78.8), 78.9, (78.9) |
| | | Public Labels (Ours) | (78.8), 78.8, (78.9) | (78.8), 78.8, (78.9) | (78.8), 78.8, (78.9) | (78.8), 78.8, (78.9) | (78.8), 78.8, (78.9) |
| | | Release Labels (Ours) | (79.0), 79.1, (79.1) | (79.1), 79.1, (79.1) | (78.9), 79.0, (79.2) | (78.8), 79.0, (79.1) | (78.9), 78.9, (79.1) |
| | PEFT Ensemble | Base Dataset (Baseline) | (86.4), 86.6, (86.7) | (86.7), 87.0, (87.1) | (87.3), 87.4, (87.4) | (87.2), 87.3, (87.8) | (86.0), 86.9, (87.1) |
| | | Public Labels (Ours) | (84.9), 85.7, (85.7) | (84.9), 85.7, (85.7) | (84.9), 85.7, (85.7) | (84.9), 85.7, (85.7) | (84.4), 84.9, (85.0) |
| | | Release Labels (Ours) | (83.8), 83.8, (84.5) | (86.0), 86.0, (86.2) | (86.7), 86.8, (87.1) | (86.5), 86.7, (86.7) | (84.8), 86.0, (86.1) |

# F   IMPACT OF VARYING DEGREES OF DOMAIN SHIFT (FOR SEC. 6)

In Fig. A4, we assess the performance of our methods on standard CL benchmarks with varying degrees of domain shift between tasks and pre-training data.

**Baselines**   We compare against the following baselines (more details in Appendix F.1:
- **Naive (Lower)**: We fine-tune one pre-trained model with DP-SGD over all $T$ tasks sequentially, which is a lower bound as no means to mitigate catastrophic forgetting are in place (Alg. A1).
- **Non CL Baseline (Upper)**: We fine-tune one pre-trained model with DP-SGD with all data at once, showing the cost of DP training without CL since there is no split into tasks (Alg. A2).

PEFT Ensemble outperforms the Cosine Classifier in all experiments in test accuracy but is more expensive in storage and compute. Fig. A4 (left) shows that the Cosine Classifier is a viable alternative when storage or compute are limited while the domain shift to the pre-training data is small. However, with larger domain shift (see right panel of Fig. A4), PEFT is the only option to achieve good privacy/utility trade-offs. All methods show similar trends when the privacy budget changes, but the PEFT Ensemble has a larger variability in utility, especially without DP, due to the overconfidence of models on unseen classes. Neither method is particularly affected by growing shifts between tasks (as can be seen with 5-dataset in Appendix F.4).

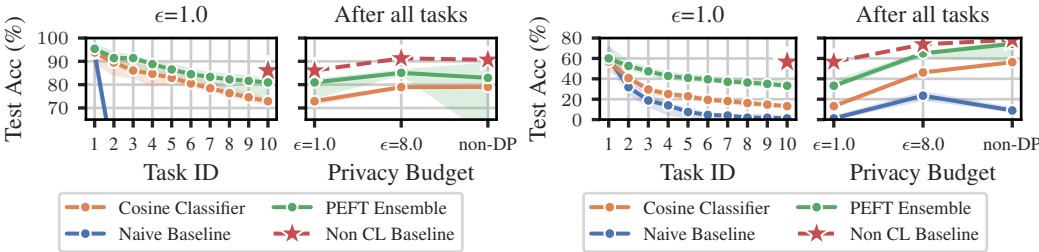

Figure A4: Split-CIFAR-100 (left) and Split-ImageNet-R (right) with $\delta=10^{-5}$: PEFT Ensemble outperforms, but Cosine Classifier feasible at lower domain shift (left). Detailed results in Appendices F.2 to F.4

### F.1 BASELINE DETAILS

Below, we provide pseudo-codes for the baselines introduced in Appendix F.

#### F.1.1 NAIVE BASELINE

We start training a model $f$ at task $t = 1$ under DP and continue further training it for all tasks $T$ using DP-SGD. This is a lower bound as no measures are in place to mitigate catastrophic forgetting.

---
**Algorithm A1** Naive Baseline
---
1: **Initialise model** $f$
2: $f \leftarrow \text{init}()$
3: **for** $t \in [T]$ **do**
4:     **Continue to train the model with the data of the current task** $t$
5:     $f \leftarrow \text{DP-SGD}(f; D_t)$
6: **end for**

**Output:** a set of $T$ model checkpoints $\{f_1 \ldots f_T\}$ (note that this is for evaluating)

**Test of the models**

1: **for** $t \in [T]$ **do**
2:     **Predict test data of all tasks seen so far**
3:     **for** $x_i^* \in \cup_i^t \mathcal{D}_i^{\text{test}}$ **do**
4:         $\hat{y_i}^* \leftarrow \arg\max_k f_t(x_i*)$
5:     **end for**
6:     **Evaluate average accuracy and forgetting**
7:     average accuracy$_t$, forgetting$_t \leftarrow \text{eval}(\{y_i \ldots\}, \{y_i^* \ldots\}, \text{per-task accuracy history})$
8: **end for**

**Output:** average accuracy for all tasks $T$ {average accuracy$_1$ ... average accuracy$_T$}, average forgetting for all tasks $T$ {forgetting$_1$ ... forgetting$_T$}

---

#### F.1.2 FULL DATA BASELINE

We assume the availability $\cup_i^t \mathcal{D}_i$ and train a model with DP-SGD (Abadi et al., 2016). While this is DP it is not CL.

---
**Algorithm A2** Full Data Baseline
---
**Training of the model**
1: **Initialise model** $f$
2: $f \leftarrow \text{init}()$
3: **Train a model with all the data**
4: $f \leftarrow \text{DP-SGD}(f; \cup_i^T D_i)$

**Output:** a model $f$

**Test of the models**

1: **Predict test data**
2: **for** $x_i^* \in \cup_i^T \mathcal{D}_i^{\text{test}}$ **do**
3:     $\hat{y_i}^* \leftarrow \arg\max_k f(x_i*)$
4: **end for**
5: **Evaluate average accuracy**
6: average accuracy $\leftarrow \text{eval}(\{y_i \ldots\}, \{y_i^* \ldots\})$

**Output:** average accuracy

---

## F.2 SPLIT-CIFAR-100

This subsection complements the results of Fig. A4 (left).

Table A8: Average accuracy (AA) and average forgetting (AF) (scaled by 100) after learning the final task in % on 10-task Split-CIFAR-100. We report the mean and std of the metrics averaged over 5 seeds. We did not compute the AF numbers of the naive baseline at other privacy levels than $\epsilon = 1$ as the performance is expected to be bad.

| | $\epsilon = 1, \delta = $1e-5 | | $\epsilon = 8, \delta = $1e-5 | | non-DP | |
| Method | AA ($\uparrow$) | AF ($\downarrow$) | AA ($\uparrow$) | AF ($\downarrow$) | AA ($\uparrow$) | AF ($\downarrow$) |
|---|---|---|---|---|---|---|
| Naive | $9.35 \pm 0.14$ | $94.10 \pm 0.00$ | $9.55 \pm 0.23$ | - | $9.63 \pm 0.05$ | - |
| Cosine classifier | $72.78 \pm 0.51$ | $9.92 \pm 0.51$ | $78.93 \pm 0.10$ | $6.15 \pm 0.24$ | $79.02 \pm 0.00$ | $6.02 \pm 0.56$ |
| PEFT Ensemble (FiLM) | $79.79 \pm 3.02$ | $7.17 \pm 2.56$ | $85.26 \pm 0.633$ | $6.07 \pm 0.59$ | $79.39 \pm 12.20$ | $10.43 \pm 9.28$ |
| Non CL Baseline (FiLM) | 86.06 | - | 91.31 | - | 90.68 | - |
| PEFT Ensemble (last layer) | $78.81 \pm 0.48$ | $0.06 \pm 0.00$ | $82.48 \pm 0.31$ | - | $82.60 \pm 0.38$ | - |
| Non CL Baseline (last layer) | 85.1 | - | 88.5 | - | 88.4 | - |

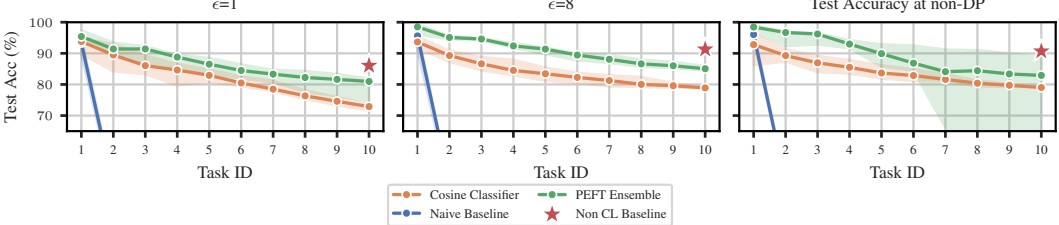

Figure A5: Median test accuracy per task on Split-CIFAR-100 (ViT-B pre-trained on ImageNet-21k). The error bars are the min/max accuracies obtained over five repeats with different class ordering, hyperparameter tuning runs and DP noise.

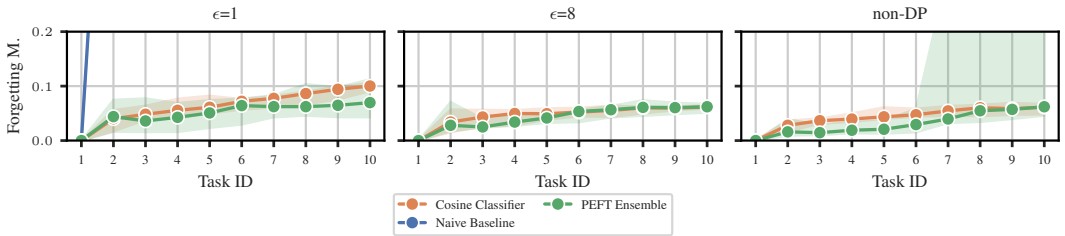

Figure A6: Median forgetting measure per task on Split-CIFAR-100 (ViT-B pre-trained on ImageNet-21k). The error bars are the min/max forgetting measure obtained over five repeats with different class ordering, hyperparameter tuning runs and DP noise.

### F.3 IMAGENET-R

This subsection complements the results of Fig. A4 (right).

Table A9: Average accuracy (AA) and average forgetting (AF) (scaled by 100) after learning the final task in % on 10-task Split ImageNet-R. We report the mean and std of the metrics averaged over three seeds.

| Method | $\epsilon = 1, \delta = $1e-5 | | $\epsilon = 8, \delta = $1e-5 | | non-DP | |
| | AA ($\uparrow$) | AF ($\downarrow$) | AA ($\uparrow$) | AF ($\downarrow$) | AA ($\uparrow$) | AF ($\downarrow$) |
|---|---|---|---|---|---|---|
| Naive | $1.52 \pm 0.79$ | $37.90 \pm 8.05$ | $23.49 \pm 2.50$ | $66.92 \pm 1.94$ | $9.12 \pm 0.99$ | $87.51 \pm 1.59$ |
| Cosine classifier | $13.04 \pm 1.23$ | $13.89 \pm 2.97$ | $46.17 \pm 0.21$ | $12.21 \pm 2.15$ | $56.30 \pm 0.00$ | $7.51 \pm 1.49$ |
| PEFT Ensemble (FiLM) | $33.19 \pm 2.37$ | $12.22 \pm 2.30$ | $64.91 \pm 1.75$ | $8.87 \pm 0.93$ | $74.32 \pm 7.63$ | $6.07 \pm 1.11$ |
| Non CL Baseline (FiLM) | $56.62$ | - | $73.77$ | - | $78.24$ | - |
| PEFT Ensemble (last layer) | $7.29 \pm 2.82$ | $3.54 \pm 0.62$ | $47.97 \pm 2.22$ | $6.33 \pm 0.72$ | $48.16 \pm 12.80$ | $6.54 \pm 4.19$ |
| Non CL Baseline (last layer) | $16.57 \pm 2.86$ | - | $52.16 \pm 4.35$ | - | $62.17 \pm 2.03$ | - |

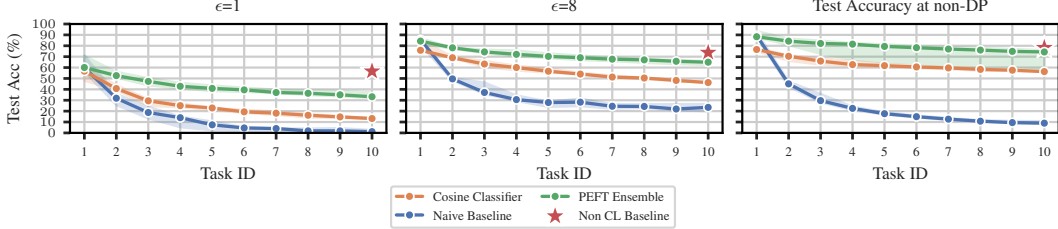

Figure A7: Median test accuracy per task on ImageNet-R (ViT-B pre-trained on ImageNet-21k). The error bars are the min/max accuracies obtained over five repeats with different class ordering, hyperparameter tuning runs and DP noise.

## F.4 5-DATASET

5-Datasets (Ebrahimi et al., 2020) consist of the five datasets, MNIST (LeCun et al., 2010), SVHN (Netzer et al., 2011), notMNIST (Bulatov, 2011), FashionMNIST (Xiao et al., 2017) and CIFAR-10 (Krizhevsky, 2009) where each forms one task.

In this section we analyse the results the results of 5-datasets. They are displayed in Figs. A8 and A9 and Table A10.

Table A10: Average accuracy (AA) and average forgetting (AF) (scaled by 100) after learning the final task in % on 5-dataset. We report the mean and std of the metrics averaged over all task order permutations.

| | $\epsilon = 1, \delta =$1e-5 | | $\epsilon = 8, \delta =$1e-5 | | non-DP | |
|---|---|---|---|---|---|---|
| Method | AA ($\uparrow$) | AF ($\downarrow$) | AA ($\uparrow$) | AF ($\downarrow$) | AA ($\uparrow$) | AF ($\downarrow$) |
| Naive | $15.93 \pm 1.35$ | $85.00 \pm 5.00$ | $17.56 \pm 1.35$ | $90.00 \pm 2.00$ | $13.15 \pm 5.84$ | $79.00 \pm 12.00$ |
| Cosine classifier | $58.54 \pm 0.35$ | $1.00 \pm 0.00$ | $59.78 \pm 0.07$ | $0.00 \pm 0.00$ | $59.87 \pm 0.00$ | $0.00 \pm 0.00$ |
| PEFT Ensemble (FiLM) | $79.69 \pm 3.51$ | $5.00 \pm 4.00$ | $87.83 \pm 0.00$ | $3.00 \pm 2.00$ | $65.75 \pm 15.07$ | $14.00 \pm 11.00$ |

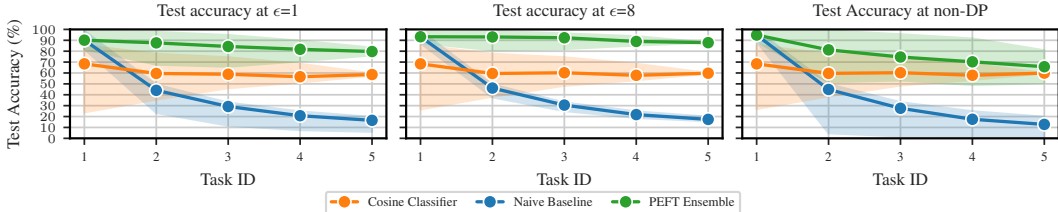

Figure A8: Median test accuracy per task on 5-dataset (ViT-B pre-trained on ImageNet-21k). The error bars are the min/max test accuracy obtained over all permutations of tasks after training three models with hyperparameter tuning runs and DP noise.

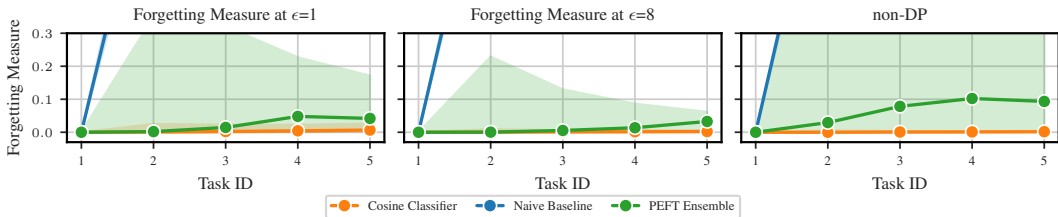

Figure A9: Median forgetting measure per task on 5-dataset (ViT-B pre-trained on ImageNet-21k). The error bars are the min/max forgetting measure obtained over all permutations of tasks after training three models with hyperparameter tuning runs and DP noise.

## G   LICENSES AND ACCESS FOR MODELS AND DATASETS

The Vision Transformer ViT-Base-16 (ViT-B) (Dosovitskiy et al., 2021) is licensed with the Apache-2.0 license and can be obtained through the instructions on `https://github.com/google-research/vision_transformer`.

The licenses and means to access the data sets can be found below. We downloaded all data sets but notMNIST and ImageNet-R from torchvision (version 0.18.1).

- CIFAR10 (Krizhevsky, 2009) is licensed with an unknown license and the data set as specified on `https://pytorch.org/vision/main/generated/torchvision.datasets.CIFAR10.html#torchvision.datasets.CIFAR10`.
- CIFAR100 (Krizhevsky, 2009) is licensed with an unknown license and we the data set as specified on `https://pytorch.org/vision/stable/generated/torchvision.datasets.CIFAR100.html#torchvision.datasets.CIFAR100`.
- FashionMNIST (Xiao et al., 2017) is licensed under MIT and we use the data set as specified on `https://pytorch.org/vision/stable/generated/torchvision.datasets.FashionMNIST.html#torchvision.datasets.FashionMNIST`.
- ImageNet-R (Hendrycks et al., 2021) is licensed with an unknown license and we use the version from `https://people.eecs.berkeley.edu/~hendrycks/imagenet-r.tar`.
- MNIST (LeCun et al., 2010) is licensed with an unknown license and wethe data set as specified on `https://pytorch.org/vision/stable/generated/torchvision.datasets.MNIST.html#torchvision.datasets.MNIST`.
- notMNIST (Bulatov, 2011) is licensed under an unknown license and we use the version at git hash 339df59 found at `https://github.com/facebookresearch/Adversarial-Continual-Learning/blob/main/data/notMNIST.zip`
- SVHN (Netzer et al., 2011) is licensed under CC and the data set as specified on `https://pytorch.org/vision/stable/generated/torchvision.datasets.SVHN.html#torchvision.datasets.SVHN`.

## H    LLM USAGE

We primarily used large language models (LLMs) to assist with the editing of this manuscript. Particularly, to improve the readability and conciseness of certain sentences. We used LLMs for code completion in visual studio code, but manually verified the suggestions. The models were not used for generating novel content.

