# OpenReview forum: "Privacy Leakage via Output Label Space and Differentially Private Continual Learning"
_ICLR.cc/2026/Conference — Submitted to ICLR 2026_

### Official Review · Reviewer_wjgU · 2025-10-29

**Soundness:** 2
**Presentation:** 2
**Contribution:** 1
**Rating:** 4
**Confidence:** 2

**Summary:**

This paper studies privacy leakage in publishing output label space, and proposes algorithm of differentially private(DP) continual learning(CL) algorithm. The basic idea is that the output label space is privacy side channel that can leak information on the training data. In the case of continual training, this is a severe issue. The authors use methods from private partition selection to generate a private output label space, then propose multiple continual learning algorithms that take the output label space into consideration. Experiments are conducted to compare different label space constructions and DP CL algorithms.

**Strengths:**

- The observation of output label space as a privacy side channel is novel and interesting.
- The experiments are extensive.

**Weaknesses:**

- I do not quite follow the concept of "output label space". The authors say it is "the set of labels that a model outputs". But over what input? Is it all possible output labels that the model can produce? For ordinary classifiers like logistic regression or Vit used in this paper, the output space is known in advance; in the end the output is a distribution on each possible class. The authors proposes a scenario where the learning has multiple phases, and in each phase new labels are included gradually. Does this scenario has practical applications? It seems that for each phase ("task" in the paper) we need to release a model; of course in this way the output space will be revealed,  but why does this have to be a privacy side channel? Prop 4.1 suggests that revealing the labels of the training data leaks privacy. Of course this is true, but this is because you are releasing part of the input, it has nothing to do with the ordinary DP training algorithms. After all, the model does not have to fit on all training samples. More clarification on the motivation and use case of the output label space is highly appreciated.
- The contribution seems to be limited. There are 2 constructions of private label space, but they both seem to come from the existing work [1]. There are also 2 DP CL algorithms, one is compute the embeddings for the labels, the other is to use PEFT. It looks like this work just combines existing ideas, and novelty/technical contribution is limited.

[1] Desfontaines, D., Voss, J., Gipson, B., & Mandayam, C. (2022). Differentially private partition selection. Proceedings on Privacy Enhancing Technologies, 1, 339-352.

**Questions:**

When I check the code, I can only see "The requested file is not found." Can you double check if the repo is working?

---

> ### Author Response · Authors · 2025-11-18
>
> We appreciate your thoughtful feedback and valuable suggestions. We will address question/comment separately:
>
> > I do not quite follow the concept of "output label space". The authors say it is "the set of labels that a model outputs". But over what input? Is it all possible output labels that the model can produce? For ordinary classifiers like logistic regression or Vit used in this paper, the output space is known in advance; in the end the output is a distribution on each possible class.
>
> Thank you for raising this point. The output label space refers to the set of labels the model is configured to predict. We edited the second paragraph in Section 1 to explain this more clearly. The labels are not necessarily known in advance, if you don’t know what the task is, e.g. “MNIST classification”.
>
> Particularly in continual learning, one does not know the task in advance (see the answer to the next question). Moreover, the output label space of the model does not have to match the labels in the dataset. Taking the model’s output labels directly from the dataset is not DP according to Proposition 4.1 and Figure 1.
>
> > The authors proposes a scenario where the learning has multiple phases, and in each phase new labels are included gradually. Does this scenario has practical applications?
>
> Thank you for raising this point. This is a standard setting called continual learning (CL) (see Wang et al., 2024 [1] where you can also find details regarding practical applications). In continual learning (CL), a continuous stream of tasks arrive sequentially. In each task, a new dataset is introduced, and we want to update the model without forgetting past tasks. These task datasets are not known in advance.
>
> Work on DP CL already exists which we cite in our paper (Desai et al., 2021; Hassanpour et al., 2022; Lai et al., 2022). However, one of our main contributions is identifying a DP issue that these works overlook, which is how to choose the labels for the model(s) as new classes appear. The reviewer can kindly refer to “Continual Learning” of Section 3 and Section 4 for more details about this.
>
> [1] Wang L., Zhang X., Su H., and Zhu, J. (2024) “A Comprehensive Survey of Continual Learning: Theory, Method and Application”. https://arxiv.org/abs/2302.00487
>
> > It seems that for each phase ("task" in the paper) we need to release a model; of course in this way the output space will be revealed, but why does this have to be a privacy side channel? Prop 4.1 suggests that revealing the labels of the training data leaks privacy. Of course this is true, but this is because you are releasing part of the input, it has nothing to do with the ordinary DP training algorithms. After all, the model does not have to fit on all training samples.
>
> You are correct in that the privacy leakage is due to releasing information about the input, which is precisely that there is at least one sample associated with a given label. Thus, releasing the labels directly from the sensitive dataset violates the DP according to Proposition 4.1 (Section 4.1).
>
> We mitigate this issue by either using an optimal method used in solving a specific private partition selection problem (Desfontaines et al., 2022) that matches the problem of selecting the labels from the dataset (Section 4.2), or by using a data-independent prior label space (Section 4.3).
>
> We are the first to point out this privacy side-channel in classification models and apply methods to mitigate this in CL, which the other reviewers, especially reviewer EmMc and K2LY have acknowledged as an important contribution, and we kindly refer you to their reviews.
>
> > When I check the code, I can only see "The requested file is not found." Can you double check if the repo is working?
>
> We apologize for this, the open science anonymized repo service apparently was down as noted by other users, kindly see:
> * https://github.com/tdurieux/anonymous_github/issues/545 (on October 26), and
> * https://github.com/tdurieux/anonymous_github/issues/551 (on November 6).
>
> Now it is working, and we double checked that as you requested. Please inform us the problem persists, and we will find an alternative way to upload the code.

---

> > ### Comment · Reviewer_wjgU · 2025-11-24
> >
> > Thank you for the response and the revised draft. Some of my questions are answered, but I still do not follow the part of "The labels are not necessarily known in advance". Let us just focus on ViT, the model you use in the experiment. If I understand correctly, in the end there is a classification layer that generates a distribution over possible labels. Namely, the possible labels are merely determined by the model structure. Are you suggesting that your model structure will vary based on input labels? Furthermore, I do check the newly added explanation in Sec 1, but it seems that you are assuming that the model should be able to fit all training samples, which essentially reveals all input labels. But why is this assumption necessary in model training? What if we just discard the training samples that have "outlier" labels?

---

> > > ### Author Response · Authors · 2025-11-26
> > >
> > > > Some of my questions are answered, but I still do not follow the part of "The labels are not necessarily known in advance". Let us just focus on ViT, the model you use in the experiment. If I understand correctly, in the end there is a classification layer that generates a distribution over possible labels. Namely, the possible labels are merely determined by the model structure. Are you suggesting that your model structure will vary based on input labels?
> > >
> > >
> > > Thank you for raising this point. Yes the model will vary, for example in the PEFT ensemble, the ViT will have a different classification layer/head for each task. If there are T tasks, then we have T different PEFT parameters and classification heads. We need to determine each classification head's size and the label associated with each logit. If these properties are determined based on the task training dataset directly (e.g. in an automated system or pipeline), then one of our main contributions is showing that this is not DP (Proposition 4.1 in Section 4).
> > >
> > >
> > > > Furthermore, I do check the newly added explanation in Sec 1, but it seems that you are assuming that the model should be able to fit all training samples, which essentially reveals all input labels. But why is this assumption necessary in model training? What if we just discard the training samples that have "outlier" labels?
> > >
> > > Thank you for raising this point. The argument in Figure 1 is about any system or pipeline that trains a classifier based on an incoming dataset. The composition of these datasets (including the labels) are assumed to be unknown in advance, that is why the system has to determine the output labels somehow (e.g. to determine the size of the last layer for the ViT and the concept/label corresponding to each logit).
> > >
> > > Figure 1 illustrates a common setting in CL where the classifier output label space is determined directly from the training datasets. In DP, one does not know which dataset the classifier is going to be trained on, you only observe one of the two adjacent training dataset, whether it has an “outlier” or not.
> > >
> > > Filtering out an “outlier”, i.e. when there is only one individual associated with the label, does not automatically render the system to be DP. Let’s say the filtering criteria is: remove the label if only one individual is associated with it, and let’s assume that D has a label $y$ with only two individuals’ data associated with it, e.g., $(x_1, y)$ and $(x_2, y)$. Since the adjacency is add/remove, we can define D’ by removing $(x_2, y)$, so D’ only has $x_1$ associated with $y$. Feeding D’ into the system will make the label $y$ disappear from the output label space, because $(x_1, y)$ will get filtered out by the filtering criteria. This argument can be expanded to any filtering criteria that produces different outputs for two adjacent training datasets D and D’. See the generalization of Proposition 4.1 under the proof in Appendix C.1.
> > >
> > > To improve the confidence in our theoretical results, we formalise the main proofs in Lean 4 which can be found in (to verify, you can paste the code in https://live.lean-lang.org/): https://anonymous.4open.science/r/iclr-2026-rebuttal-058E/DPCL.lean

---

### Official Review · Reviewer_HnBC · 2025-10-30

**Soundness:** 2
**Presentation:** 2
**Contribution:** 2
**Rating:** 4
**Confidence:** 3

**Summary:**

This paper focuses on privacy leakage in differentially private continual learning. Existing DP-CL methods use the entire privacy budget for training weights, which protects weights under DP guarantee. However, the output label space, which is naively derived from sensitive data, can lead to a huge privacy leakage. Therefore, the author proposes two methods to address this: spending part of the privacy budget on releasing labels and leveraging a large public prior label set independent of sensitive data. Both theoretical analysis and experiments with pre-trained models on CL benchmarks demonstrate that both methods effectively eliminate the privacy leakage while maintaining utility under various continual learning scenarios.

**Strengths:**

+ Important topic

**Weaknesses:**

- Strong assumptions
- Presentation needs improvement

**Questions:**

1. In line 121, “eliminate” was incorrectly written as “elimiate”. The author should carefully check the whole paper for similar errors.

2. The definition of task-level DP assumes each data point only appears in one task, and then they get privacy guarantees through parallel composition. However, when a user's data spans multiple tasks, the parallel composition no longer holds, and the privacy safeguard requires reassessment.

3. In the first method $S_{learned}$, the privacy budget was divided between selecting class labels and training weights. For model training, the allocated privacy budget is reduced. I am wondering how much this impacts the model performance? And in the experiments, the grid search for the budget division is purely heuristic and lacks theoretical analysis. The author should discuss how to optimally allocate the privacy budget to enhance utility.

4. In the second method $S_{prior}$, it relies on a large data-independent prior label that covers all or most private labels. However, in many real-world scenarios, such a public dataset may not exist. The paper's experiments simulate this by “artificially enlarge the label space,” thereby sidestepping the most critical challenge in real-world scenarios—how to construct such a dataset and the related costs. The author should add some discussion about this.

5. In the $S_{prior}$, when the dataset label is not in the prior label set, these unmatched labels are remapped or simply dropped. The paper does not discuss how the label noise introduced by such remapping or dropping affects model performance and privacy.

---

> ### Author Response · Authors · 2025-11-18
>
> We appreciate your thoughtful feedback and valuable suggestions. We will address question/comment separately:
>
> > 1. In line 121, “eliminate” was incorrectly written as “elimiate”. The author should carefully check the whole paper for similar errors.
>
> Thank you for raising this point. We fixed the typos and updated the manuscript.
>
> > 2. The definition of task-level DP assumes each data point only appears in one task, and then they get privacy guarantees through parallel composition. However, when a user's data spans multiple tasks, the parallel composition no longer holds, and the privacy safeguard requires reassessment.
>
> Thank you for raising this point. To make the main paper more readable, we decided to keep a simple version of the task-wise adjacency relation (Definition 3.2) in Section 3, which matches the experimental setting we have. In Appendix B.2, you can find an extended version (Definition B.4), which applies to the case when a user’s data spans multiple tasks. See also the discussion at the beginning of Appendix B.3. about general composition methods. We added this discussion to Section 3 of the manuscript.
>
>
> > 3. In the first method $S_{learned}$, the privacy budget was divided between selecting class labels and training weights. For model training, the allocated privacy budget is reduced. I am wondering how much this impacts the model performance? And in the experiments, the grid search for the budget division is purely heuristic and lacks theoretical analysis. The author should discuss how to optimally allocate the privacy budget to enhance utility.
>
> Thank you for raising this point. While we can compute bounds for the probability of dropping classes (see Figure 2, Section 4.2), computing exact bounds for the utility of DP models (cosine classifier, PEFT Ensemble) is very difficult. Such bounds require strong and possibly unrealistic assumptions about cluster structure or prototype separation, so that the noise addition might behave in a more predictable manner. We therefore treat optimal split selection as future work and state this limitation explicitly in Section 7.
>
> > 4. In the second method $S_{prior}$, it relies on a large data-independent prior label that covers all or most private labels. However, in many real-world scenarios, such a public dataset may not exist.
>
> Thank you for raising this point. We agree that this scenario might happen, and thus it is advised in this case to use $S_{learned}$ instead with the optimal label release mechanism. Please refer to “Recommendations for Label Methods” in Section 7.
>
> > 5. In the $S_{prior}$, when the dataset label is not in the prior label set, these unmatched labels are remapped or simply dropped. The paper does not discuss how the label noise introduced by such remapping or dropping affects model performance and privacy.
>
> We do show how both remapping and dropping labels affect privacy in Proposition 4.2 of Section 4.3. Dropping unmatched labels behaves similarly to dropping labels in the DP label-release mechanism, as shown in Sec. 6.1 (“Details on Release Labels”).
>
> However, it is very difficult to say anything general about label remapping and is highly problem-dependent, i.e. particularly depends on the label hierarchy. For example, one can use the International Classification of Diseases 11 (ICD-11) disease taxonomy as a prior set of labels for disease classification. ICD-11 categorizes the diseases into certain sections at each level, which become more specific as you go deeper. For an unknown disease, one can remap it to one of these section codes at a particular level.

---

> > ### Comment · Reviewer_HnBC · 2025-11-25
> >
> > Thank you for your detailed responses and revisions. I have two additional suggestions to further improve the paper's completeness and practical value:
> >
> > - The privacy budget split in $S_{learned}$: I understand the theoretical difficulty of determining the optimal split. Nevertheless, the practical usability of the $S_{learned}$ method relies heavily on finding this optimal ratio. I suggest the authors add some heuristic guidelines (based on the empirical results in Figure 4 and Tables A4/A5) to the paper. For example, could the optimal split be related to dataset characteristics like the number of classes or the proportion of small classes? Should more budget be allocated to label release if there are many small classes, as they are more likely to be dropped (Figure 2)? Such discussion would enhance the practical guidance for using this method.
> >
> > - The utility impact of remapping in $S_{prior}$: You state in your comments that the impact of label remapping on utility is "highly problem-dependent". The ICD-11 example you mentioned effectively illustrates the importance of label structure. I recommend incorporating this example into the discussion section (Section 7) or the $S_{prior}$ section (Section 4.3). And briefly discuss the expected utility trade-off of remapping: for example, mapping to a less specific/higher-level label might improve generalization but reduce fine-grained classification performance. This would enhance the robustness of the overall discussion.

---

> > > ### Author Response · Authors · 2025-11-26
> > >
> > > > I suggest the authors add some heuristic guidelines (based on the empirical results in Figure 4 and Tables A4/A5) to the paper. For example, could the optimal split be related to dataset characteristics like the number of classes or the proportion of small classes? Should more budget be allocated to label release if there are many small classes, as they are more likely to be dropped (Figure 2)? Such discussion would enhance the practical guidance for using this method.
> > >
> > > Thank you for this valuable suggestion. For a very simple heuristic based on our baselines and experiments, when $\epsilon \geq 1$, one can set the budget split percentage for $S_{learned}$ as 10% of the target privacy budget, i.e. $(0.1 \times \epsilon, 0.1 \times \delta)$.
> > >
> > > If one has data-independent prior knowledge about the class sizes (e.g. each class size is at least 500 samples), then it is possible to use the distribution of the mechanism in Equation 3 of Section 4.2 to find the minimum budget split percentage that will enable the mechanism to release all the labels with numerical certainty, i.e. with probability $\geq 1 - \text{error}$. However, this is not guaranteed to give an optimal accuracy, since sometimes a slightly lower privacy for the training mechanism (higher noise) can improve generalization.
> > >
> > > If there is no data-independent prior knowledge about the class sizes, then we cannot release the minimum privacy budget split percentage for $S_{learned}$, if it is computed based on the sensitive training data class sizes, since this is not DP. This returns us to our original problem, and would require another mechanism to privatize the budget split percentage. This new mechanism would also need a privacy budget split.
> > >
> > > We added this discussion to Section 7 of the paper as requested.
> > >
> > > > The utility impact of remapping in S_prior You state in your comments that the impact of label remapping on utility is "highly problem-dependent". The ICD-11 example you mentioned effectively illustrates the importance of label structure. I recommend incorporating this example into the discussion section (Section 7) or the section (Section 4.3). And briefly discuss the expected utility trade-off of remapping: for example, mapping to a less specific/higher-level label might improve generalization but reduce fine-grained classification performance. This would enhance the robustness of the overall discussion.
> > >
> > > Thank you again for your valuable suggestions. We added this example and the discussion about generalization to Section 4.3 of the paper as requested.

---

### Official Review · Reviewer_K2LY · 2025-10-31

**Soundness:** 4
**Presentation:** 4
**Contribution:** 3
**Rating:** 6
**Confidence:** 3

**Summary:**

The paper identifies the output label space in continual learning (CL) as a potential privacy side-channel, showing that releasing the set of available labels can leak sensitive information even from DP-trained models. To mitigate this, the authors propose two strategies: (1) spending part of the privacy budget to privately release labels via an optimal DP partition-selection mechanism, and (2) operating within a large, public, data-independent label space. They adapt DP cosine classifiers and DP PEFT ensembles to evaluate both approaches on CIFAR-100 and ImageNet-R, showing that private label release generally achieves better utility–privacy trade-offs.

**Strengths:**

* Privacy side-channels are an important and underexplored topic. This is often overlooked in Differential Privacy research and real-world deployments. Showing that model architecture itself can leak data is a meaningful contribution.
* The label-space leakage in continual learning is well-motivated and clearly demonstrated.
* Mitigation strategies are theoretically sound and consistent with DP best practices.
* The paper is well written and easy to follow, with clear threat models, definitions, and reproducibility details. I wasn't very familiar with the continual learning space, but was able to understand the setup easily.
* Most reported results seem intuitive and correct.

**Weaknesses:**

* The methodological novelty is somewhat limited. Both mitigation strategies (DP partition selection, data-independent priors) are well-known; the contribution lies mainly in _applying_ them to CL. While this is valuable, it makes the work more incremental than conceptual.
* Figure 3 results are a bit puzzling: for the Cosine Classifier, the accuracy of the label-oracle baseline barely changes between $\epsilon=1$ and  $\epsilon=8$, suggesting either the DP noise has little effect or the privacy accounting isn’t tight enough to show a trend. This deserves clarification.
* The experiments lack comparison to existing DP-CL baselines, which would contextualize how severe the leakage is in prior methods.

**Questions:**

Figure 4 (CIFAR-100): the first point left of the red dashed line shows nearly all labels (>99%) released, yet accuracy drops by more than 5%. Can you clarify why this could be the case? I would suggest plotting "maximum possible accuracy" given the available labels and the exact test set class composition.

---

> ### Author Response · Authors · 2025-11-18
>
> We appreciate your thoughtful feedback and valuable suggestions. We will address question/comment separately:
>
> > The methodological novelty is somewhat limited. Both mitigation strategies (DP partition selection, data-independent priors) are well-known; the contribution lies mainly in applying them to CL. While this is valuable, it makes the work more incremental than conceptual.
>
> We appreciate the reviewer’s perspective and recognize that we may not have stressed enough the novel parts of our approach.
>
> The core novelty is identifying the label-space side-channel, which has not been discussed or formalized in any prior DP work. The mitigation strategies are probably well known in other domains, but their application to CL with pre-trained models—and the privacy implications for evolving label spaces—are new.
>
> We would also like to point out that one of our main contributions, which we might have not stressed enough, is that we are the first to use pretrained models in DP CL on top of the other mitigation strategies. We amended the last sentence in Section 1 before the contributions to emphasize this: *“However, to the best of our knowledge, there is no prior work that uses pre-trained models in DP CL.”*
>
>
> > Figure 3 results are a bit puzzling: for the Cosine Classifier, the accuracy of the label-oracle baseline barely changes between $\epsilon=1$ and $\epsilon=8$, suggesting either the DP noise has little effect or the privacy accounting isn’t tight enough to show a trend. This deserves clarification.
>
> Thank you for raising this point.
>
> The non-DP performance of the Cosine Classifier is 79% (Table A.8) while indeed the label oracle numbers are 78.4% ($\epsilon=1$) and 79% ($\epsilon=8$) as can be seen in Table A.3. There are two reasons for this:
> * The magnitude of the added noise is small and with the label-oracle baseline Gaussian noise only gets added when the class is in the data and this is in this non-blurry setting only in one task.  For $\delta = 10^{-5}$, the standard deviation of the Gaussian noise for $\epsilon = 1.0$ is roughly 3.73, and for $\epsilon = 8.0$ is roughly 0.6.
> * The embeddings for CIFAR-100 are so well separated that the effect of the DP noise is minimal.
>
> Only when more noise gets added multiple times to each prototype the accuracy drops significantly around 72% for $\epsilon=1$ for e.g., Base Dataset or Public Labels that add 10x more noise as label-oracle. Another reason for a big difference is when the embeddings are less separated as in ImageNet-R (see Table A9): There $\epsilon=8$ has an accuracy of 46% whereas $\epsilon=1$ has 13%.
>
>
> > The experiments lack comparison to existing DP-CL baselines, which would contextualize how severe the leakage is in prior methods.
>
> Thank you for raising this point. The leakage is formally proven to hold in previous methods (Farquhar & Gal, 2019; Desai et al., 2021; Hassanpour et al., 2022; Lai et al., 2022), and any method that derives the label space directly from the sensitive data, without the need for any experiments, see Proposition 4.1 in Section 4.1. The Membership Inference Attack (MIA) attack in Figure 1 of Section 1 also shows this clearly by achieving 100% attack accuracy from observing the additional label. We also kindly refer you to the response of EmMc’s review, regarding the “Limited Scope of Baselines”.
>
> > Figure 4 (CIFAR-100): the first point left of the red dashed line shows nearly all labels (>99%) released, yet accuracy drops by more than 5%. Can you clarify why this could be the case?
>
> We apologize for the confusion due to the plot scale differences.  We updated Figure 4 (panels 2 and 4) to highlight this more by adjusting the “# Classes” axis.
>
> If you look at Tables A4, you can find the exact test accuracies (2nd best, median, and 4th best). Table A5 contains the percentage of release labels:
> * At the dashed line, the median number of released classes was 100 with 78.5% median accuracy for the cosine classifier, **the left point has a median of 90 (not > 99%)** of released labels with 71.9% median accuracy.
> * The relative drop in accuracy (78.5 - 71.9) / 78.5 is approximately 8% which is less than 10% of the relative drop in released labels.
> * This is expected because the chance of misclassification can drop when there are less classes, such as in the extreme case when there is only one class which yields 100% accuracy on that class, even if the class is hard to learn.

---

> > ### Comment · Reviewer_K2LY · 2025-11-26
> >
> > I appreciate author's responses and I thank them for the effort running more experiments and clarifying their contributions.
> >
> > I do believe the results presented in the paper are correct and solid. I am, however, still not convinced on the novelty aspect. To summarise, the way I understand the paper's contributions are:
> >
> > * Identifying privacy leakage from label space in CL - **clearly novel and important**.
> > * Addressing the leakage with private label release (S_prior, S_learned) - adopting existing DP approaches and best practices to a new problem
> > * Improving utility by using pre-trained models - I'm not familiar with CL literature, but this is again a well-known approach in DP.
> >
> > These contributions to me justify my current score of 6, and so I choose to maintain it at the moment.
> >
> > I want to, however, flag that I disagree with other reviewers on their criticisms of the paper. In many cases fellow reviewers list weaknesses that are applicable to the entire field rather than this paper in particular. I believe that within the fields of Differential Privacy for Continual Learning authors demonstrate their expertise and follow best practices of the fields.
> >
> > Some examples:
> >
> > > (**EmMc**) Assumption on Public Pre-training
> >
> > This is a common assumption in DP literature, as in most cases private data comes from a domain with wast amounts of public data (text, images, audio). The utility benefits are present even if public data is substantially different from the private distribution, as long as they share a domain
> >
> > > (**HnBC**) Strong assumptions
> >
> > The assumptions made by authors look quite standard to me
> >
> > > (**wjgU**) I do not quite follow the concept of "output label space".
> >
> > Authors use standard setup in Continual Learning, which, whilst not the most popular branch of ML, certainly has its applications.

---

> > > ### Author Response · Authors · 2025-11-27
> > >
> > > We thank the reviewer for the thoughtful follow-up, and we appreciate the reviewer's positive assessment of the paper's correctness and adherence to best practices in DP and CL.
> > >
> > > In terms of novelty, we would like to stress that we propose the first practical DP CL method. The methodology builds upon prior work, but includes a number of improvements that are critical in achieving this goal.
> > >
> > > We hope our clarifications help contextualize our contributions within the broader DP and CL literature. Thank you again for your time and thoughtful engagement with our work.

---

### Official Review · Reviewer_EmMc · 2025-11-01

**Soundness:** 3
**Presentation:** 3
**Contribution:** 2
**Rating:** 4
**Confidence:** 2

**Summary:**

This paper identifies and rigorously analyzes a previously overlooked privacy side-channel in differentially private (DP) machine learning: the **output label space** of a classifier. The authors demonstrate that even if model weights are trained under DP guarantees, **directly revealing the set of labels observed during training** can completely break DP—especially in **continual learning (CL)** settings, where the label space evolves over time.

The core contribution is twofold:
1. **Problem Identification**: The paper formalizes how releasing the empirical label set (e.g., all classes seen up to task *t*) violates DP, using a simple but devastating membership inference attack (Fig. 1).
2. **Mitigation Strategies**: Two DP-compliant alternatives are proposed:
   - **S_learned**: Use a DP mechanism (based on private partition selection) to release a noisy subset of labels, consuming part of the privacy budget.
   - **S_prior**: Use a fixed, data-independent public label space (e.g., from pre-training), remapping or dropping private labels as needed.

The authors instantiate these strategies within two practical DP-CL frameworks leveraging pre-trained ViTs:
- A **cosine similarity classifier** that accumulates DP-noisy class prototypes.
- A **PEFT ensemble** that fine-tunes lightweight adapters per task under DP-SGD.

Experiments on Split-CIFAR-100 and Split-ImageNet-R show that both strategies successfully close the privacy loophole while maintaining utility, with trade-offs depending on privacy budget, label distribution (e.g., "blurry" tasks), and model choice.

**Strengths:**

1. **Novel and Important Insight**: The identification of the output label space as a privacy side-channel is both simple and profound. It exposes a critical flaw in prior DP-CL work that assumed protecting model weights was sufficient. This has broad implications beyond CL (e.g., any DP classification with unknown label sets).
2. **Rigorous Formalization**: The attack and non-DP nature of naive label release are formally proven (Prop. 4.1). The proposed solutions are grounded in established DP theory (private partition selection) and accompanied by formal privacy proofs (Prop. 4.2, Appendix C).
3. **Practical and Well-Evaluated Solutions**: The two mitigation strategies are realistic and well-motivated. The use of pre-trained models aligns with state-of-the-art DP and CL practices. Experiments are thorough, covering idealized and realistic ("blurry") CL settings, and include ablations on privacy budget allocation and public label space size.

**Weaknesses:**

1. **Limited Scope of Baselines**: The comparison to prior DP-CL methods is indirect (via the identified flaw). A direct empirical comparison to, e.g., Desai et al. (2021) or Hassanpour et al. (2022)—even if flawed—would strengthen the practical impact claim.
2. **Assumption on Public Pre-training**: The reliance on public pre-training data (ImageNet-21k) is standard but increasingly debated. A brief discussion on the implications if pre-training data were private would be valuable.
3. **Scalability of S_prior**: While S_prior avoids privacy budget splitting, using a very large public label space (e.g., 1000×) may incur computational overhead (e.g., in the cosine classifier). The paper notes memory scales with |O_t| but doesn’t quantify runtime costs.
4. **Minor Clarity Issues**: The definition of "blurry tasks" could be clarified earlier. The distinction between task-wise adjacency and standard DP adjacency is well-handled in the appendix but could be streamlined in the main text.

**Questions:**

As described in weakness.

---

> ### Author Response · Authors · 2025-11-18
>
> We appreciate your thoughtful feedback and valuable suggestions. We will address question/comment separately:
>
> > Limited Scope of Baselines: The comparison to prior DP-CL methods is indirect (via the identified flaw). A direct empirical comparison to, e.g., Desai et al. (2021) or Hassanpour et al. (2022)—even if flawed—would strengthen the practical impact claim.
>
> We thank the reviewer for this valuable suggestion. We did not do a comparison before because prior DP CL methods are not directly compatible with pre-trained models, or that the methods assume different CL settings. For the sake of a more fair comparison, we applied the method of Desai et al. (2021), which has the closest setting to ours, to fine-tuning the same pre-trained model that we use in our experiments on CIFAR-100 and ImageNet-R. That said, the CL setting of Desai et al. (2021) is not exactly equivalent to ours because of their use of memory buffers.
>
> We implemented an enhanced version of Algorithm 1 of Desai et al. (2021) with better privacy accounting, as the original version had the performance of a random classifier. We added the baseline to Section 6.1 in the manuscript. Clearly, our methods (Cosine Classifier, PEFT Ensemble) outperforms their method on $\epsilon = 1.0, 8.0$.
>
> Hassanpour et al. (2022) has a different CL setting which is task-incremental, i.e. the learner observes the task label and it is used at inference time. Our CL setting is harder, such that we do not know what the task is at inference or test time. Furthermore, their method depends on replay memory and robust continual-learning updates built up during training, which a pretrained model lacks, making the approach incompatible with simple fine-tuning.
>
> The method of Lai et al. (2021) can not be applied to general models (see Section 4 of their paper under “Network Design”), and particularly can not be used to fine-tune pre-trained models.
>
> We also added this discussion about the differences between CL setting of prior DP CL works and ours to Section 2.
>
> > Assumption on Public Pre-training: The reliance on public pre-training data (ImageNet-21k) is standard but increasingly debated. A brief discussion on the implications if pre-training data were private would be valuable.
>
> Thanks for raising this important point. Although ImageNet-21K is large it does not reach the billion-sample scale, and there are some examples of fully compliant or low-risk datasets as an alternative, such as PASS (Pictures Without HumAns; Asano et al., 2021) and Open Images V4 (Kuznetsova et al., 2020).
>
> We point to the discussion paper of Tramèr et al, 2024 in our background, but agree that this discussion could be made more prominent, thus we changed the sentence to: *”However, if any private information is contained in the pre-training data, the DP privacy guarantees w.r.t. the fine-tuning data become meaningless (Tramèr et al., 2024), but low-risk pre-training data exist (Kuznetsova et al., 2020; Asano et al., 2021).”*
>
> > Scalability of S_prior: While S_prior avoids privacy budget splitting, using a very large public label space (e.g., 1000×) may incur computational overhead (e.g., in the cosine classifier). The paper notes memory scales with |O_t| but doesn’t quantify runtime costs.
>
> Thank you for raising this point.
>
> Regarding training the cosine classifier:
> * The pre-trained model forward pass does not depend on the number of labels.
> * The prototype noise addition scales linearly with the number of labels in the output label space $\lvert \cup_{i = 1}^{t} \mathcal{O}_i \rvert$.
> * The inference runtime scales linearly with the number of prototypes which is $\lvert \cup_{i = 1}^{t} \mathcal{O}_i \rvert$.
> * In our experiments we did not measure any significant runtime differences in terms of wall-clock runtime.
>
> On the other hand, regarding the PEFT model:
> * The runtime complexity depends on the number of FiLM parameters and the size of the last layer.
> * Theoretically speaking, since the last layer is a linear operation $Wx + b$, the rows of the $W$ and the size of the vector $b$ depends on $\lvert \cup_{i = 1}^{t} \mathcal{O}_i \rvert$
> * The gradient computations in DP-SGD for the last layer will also scale linearly with the number of labels.
> * The inference runtime also scales by the number of FiLM parameters and the size of the last layer which depends on $\lvert \cup_{i = 1}^{t} \mathcal{O}_i \rvert$.
>
> > Minor Clarity Issues: The definition of "blurry tasks" could be clarified earlier. The distinction between task-wise adjacency and standard DP adjacency is well-handled in the appendix but could be streamlined in the main text.
>
> Thanks for pointing out these issues. We edited the manuscript to explain the concept of “blurry tasks” earlier at the end of Section 3; however, regarding the DP adjacency relation, we avoided including the lengthy discussion in the main paper due to page count limitations. We included details about this in both the contributions and background.

---

> ### Author Response · Authors · 2025-11-26
>
> Dear reviewer EmMc, we updated our comment, by saying that we now include the Desai et al. (2021) baseline in the revised version of the manuscript, and that we added the discussion about the differences between CL setting of prior DP CL works and ours to Section 2.
>
> The additional code for the Desai et al. (2021) baseline can be found in: https://anonymous.4open.science/r/iclr-2026-rebuttal-058E

---

### Author Response · Authors · 2025-11-18
**Global comment**

We thank all the reviewers for their valuable suggestions which improved the clarity and readability of the paper. In the following, we list all the main changes that we have made which are marked in blue in the revision of the manuscript:

1. Section 1: In the second paragraph, we explained the model output label space earlier, as requested by reviewer wjgU, using Figure 1 to make the concept and side-channel more clear.
2. Section 1: We emphasize in the contributions at the end of Section 1 that we are the first to use pre-trained models in DP CL, as a response to the first point raised by reviewer K2LY.
3. Section 1 & 3: We add the task-wise DP formalism to the contributions of Section 1, and add more details about it in Section 3 under “Differential Privacy”, as requested by reviewer EmMc and regarding the response to the second point raised by reviewer HnBC.
4. Section 2: Added two references to safer pre-training data as raised by reviewer EmMc.  We added a paragraph to highlight the differences between our CL setting and the CL settings of prior works.
5. Section 3: We also discuss the concept of “Blurry tasks” earlier at the end of Section 3, as requested by reviewer EmMc. We added a paragraph to “Continual Learning” to explain our CL setting in further detail.
6. Section 4.3: We added a discussion on how label remapping affects utility as requested by reviewer HnBC.
7. Section 6: We updated Figure 4 to highlight better the drop in released labels left of the red dashed line in the panels 2 and 4. This is a response to the review of reviewer K2LY. We added the Desai et al. (2021) baseline as current SOTA, as requested by reviewer EmMc, demonstrating that our methods clearly outperform theirs without using memory buffers.
8. Section 7: We added a heuristic to determine the privacy budget split for $S_{learned}$ as requested by reviewer HnBC. We also added a limitation that reviewer HnBC pointed out: we do not have theoretical guarantees for finding an optimal privacy budget split between learning the labels and the DP training, but leave this as future work.
9. We fixed several typos as suggested by reviewer HnBC.
10. Appendix D: we added a subsection (D.5) about the runtime scalability of our models with respect to the number of classes, as pointed out by reviewer EmMc. We also added a subsection (D.4) for further details on the enhanced implementation of the method of Desai et al. (2021).
11. To improve the confidence in our theoretical results, we formalise the main proofs in Lean 4 which can be found in (to verify, you can paste the code in https://live.lean-lang.org/): https://anonymous.4open.science/r/iclr-2026-rebuttal-058E/DPCL.lean

The additional code for the Desai et al. (2021) baseline can be found in: https://anonymous.4open.science/r/iclr-2026-rebuttal-058E

---

### Meta-Review · Area_Chair_3b2A · 2026-01-05

**Summary:**

Reviewers agreed that the paper studies an important problem in private classification that's overlooked in prior work. However, the techniques used in the paper are well-know methods and paper doesn't have methodological novelty. This makes reviewers not enthusiastic about this paper.

**Reviewer Concerns:**

Reviewers had raised concerns about the clarity of the definitions, methods, and results which seem to be mostly resolved in the rebuttal.

The concerns about the novelty of the methods still remain.

**Reviewer Scores:**

I believe all reviewers would remain on their current scores.

---

### Decision · Program_Chairs · 2026-01-26

Reject